# Sample Complexity Reduction via Policy Difference Estimation in Tabular Reinforcement Learning

**Adhyyan Narang**
University of Washington
adhyyan@uw.edu

**Andrew Wagenmaker**
University of California, Berkeley
ajwagen@berkeley.edu

**Lillian J. Ratliff**
University of Washington
ratliffl@uw.edu

**Kevin Jamieson**
University of Washington
jamieson@cs.washington.edu

## Abstract

In this paper, we study the non-asymptotic sample complexity for the pure exploration problem in contextual bandits and tabular reinforcement learning (RL): identifying an $\epsilon$-optimal policy from a set of policies $\Pi$ with high probability. Existing work in bandits has shown that it is possible to identify the best policy by estimating only the *difference* between the behaviors of individual policies– which can be substantially cheaper than estimating the behavior of each policy directly —yet the best-known complexities in RL fail to take advantage of this, and instead estimate the behavior of each policy directly. Does it suffice to estimate only the differences in the behaviors of policies in RL? We answer this question positively for contextual bandits, but in the negative for tabular RL, showing a separation between contextual bandits and RL. However, inspired by this, we show that it *almost* suffices to estimate only the differences in RL: if we can estimate the behavior of a *single* reference policy, it suffices to only estimate how any other policy deviates from this reference policy. We develop an algorithm which instantiates this principle and obtains, to the best of our knowledge, the tightest known bound on the sample complexity of tabular RL.

## 1 Introduction

Online platforms, such as AirBnB, often try to improve their services by A/B testing different marketing strategies. Based on the inventory, their strategy could include emphasizing local listings versus tourist destinations, providing discounts for longer stays, or de-prioritizing homes that have low ratings. In order to choose the best strategy, the standard approach would be to apply each strategy sequentially and measure outcomes. However, recognize that the choice of strategy (policy) affects the future inventory (state) of the platform. This complex interaction between different strategies makes it difficult to estimate the impact of any strategy, if it were to be applied independently. To address this, we can model the platform as an Markov Decision Process (MDP) with an observed state [17, 15] and a finite set of policies $\Pi$ corresponding to possible strategies. We wish to collect data by playing *exploratory* actions which will enable us to estimate the true value of each policy $\pi \in \Pi$, and identify the best policy from $\Pi$ as quickly as possible.

In addition to A/B testing, similar challenges arise in complex medical trials, learning robot policies to pack totes, and autonomous navigation in unfamiliar environments. All of these problems can be formally modeled as the PAC (Probably Approximately Correct) policy identification problem in reinforcement learning (RL). An algorithm is said to be $(\epsilon, \delta)$-PAC if, given a set of policies $\Pi$, it returns a policy $\pi \in \Pi$ that performs within $\epsilon$ of the optimal policy in $\Pi$, with probability $1 - \delta$. The

goal is to satisfy this condition whilst minimizing the number of interactions with the environment (the *sample complexity*).

Traditionally, prior work has aimed to obtain *minimax* or *worst-case* guarantees for this problem—guarantees that hold across *all* environments within a problem class. Such worst-case guarantees typically scale with the "size" of the environment, for example, scaling as $\mathcal{O}(\text{poly}(S, A, H)/\epsilon^2)$, for environments with $S$ states, $A$ actions, horizon $H$. While guarantees of this form quantify which classes of problems are efficiently learnable, they fail to characterize the difficulty of particular problem instances—producing the same complexity on both "easy" and "hard" problems that share the same "size". This is not simply a failure of analysis—recent work has shown that algorithms that achieve the minimax-optimal rate could be very suboptimal on particular problem instances [46]. Motivated by this, a variety of recent work has sought to obtain *instance-dependent* complexity measures that capture the hardness of learning each particular problem instance. However, despite progress in this direction, the question of the *optimal* instance-dependent complexity has remained elusive, even in tabular settings.

Towards achieving instance-optimality in RL, the key question is: *what aspects* of a given environment must be learned, in order to choose a near-optimal policy? In the simpler bandit setting, this question has been settled by showing that it is sufficient to learn the *differences* between values of actions rather than learning the value of each individual action: it is only important whether a given action's value is greater or lesser than that of other actions. This observation can yield significant improvements in sample efficiency [37, 16, 13, 30]. Precisely, the best-known complexity measures in the bandit setting scale as:

$$\inf_{\pi_{\exp}} \max_{\pi \in \Pi} \frac{\|\phi^\pi - \phi^\star\|^2_{\Lambda(\pi_{\exp})^{-1}}}{\Delta(\pi)^2}, \tag{1.1}$$

where $\phi^\pi$ is the feature vector of action $\pi$, $\phi^\star$ the feature vector of the optimal action, $\Delta(\pi)$ is the suboptimality of action $\pi$. Here, $\Lambda(\pi_{\exp})$ are the covariates induced by $\pi_{\exp}$, our distribution of exploratory actions. The denominator of this expression measures the performance gap between action $\pi$ and the optimal action. The numerator measures the variance of the estimated (from data collected by $\pi_{\exp}$) difference in values between $(\pi, \pi^\star)$. The max over actions follows because to choose the best action, we have to rule out every sub-optimal action from the set of candidates $\Pi$; the infimum optimizes over data collection strategies.

In contrast, in RL, instead of estimating the difference between policy values *directly*, the best known algorithms simply estimate the value of each individual policy *separately* and then take the difference. This obtains instance-dependent complexities which scale as follows [42]:

$$\sum_{h=1}^{H} \inf_{\pi_{\exp}} \max_{\pi \in \Pi} \frac{\|\phi_h^\pi\|^2_{\Lambda_h(\pi_{\exp})^{-1}} + \|\phi_h^\star\|^2_{\Lambda_h(\pi_{\exp})^{-1}}}{\Delta(\pi)^2} \tag{1.2}$$

where $\phi_h^\pi$ is the state-action visitation of policy $\pi$ at step $h$. Since now the difference is calculated *after* estimation, the variance of the difference is the sum of the individual variances of the estimates of each policy, captured in the numerator of (1.2). Comparing the numerator of (1.2) to that of (1.1) begs the question: in RL can we estimate the *difference* of policies directly to reduce the sample complexity of RL?

To motivate why this distinction is important, consider the tabular MDP example of Figure 1. In this example, the agent starts in state $s_1$, takes one of three actions, and then transitions to one of states $s_2, s_3, s_4$. Consider the policy set $\Pi = \{\pi_1, \pi_2\}$, where $\pi_1$ always plays action $a_1$, and $\pi_2$ is identical, except plays actions $a_2$ in the red states. If $\phi_h^{\pi_i} \in \triangle_{\mathcal{S} \times \mathcal{A}}$ denotes the state-action visitations of policy $\pi_i$ at time $h = 1, 2$, then we see that $\phi_1^{\pi_1} = \phi_1^{\pi_2}$ since $\pi_1$ and $\pi_2$ agree on the action in $s_1$. But $\phi_2^{\pi_1} \neq \phi_2^{\pi_2}$ as their actions differ on the red states.

Since these red states will be reached with probability at most $3\epsilon$, the norm of the *difference*

$$\|\phi_2^\pi - \phi_2^\star\|^2_{\Lambda_2(\pi_{\exp})^{-1}} = \sum_{s,a} \frac{(\phi_2^\pi(s,a) - \phi_2^\star(s,a))^2}{\phi_2^{\pi_{\exp}}(s,a)}$$

is significantly less than the sum of the individual norms

$$\|\phi_2^\pi\|^2_{\Lambda_2(\pi_{\exp})^{-1}} + \|\phi_2^\star\|^2_{\Lambda_2(\pi_{\exp})^{-1}} = \sum_{s,a} \frac{\phi_2^\pi(s,a)^2 + \phi^\star(s,a)^2}{\phi_2^{\pi_{\exp}}(s,a)}.$$

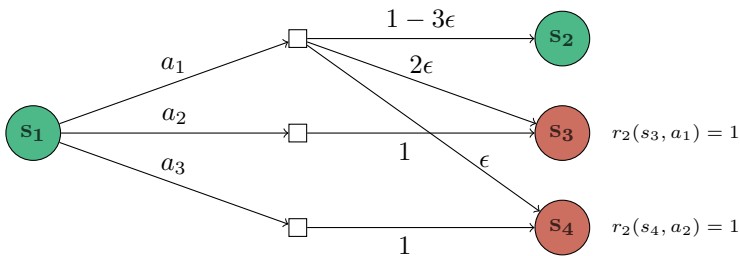

Figure 1: A motivating example for differences. The rewards for all actions other than the ones specified in the figure are 0. Define policy set $\Pi = \{\pi_1, \pi_2\}$ so that $\pi_1$ always plays $a_1$, whereas $\pi_2$ plays $a_1$ on green states but $a_2$ on red states. The difference of their state-action visitation probabilities is only non-zero in states $s_3, s_4$ and are just $O(\epsilon)$ apart.

Intuitively, to minimize differences $\pi_{\exp}$ can explore just states $s_3, s_4$ where the policies differ, whereas minimizing the individual norms requires wasting lots of energy in state $s_2$ where the two policies and the difference is zero. Formally:

**Proposition 1.** *On the MDP and policy set $\Pi$ from Figure 1, we have that*

$$\inf_{\pi_{\exp}} \max_{\pi \in \Pi} \|\phi_2^\pi\|_{\Lambda_2(\pi_{\exp})^{-1}}^2 \geq 1 \quad \text{and} \quad \inf_{\pi_{\exp}} \max_{\pi \in \Pi} \|\phi_2^\star - \phi_2^\pi\|_{\Lambda_2(\pi_{\exp})^{-1}}^2 \leq 15\epsilon^2.$$

Proposition 1 shows that indeed, the complexity of the form Equation (1.1) (generalized to RL) in terms of differences could be significantly tighter than Equation (1.2); in this case, it is a factor of $\epsilon^2$ better. But achieving a sample complexity that depends on the differences requires more than just a better analysis: it requires a new estimator and an algorithm to exploit it.

**Contributions.** In this work, we aim to understand whether such a complexity is achievable in RL. Letting $\rho_\Pi$ denote the generalization of (1.1) to the RL case—that is, (1.2) but with $\|\phi_h^\pi\|_{\Lambda_h(\pi_{\exp})^{-1}}^2$ replaced by $\|\phi_h^\pi - \phi_h^{\pi^\star}\|_{\Lambda_h(\pi_{\exp})^{-1}}^2$, our contributions are as follows:

1. In the Tabular RL case, [2] recently showed that $\rho_\Pi$ is a lower bound on the sample complexity of RL by characterizing the difficulty of learning the unknown reward function; however, they did not resolve whether it is achievable when the state-transitions are unknown as well. We provide a lower bound which demonstrates that $\mathcal{O}(\rho_\Pi)$ is *not* sufficient for learning with state transitions.

2. We provide an algorithm PERP, which first learns the behavior a particular reference policy $\bar\pi$, and then estimates the difference in behavior between $\bar\pi$ and every other policy $\pi$, rather than estimating the behavior of each $\pi$ directly.

3. In the case of tabular RL, we show that PERP obtains a complexity that scales with $\mathcal{O}(\rho_\Pi)$, in addition to an extra term which measures the cost of learning the behavior of the reference policy $\bar\pi$. We argue that this additional term is critical to achieving instance-optimal guarantees in RL, and that PERP leads to improved complexities over existing work.

4. In the contextual bandit setting, we provide an upper bound that scales (up to lower order terms) as $\mathcal{O}(\rho_\Pi)$ for the *unknown-context* distribution case. This matches the lower bound from [30] for the known context distribution case, thus showing that $\rho_\Pi$ is necessary and sufficient in contextual bandits even when the context distribution is unknown. Hence, we observe a qualitative information-theoretic separation between contextual bandits and RL.

The key insight from our work is that it does not suffice to *only* learn the differences between policy values in RL, but it *almost* suffices to—if we can learn how a single policy behaves, it suffices to learn the difference between this policy and every other policy.

## 2 Related Work

The reinforcement learning literature is vast, and here we focus on results in tabular RL and instance-dependent guarantees in RL.

**Minimax Guarantees Tabular RL.** Finite-time minimax-style results on policy identification in tabular MDPs go back to at least the late 90s and early 2000s [24, 26, 25, 8, 21]. This early work was built upon and refined by a variety of other works over the following decade [38, 4, 34, 39], leading up to works such as [28, 9], which establish sample complexity bounds of $\mathcal{O}(S^2 A \cdot \mathrm{poly}(H)/\epsilon^2)$. More recently, [10, 11, 33] have proposed algorithms which achieve the optimal dependence of $\mathcal{O}(SA \cdot \mathrm{poly}(H)/\epsilon^2)$, with [11, 33] also achieving the optimal $H$ dependence. The question of regret minimization is intimately related to that of policy identification—any low-regret algorithm can be used to obtain a near-optimal policy via an online-to-batch conversion [19]. Early examples of low-regret algorithms in tabular MDPs are [3, 4, 5, 48], with more recent works removing the horizon dependence or achieving the optimal lower-order terms as well [50, 51]. Recently, [6, 7] provide minimax guarantees in the multi-task RL setting as well.

**Instance-Dependence in RL.** While the problem of obtaining worst-case optimal guarantees in tabular RL is nearly closed, we are only beginning to understand what types of instance-dependent guarantees are possible. In the setting of regret minimization, [35, 14] achieve instance-optimal regret for tabular RL asymptotically. Simchowitz and Jamieson [36] show that standard optimistic algorithms achieve regret bounded as $\mathcal{O}(\sum_{s,a,h} \frac{\log K}{\Delta_h(s,a)})$, a result later refined by [47, 12]. In settings of RL with linear function approximation, several works achieve instance-dependent regret guarantees [18, 44]. Recently, Wagenmaker and Foster [45] achieved finite-time guarantees on instance-optimal regret in general decision-making settings, a setting encompassing much of RL.

On the policy identification side, early works obtaining instance-dependent guarantees for tabular MDPs include [49, 20, 31, 32], but they all exhibit shortcomings such as requiring access to a generative model or lacking finite-time results. The work of Wagenmaker et al. [46] achieves a finite-time instance-dependent guarantee for tabular RL, introducing a new notion of complexity, the *gap-visitation complexity*. In the special case of deterministic, tabular MDPs, Tirinzoni et al. [41] show matching finite-time instance-dependent upper and lower bounds. For RL with linear function approximation, [42, 43] achieve instance-dependent guarantees on policy identification, in particular, the complexity given in (1.2), and propose an algorithm, PEDEL, which directly inspires our algorithmic approach. On the lower bound side, Al-Marjani et al. [2] show that $\rho_\Pi$ is necessary for tabular RL, but fail to close the aforementioned gap between $\rho_\Pi$ and (1.2). We will show instead that this gap is real and both the lower bound of Al-Marjani et al. [2] and upper bound of Wagenmaker and Jamieson [42] are loose.

Several works on linear and contextual bandits are also relevant. In the seminal work, [37] posed the best-arm identification problem for linear bandits and beautifully argued—without proof—that estimating differences were crucial and that (1.1) ought to be the true sample complexity of the problem. Over time, this conjecture was affirmed and generalized [16, 13, 22]. This improved understanding of pure-exploration directly led to instance-dependent optimal linear bandit algorithms for regret [29, 27]. More recently, contextual bandits have also been given a similar treatment [40, 30].

# 3 Preliminaries and Problem Setting

Let $\|x\|_\Lambda^2 = x^\top \Lambda x$ for any $(x, \Lambda)$. We let $\mathbb{E}_\pi$ denote the probability measure induced by playing policy $\pi$ in our MDP.

**Tabular Markov Decision Processes.** We study episodic, finite-horizon, time inhomogenous and tabular Markov Decision Processes (MDPs), denoted by the tuple $(\mathcal{S}, \mathcal{A}, H, \{P_h\}_{h=1}^H, \{\nu_h\}_{h=1}^H)$ where the state space $\mathcal{S}$ and action space $\mathcal{A}$ are finite, $H$ is the horizon, $P_h \in \mathbb{R}^{S \times SA}$ denote the transition matrix at stage $h$ where $[P_h]_{s',sa} = \mathbb{P}(s_{h+1} = s' | s_h = s, a_h = a)$, and $\nu_h(s,a) \in \triangle_{[0,1]}$ denote the distribution over reward at stage $h$ when the state of the system is $s$ and action $a$ is chosen. Let $r_h(s,a)$ be the expectation of a reward drawn from $\nu_h(s,a)$. We assume that every episode starts in state $s_1$, and that $\nu_h$ and $P_h$ are initially unknown and must be estimated over time.

Let $\pi = \{\pi_h\}_{h=1}^H$ denote a policy mapping states to actions, so that $\pi_h(s) \in \triangle_\mathcal{A}$ denotes the distribution over actions for the policy at $(s,h)$; when the policy is deterministic, $\pi_h(s) \in \mathcal{A}$ outputs a single action. An episode begins in state $s_1$, the agent takes action $a_1 \sim \pi_1(s_1)$ and receives reward $R_1 \sim \nu_1(s_1, a_1)$ with expectation $r_1(s_1, a_1)$; the environment transitions to state $s_2 \sim P_h(s_1, a_1)$. The process repeats until timestep $H$, at which point the episode ends and the agent returns to state $s_1$. Let $V_h^\pi(s) = \mathbb{E}_\pi[\sum_{h'=h}^H r_{h'}(s_{h'}, a_{h'}) | s_h = s]$, $V_0^\pi$ the total expected reward, $V_0^\pi := V_1^\pi(s_0)$,

and $Q_h^\pi(s,a) = \mathbb{E}_\pi[\sum_{h'=h}^H r_{h'}(s_{h'}, a_{h'})|s_h = s, a_h = a]$ the amount of reward we expect to collect if we are in state $s$ at step $h$, play action $a$ and then play policy $\pi$ for the remainder of the episode. Note that we can understand these functions as $S$ and $SA$-dimensional vectors respectively. We use $V^\pi = V_0^\pi$ when clear from context.

We call $w_h^\pi \in \triangle_S$ the *state visitation vector* at step $h$ for policy $\pi$, so that $w_h^\pi(s)$ captures the probability that policy $\pi$ would land in state $s$ at step $h$ during an episode. Let $\boldsymbol{\pi}_h \in \mathbb{R}^{SA \times S}$ denote the policy matrix for policy $\pi$, that maps states to state-actions as follows

$$[\boldsymbol{\pi}_h]_{(s,a),s'} = \mathbb{I}(s = s')[\pi_h(s)]_a.$$

Denote $\phi_h^\pi \in \triangle_{SA}$ as $\phi_h^\pi := \boldsymbol{\pi}_h w_h^\pi$ as the *state-action visitation vector*: $\phi_h^\pi(s,a)$ measures the the probability that policy $\pi$ would land in state $s$ and play action $a$ at step $h$ during an episode. From these definitions, it follows that $[P_h \phi_h^\pi]_s = [P_h \boldsymbol{\pi}_h w_h^\pi]_s = w_{h+1}^\pi(s)$. For policy $\pi$, denote the covariance matrix at timestep $h$ as $\Lambda_h(\pi) = \sum_{s,a} \phi_h^\pi(s,a) \mathbf{e}_{(s,a)} \mathbf{e}_{(s,a)}^\top$.

**$(\epsilon, \delta)$-PAC Best Policy Identification.** For a collection of policies $\Pi$, define $\pi^\star := \arg\max_{\pi \in \Pi} V^\pi$ as the optimal policy, $V^\star$ its value, and $\phi_h^\star$ as its state-action visitation vector. Let $\Delta_{\min} := \min_{\pi \in \Pi \setminus \{\pi^\star\}} V^\star - V^\pi$ in the case when $\pi^\star$ is unique, and otherwise $\Delta_{\min} := 0$. Define $\Delta(\pi) := \max\{V^\star - V^\pi, \Delta_{\min}\}$. Given $\epsilon \geq 0, \delta \in (0,1)$ an algorithm is said to be $(\epsilon, \delta)$-PAC if at a stopping time $\tau$ of its choosing, it returns a policy $\widehat{\pi}$ which satisfies $\Delta(\pi) \leq \epsilon$ with probability $1 - \delta$. Our goal is to obtain an $(\epsilon, \delta)$-PAC algorithm that minimizes $\tau$. A fundamental complexity measure used throughout this work is defined as

$$\rho_\Pi := \sum_{h=1}^H \inf_{\pi_{\exp}} \max_{\pi \in \Pi} \frac{\|\phi_h^\star - \phi_h^\pi\|_{\Lambda_h(\pi_{\exp})^{-1}}^2}{\max\{\epsilon^2, \Delta(\pi)^2\}} \quad \text{for} \quad \|\phi_h^\star - \phi_h^\pi\|_{\Lambda_h(\pi_{\exp})^{-1}}^2 := \sum_{s,a} \frac{(\phi_h^\star(s,a) - \phi_h^\pi(s,a))^2}{\phi_h^{\pi_{\exp}}(s,a)}$$

where the infimum is over all exploration policies $\pi_{\exp}$ (not necessarily just those in $\Pi$). Recall that for $\epsilon = 0$, [2] showed any $(\epsilon, \delta)$-PAC algorithm satisfies $\mathbb{E}[\tau] \geq \rho_\Pi \log(\frac{1}{2.4\delta})$.

## 4 What is the Sample Complexity of Tabular RL?

In this section, we seek to understand the complexity of tabular RL. We start by showing that $\rho_\Pi$ is not sufficient. We have the following result.

**Lemma 1.** *For the MDP $\mathcal{M}$ and policy set $\Pi$ from Figure 1,*

1. $\sum_{h=1}^H \inf_{\pi_{\exp}} \max_{\pi \in \Pi} \frac{\|\phi_h^\star - \phi_h^\pi\|_{\Lambda_h(\pi_{\exp})^{-1}}^2}{\max\{\epsilon^2, \Delta(\pi)^2\}} \leq 15$,

2. *Any $(\epsilon, \delta)$-PAC algorithm must collect at least $\mathbb{E}^{\mathcal{M}}[\tau] \geq \frac{1}{\epsilon} \cdot \log \frac{1}{2.4\delta}$. samples.*

Where does the additional complexity arise on the instance of Figure 1? As described in the introduction, $\pi_1$ and $\pi_2$ differ only on the red states, and a complexity scaling as $\rho_\Pi$ quantifies only the difficulty of distinguishing $\{\pi_1, \pi_2\}$ on these states. Note that on this example $\pi_1$ plays the optimal action in state $s_3$ and a suboptimal action in state $s_4$, and $\pi_2$ plays a suboptimal action in $s_3$ and the optimal action in $s_4$. The total reward of policy $\pi_1$ is therefore equal to the reward achieved at state $s_3$ times the probability it reaches state $s_3$, and the total reward of policy $\pi_2$ is the reward achieved at state $s_4$ times the probability it reaches state $s_4$. Here, $\rho_\Pi$ would quantify the difficulty of learning the reward achieved at each state. However, it fails to quantify *the probability of reaching each state*, since this depends on the behavior at step 1, not step 2.

Thus, on this example, to determine whether $\pi_1$ or $\pi_2$ is optimal, we must pay some additional complexity to learn the outgoing transitions from the initial state, giving rise to the lower bound in Lemma 1. Inspecting the lower bound of [2], one realizes that the construction of this lower bound only quantifies the cost of learning the reward distributions $\{\nu_h\}_h$ and *not* the state transition matrices $\{P_h\}_h$. On examples such as Figure 1, this lower bound then does not quantify the cost of learning the probability of visiting each state, which we've argued is necessary. We therefore conclude that, while $\rho_\Pi$ may be enough for learning the rewards, it is *not* sufficient for solving the full tabular RL problem. Our main algorithm builds on this intuition, and, in addition to estimating the rewards, aims to estimate where policies visit as efficiently as possible.

## 4.1 Main Result

If $\rho_\Pi$ is not achievable as the sample complexity for Tabular RL, what is the best that we can do? In this section, we answer this question with our sample complexity bound; we later describe the algorithmic insights that enable us to achieve this result in the following section. First, for any $\pi, \bar{\pi} \in \Pi$, we define

$$U(\pi, \bar{\pi}) := \sum_{h=1}^{H} \mathbb{E}_{s_h \sim w_h^{\bar{\pi}}}[(Q_h^\pi(s_h, \pi_h(s_h)) - Q_h^\pi(s_h, \bar{\pi}_h(s_h)))^2]. \tag{4.1}$$

Now, we state our main result.

**Theorem 1.** *There exists an algorithm (Algorithm 1) which, with probability at least $1 - 2\delta$, finds an $\epsilon$-optimal policy and terminates after collecting at most*

$$\sum_{h=1}^{H} \inf_{\pi_{\exp}} \max_{\pi \in \Pi} \frac{H^4 \|\phi_h^\star - \phi_h^\pi\|_{\Lambda_h(\pi_{\exp})^{-1}}^2}{\max\{\epsilon^2, \Delta(\pi)^2\}} \cdot \iota\beta^2 + \max_{\pi \in \Pi} \frac{HU(\pi, \pi^\star)}{\max\{\epsilon^2, \Delta(\pi)^2\}} \log \frac{H|\Pi|\iota}{\delta} + \frac{C_{\text{poly}}}{\max\{\epsilon^{\frac{5}{3}}, \Delta_{\min}^{\frac{5}{3}}\}}$$

*episodes, for $C_{\text{poly}} := \text{poly}(S, A, H, \log 1/\delta, \iota, \log |\Pi|)$, $\beta := C\sqrt{\log(\frac{SH|\Pi|}{\delta} \cdot \frac{1}{\Delta_{\min} \vee \epsilon})}$ and $\iota := \log \frac{1}{\Delta_{\min} \vee \epsilon}$.*

Theorem 1 shows that, up to terms lower-order in $\epsilon$ and $\Delta_{\min}$, $\rho_\Pi$ is almost sufficient, if we are willing to pay for an additional term scaling as $U(\pi, \pi^\star)/\Delta(\pi)^2$. Recognize the similarity of this term to the that from the performance difference lemma: if there were no square inside the expectation, the quantity $U(\pi, \pi^\star)$ would be equal to $\Delta(\pi)$. However, the square may change the scaling in some instances. Below, Lemma 2 shows that there exist settings where the complexity of Theorem 1 could be significantly tighter than Equation (1.2), the complexity achieved by the PEDEL algorithm of [42]. We revisit the instance from Figure 1 to show this; recall from Lemma 1 that the first term from Theorem 1 is a universal constant for this instance.

**Lemma 2.** *On MDP $\mathcal{M}$ and policy set $\Pi$ from Figure 1, we have:*

1. $\max_{\pi \in \Pi} \frac{HU(\pi, \pi^\star)}{\max\{\epsilon^2, \Delta(\pi)^2\}} = \frac{3H}{\epsilon}$,

2. $\sum_{h=1}^{H} \inf_{\pi_{\exp}} \max_{\pi \in \Pi} \frac{\|\phi_h^\pi\|_{\Lambda_h(\pi_{\exp})^{-1}}^2}{\max\{\epsilon^2, \Delta(\pi)^2\}} \geq \frac{H}{\epsilon^2}$.

Furthermore, the complexity of Theorem 1 is never worse than Equation (1.2).

**Lemma 3.** *For any MDP instance and policy set $\Pi$, we have that*

$$\max\left\{ \sum_{h=1}^{H} \inf_{\pi_{\exp}} \max_{\pi \in \Pi} \frac{\|\phi_h^\star - \phi_h^\pi\|_{\Lambda_h(\pi_{\exp})^{-1}}^2}{\max\{\epsilon^2, \Delta(\pi)^2\}}, \frac{HU(\pi, \pi^\star)}{\max\{\epsilon^2, \Delta(\pi)^2\}} \right\} \leq \sum_{h=1}^{H} \inf_{\pi_{\exp}} \max_{\pi \in \Pi} \frac{\|\phi_h^\pi\|_{\Lambda_h(\pi_{\exp})^{-1}}^2}{\max\{\epsilon^2, \Delta(\pi)^2\}}.$$

We briefly remark on the lower-order term for Theorem 1, $\frac{C_{\text{poly}}}{\max\{\epsilon^{5/3}, \Delta_{\min}^{5/3}\}}$. Note that for small $\epsilon$ or $\Delta_{\min}$, this term will be dominated by the leading-order terms, which scale with $\min\{\epsilon^{-2}, \Delta_{\min}^{-2}\}$. While we make no claims on the tightness of this term, we note that recent work has shown that some lower-order terms are necessary for achieving instance-optimality [45].

## 4.2 The Main Algorithmic Insight: The Reduced-Variance Difference Estimator

In this section, we describe how we can estimate the difference between the values of policies directly, and provide intuition for why this results in the two main terms in Theorem 1. Fix any reference policy $\bar{\pi}$ and logging policy $\mu$ (neither are necessarily in $\Pi$). Here $\mu$ can be thought of as playing the role of $\pi_{\exp}$. Or, we can consider the A/B testing scenario from the introduction, where a policy $\mu$ is taking random actions and one wishes to perform off-policy estimation over some set of policies $\Pi$ [17, 15]. For any $s \in \mathcal{S}$, we define

$$\delta_h^\pi(s) := w_h^\pi(s) - w_h^{\bar{\pi}}(s)$$

as the difference in state-visitations of policy $\pi$ from reference policy $\bar{\pi}$, and $\delta_h^\pi \in \mathbb{R}^S$ as the vectorization of $\delta_h^\pi(s')$.

**Policy selection rule.** First, we describe our procedure of data collection and estimation. We collect $K_{\bar{\pi}}$ trajectories from $\bar{\pi}$ and $K_\mu$ trajectories from $\mu$, and let $\{\widehat{w}_h^{\bar{\pi}}(s)\}_{s,h}$ denote the empirical state visitations from playing $\bar{\pi}$. From the data collected by playing $\mu$, we construct estimates $\{\widehat{P}_h(s'|s,a)\}_{s,a,s',h}$ of the transition matrices. Note that $\widehat{w}_h^{\bar{\pi}}(s)$ simply counts visitations, so that $\mathbb{E}[(\widehat{w}_h^{\bar{\pi}}(s) - w_h^{\bar{\pi}}(s))^2] \leq \frac{w_h^{\bar{\pi}}(s)}{K_{\bar{\pi}}}$ for all $h, s$. Define estimated state visitations for policy $\pi$ in terms of deviations from $\bar{\pi}$ as $\widehat{w}_h^\pi := \widehat{w}_h^{\bar{\pi}} + \widehat{\delta}_h^\pi$. Here, $\widehat{\delta}_h^\pi$ is defined recursively as:

$$\widehat{\delta}_{h+1}^\pi := \widehat{P}_h \boldsymbol{\pi}_h \widehat{\delta}_h^\pi + \widehat{P}_h(\boldsymbol{\pi}_h - \bar{\boldsymbol{\pi}}_h)\widehat{w}_h^{\bar{\pi}}$$

Then, assuming, for simplicity, that rewards are known, we recommend the following policy:

$$\widehat{\pi} = \arg\max_{\pi \in \Pi} \widehat{D}^\pi \qquad \text{where} \qquad \widehat{D}^\pi := \sum_{h=1}^H \langle r_h, \boldsymbol{\pi}_h \widehat{\delta}_h^\pi \rangle - \langle r_h, (\bar{\boldsymbol{\pi}}_h - \boldsymbol{\pi}_h)\widehat{w}_h^{\bar{\pi}} \rangle$$

**Sufficient condition for $\epsilon$-optimality.** Here, we show that if

$$\forall \pi \in \Pi, \qquad |\widehat{D}^\pi - D^\pi| \leq \frac{1}{3} \max\{\epsilon, \Delta(\pi)\} \tag{4.2}$$

then $\widehat{\pi}$ is $\epsilon$-optimal. First, write the difference between values of policies $\pi$ and $\bar{\pi}$ as:

$$\begin{aligned}
D^\pi := V_0^\pi - V_0^{\bar{\pi}} &= \sum_{h=1}^H \langle r_h, \boldsymbol{\pi}_h w_h^\pi \rangle - \sum_{h=1}^H \langle r_h, \bar{\boldsymbol{\pi}}_h w_h^{\bar{\pi}} \rangle \\
&= \sum_{h=1}^H \langle r_h, \boldsymbol{\pi}_h \delta_h^\pi \rangle - \langle r_h, (\bar{\boldsymbol{\pi}}_h - \boldsymbol{\pi}_h)w_h^{\bar{\pi}} \rangle.
\end{aligned} \tag{4.3}$$

Then, it is easy to verify that if $|\widehat{D}^\pi - D^\pi| \leq 1/3 \, \Delta(\pi)$, then $\widehat{D}^{\pi^\star} - \widehat{D}^\pi \geq 0$; hence, $\widehat{\pi} \neq \pi$. Hence, under Condition (4.2), either $\widehat{\pi} = \pi^\star$ or or $|\widehat{D}^\pi - D^\pi| \leq \epsilon$. In the first case, clearly $\widehat{\pi}$ is $\epsilon$-optimal. In the second case, we can add and subtract terms to write

$$V^\star - V^{\widehat{\pi}} \leq |D^{\pi^\star} - \widehat{D}^{\pi^\star}| + \widehat{D}^{\pi^\star} - \widehat{D}^{\widehat{\pi}} + |\widehat{D}^{\widehat{\pi}} - D^{\widehat{\pi}}| \leq \frac{2\epsilon}{3} + \widehat{D}^{\pi^\star} - \widehat{D}^{\widehat{\pi}} \leq \frac{2\epsilon}{3}.$$

The last inequality follows since $\widehat{\pi}$ maximizes $\widehat{D}^\pi$. Hence, $\widehat{\pi}$ would be $\epsilon$-optimal in this case as well.

**Sample complexity.** Now, we characterize how many samples must be collected from $\mu$ and $\bar{\pi}$ in order to meet Condition (4.2). After dropping some lower-order terms and unrolling the recursion (see Section A for details), we observe that

$$\begin{aligned}
\widehat{\delta}_{h+1}^\pi - \delta_{h+1}^\pi &\approx (\widehat{P}_h - P_h)(\phi_h^\pi - \phi_h^{\bar{\pi}}) + P_h(\boldsymbol{\pi}_h - \bar{\boldsymbol{\pi}}_h)(\widehat{w}_h^{\bar{\pi}} - w_h^{\bar{\pi}}) + P_h \boldsymbol{\pi}_h(\widehat{\delta}_h^\pi - \delta_h^\pi) \\
&= \sum_{k=0}^h \big(\prod_{j=k+1}^h P_j \boldsymbol{\pi}_j\big)\big((\widehat{P}_k - P_k)(\phi_k^\pi - \phi_k^{\bar{\pi}}) + P_k(\boldsymbol{\pi}_k - \bar{\boldsymbol{\pi}}_k)(\widehat{w}_k^{\bar{\pi}} - w_k^{\bar{\pi}})\big).
\end{aligned}$$

After manipulating this expression a bit more, we observe that

$$\sum_{h=1}^H \langle r_h, \boldsymbol{\pi}_h(\widehat{\delta}_h^\pi - \delta_h^\pi) \rangle = \sum_{k=0}^{H-1} \langle V_{k+1}^\pi, (\widehat{P}_k - P_k)(\phi_k^\pi - \phi_k^{\bar{\pi}}) + P_k(\boldsymbol{\pi}_k - \bar{\boldsymbol{\pi}}_k)(\widehat{w}_k^{\bar{\pi}} - w_k^{\bar{\pi}}) \rangle$$

Recognizing $Q_h^\pi = r_h + P_h^\top V_{h+1}^\pi$,

$$\begin{aligned}
|\widehat{D}^\pi - D^\pi| &= \left| \sum_{h=1}^H \langle r_h, \boldsymbol{\pi}_h(\widehat{\delta}_h^\pi - \delta_h^\pi) \rangle + \langle r_h, (\boldsymbol{\pi}_h - \bar{\boldsymbol{\pi}}_h)(\widehat{w}_h^{\bar{\pi}} - w_h^{\bar{\pi}}) \rangle \right| \\
&= \left| \sum_{h=0}^{H-1} \langle V_{h+1}^\pi, (\widehat{P}_h - P_h)(\phi_h^\pi - \phi_h^{\bar{\pi}}) \rangle + \langle r_h + P_h^\top V_{h+1}^\pi, (\boldsymbol{\pi}_h - \bar{\boldsymbol{\pi}}_h)(\widehat{w}_h^{\bar{\pi}} - w_h^{\bar{\pi}}) \rangle \right|
\end{aligned}$$

We can bound this as:

$$\begin{aligned}
&\lesssim \sqrt{H^2 \sum_{h=0}^{H-1} \sum_{s,a} \frac{(\phi_h^\pi(s,a) - \phi_h^{\bar{\pi}}(s,a))^2}{K_\mu \mu_h(s,a)}} + \sqrt{\sum_{h=0}^{H-1} \sum_s \big(Q_h^\pi(s,\pi_h(s)) - Q_h^\pi(s,\bar{\pi}_h(s))\big)^2 \frac{w_h^{\bar{\pi}}(s)}{K_{\bar{\pi}}}} \\
&= \sqrt{H^2 \sum_{h=0}^{H-1} \frac{\|\phi_h^\pi - \phi_h^{\bar{\pi}}\|_{\Lambda_h(\mu)^{-1}}^2}{K_\mu}} + \sqrt{\frac{U(\pi,\bar{\pi})}{K_{\bar{\pi}}}}.
\end{aligned}$$

Here, we applied Bernstein's inequality and observed that $\sum_{s'} V_{h+1}^\pi(s')^2 P_h(s'|s,a) \leq H^2$. Now, we have that if

$$K_\mu \gtrsim \max_{\pi \in \Pi} \sum_{h=0}^{H-1} \frac{H^2 \|\phi_h^\pi - \phi_h^{\bar\pi}\|_{\Lambda_h(\mu)^{-1}}^2}{\max\{\epsilon^2, \Delta(\pi)^2\}} \quad \text{and} \quad K_{\bar\pi} \gtrsim \max_{\pi \in \Pi} \frac{U(\pi, \bar\pi)}{\max\{\epsilon^2, \Delta(\pi)^2\}} \tag{4.4}$$

then Condition (4.2) holds. Notice that up to $H$ and $\log(\cdot)$ factors, this is precisely the sample complexity of Theorem 1 if we set $\bar\pi = \pi^\star$ and minimize over all logging/exploration policies $\mu/\pi_{\exp}$. Note that, if $\bar{V}$ denotes the average reward collected from rolling out $\bar\pi$ $K_{\bar\pi}$ times, then $|\bar{V} - V_0^{\bar\pi}| \leq \sqrt{\frac{H^2}{K_{\bar\pi}}}$ by Hoeffding's inequality. Thus, one could use $\widehat{V}^\pi = \widehat{D}^\pi + \bar{V}$ as an effective off-policy estimator. Likewise, $\widehat{D}^\pi - \widehat{D}^{\pi'}$ is an effective estimator for $V_0^\pi - V_0^{\pi'}$.

This calculation (elaborated on in Appendix A) suggests that our analysis is tight, and clearly illustrates that the $U(\pi, \bar\pi)$ term arises due to estimating the behavior of the reference policy $w_h^{\bar\pi}$. The $U(\pi, \bar\pi)$ term is, to the best of our knowledge, novel in the literature. More precisely, this term corresponds to the cost of estimating where $\bar\pi$ visits, if our goal is to estimate the difference in value between policy $\pi$ and $\bar\pi$. If, for a given state, the actions taken by $\pi$ and $\bar\pi$ achieve the same long-term reward, then it is not critical that the frequency with which $\bar\pi$ visits this state is estimated, as it does not affect the difference in values between $\pi$ and $\bar\pi$; if the actions take by $\pi$ and $\bar\pi$ do achieve different long-term reward at $s$, then we must estimate the behavior of each policy at this state. This is reflected by the term inside the expectation of $U(\pi, \bar\pi)$; this will be 0 in the former case, and scale with the difference between long-term action reward in the latter case.

Additionally, note that if we had offline data from *some* policy $\bar\pi$, that had been played for a long time, so that $K_{\bar\pi} \approx \infty$, then we would only incur the $K_\mu$ term; this is precisely $\rho_\Pi$, but with $\pi^\star$ replaced with our reference policy $\bar\pi$ in the numerator.

## 5 Achieving Theorem 1: PERP Algorithm

While the above section provides intuition for where the terms in Theorem 1 come from, it does not lead to a practical algorithm. This is because the desired number of samples in Equation (4.4) are in terms of unknown quantities: $\{\|\phi_h^\pi - \phi_h^{\bar\pi}\|_{\Lambda_h(\mu)^{-1}}^2, \Delta(\pi), U(\pi, \bar\pi)\}$, which depend on our unknown environment variables $\nu_h, P_h$; hence, we would not know how many samples to collect. In this section, we propose an algorithm that will proceed in rounds, successively improving our estimates of these quantities. Define

$$\widehat{U}_{\ell,h}(\pi, \pi') := \widehat{\mathbb{E}}_{\pi',\ell}[(\widehat{Q}_{\ell,h}^\pi(s_h, \pi_h(s)) - \widehat{Q}_{\ell,h}^\pi(s_h, \pi_h'(s)))^2], \tag{5.1}$$

where $\widehat{\mathbb{E}}_{\pi',\ell}$ denotes the expectation induced playing policy $\pi'$ on the MDP with transitions $\widehat{P}_{\ell,h}$, and $\widehat{Q}_{\ell,h}^\pi$ denotes the $Q$-function of policy $\pi$ on this same MDP. To compute $\widehat{P}_{\ell,h}$, we use the standard estimator: $\widehat{P}_{\ell,h}(s' \mid s,a) = \frac{N_{\ell,h}(s,a,s')}{N_{\ell,h}(s,a)}$ for $N_{\ell,h}(s,a)$ and $N_{\ell,h}(s,a,s')$ the visitation counts in $\mathfrak{D}_{\ell,h}^{\mathrm{ED}}$. We set $\widehat{P}_{\ell,h}(s' \mid s,a) = \mathrm{unif}(\mathcal{S})$ if $N_{\ell,h}(s,a) = 0$. The analogous estimator is used to estimate $\widehat{r}_{\ell,h}$. The quantity $\phi_h^\pi - \phi_h^{\bar\pi}$ is estimated as in the previous section: $(\boldsymbol{\pi}_h - \bar{\boldsymbol{\pi}}_{\ell,h})\widehat{w}_{\ell,h}^{\bar\pi} + \boldsymbol{\pi}_h \widehat{\delta}_{\ell,h}^\pi$.

Algorithm 1 proceeds in epochs. It begins with a policy set $\Pi_1$, which contains all policies of interest, $\Pi$. It then gradually begins to refine this policy set, seeking to estimate the *difference* in values between policies in the set up to tolerance $\epsilon_\ell = 2^{-\ell}$. To achieve this, it instantiates the intuition above. First, it chooses a reference policy $\bar\pi_\ell$, then running this estimate a sufficient number of times to estimate $w_h^{\bar\pi_\ell}$. Given this estimate, it then seeks to estimate $\delta_h^\pi$ for each $\pi$ in the active set of policies, $\Pi_\ell$, by collecting data covering the directions $(\boldsymbol{\pi}_h - \bar{\boldsymbol{\pi}}_{\ell,h})\widehat{w}_{\ell,h}^{\bar\pi} + \boldsymbol{\pi}_h \widehat{\delta}_{\ell,h}^\pi$ for all $\pi \in \Pi_\ell$. To efficiently collect this covering data, on line 12, we run a data collection procedure first developed in [42]. Finally, after estimating each $\delta_h^\pi$, it estimates the differences between policy values as in (4.3), and eliminates suboptimal policies.

The computational complexity of PERP is poly $(S, A, H, 1/\epsilon, |\Pi|, \log(1/\delta))$. The primary contributor to the computational complexity is the the use of the Franke-Wolfe algorithm for experiment design in the OPTCOV subroutine. Lemma 37 from Wagenmaker and Pacchiano [43] shows that the number of iterations of the Franke-Wolfe algorithm is bounded polynomially in the problem parameters,

---
**Algorithm 1** PERP: Policy Elimination with Reference Policy (informal)
---
**Require:** tolerance $\epsilon$, confidence $\delta$, policies $\Pi$
1: $\Pi_1 \leftarrow \Pi$, $\widehat{P}_0 \leftarrow$ arbitrary transition matrix
2: **for** $\ell = 1, 2, 3, \ldots, \lceil \log_2 \frac{16}{\epsilon} \rceil$ **do**
3:     Set $\epsilon_\ell \leftarrow 2^{-\ell}$
4:     `// Compute new reference policy`
5:     Compute $\widehat{U}_{\ell-1,h}(\pi, \pi')$ as in (5.1) for all $(\pi, \pi') \in \Pi_\ell$
6:     Choose $\bar{\pi}_\ell \leftarrow \min_{\bar{\pi} \in \Pi_\ell} \max_{\pi \in \Pi_\ell} \sum_{h=1}^{H} \widehat{U}_{\ell-1,h}(\pi, \bar{\pi})$
7:     Collect the following number of episodes from $\bar{\pi}_\ell$ and store in dataset $\mathfrak{D}_\ell^{\mathrm{ref}}$

$$\bar{n}_\ell = \mathcal{O}\Big( \max_{\pi \in \Pi_\ell} c \cdot \frac{H \widehat{U}_{\ell-1}(\pi, \bar{\pi}_\ell)}{\epsilon_\ell^2} \cdot \log \frac{H\ell^2 |\Pi_\ell|}{\delta} \Big)$$

8:     Compute $\{\widehat{w}_{\ell,h}^{\bar{\pi}}(s)\}_{h=1}^{H}$ using empirical state visitation frequencies in $\mathfrak{D}_\ell^{\mathrm{ref}}$
9:     `// Estimate Policy Differences`
10:    Initialize $\widehat{\delta}_1^{\pi} \leftarrow 0$
11:    **for** $h = 1, \ldots, H$ **do**
12:       Run OPTCOV (Algorithm 3) to collect dataset $\mathfrak{D}_{\ell,h}^{\mathrm{ED}}$ such that:

$$\sup_{\pi \in \Pi_\ell} \|(\boldsymbol{\pi}_h - \bar{\boldsymbol{\pi}}_{\ell,h})\widehat{w}_{\ell,h}^{\bar{\pi}} + \boldsymbol{\pi}_h \widehat{\delta}_{\ell,h}^{\pi}\|_{\Lambda_{\ell,h}^{-1}}^2 \leq \epsilon_\ell^2 / H^4 \beta_\ell^2 \quad \text{for} \quad \Lambda_{\ell,h} = \sum_{(s,a) \in \mathfrak{D}_{\ell,h}^{\mathrm{ED}}} e_{sa} e_{sa}^{\top}$$

        and $\beta_\ell \leftarrow \mathcal{O}(\sqrt{\log SH\ell^2 |\Pi_\ell|/\delta})$
13:       Use $\mathfrak{D}_{\ell,h}^{\mathrm{ED}}$ to compute $\widehat{P}_{\ell,h}(s'|s,a)$ and $\widehat{r}_{\ell,h}$
14:       Compute $\widehat{\delta}_{\ell,h+1}^{\pi} \leftarrow \widehat{P}_{\ell,h}(\boldsymbol{\pi}_h - \bar{\boldsymbol{\pi}}_{\ell,h})\widehat{w}_{\ell,h}^{\bar{\pi}} + \widehat{P}_{\ell,h}\boldsymbol{\pi}_h \widehat{\delta}_{\ell,h}^{\pi})$
15:    **end for**
16:    `// Eliminate suboptimal policies`
17:    Compute $\widehat{D}_{\bar{\pi}_\ell}(\pi) \leftarrow \sum_h \langle \widehat{r}_{\ell,h}, \boldsymbol{\pi}_h \widehat{\delta}_{\ell,h} \rangle + \sum_h \langle \widehat{r}_{\ell,h}, (\boldsymbol{\pi}_h - \bar{\boldsymbol{\pi}}_{\ell,h})\widehat{w}_{\ell,h}^{\bar{\pi}} \rangle$
18:    Update $\Pi_{\ell+1} = \Pi_\ell \backslash \{\pi \in \Pi_\ell : \max_{\pi'} \widehat{D}_{\bar{\pi}_\ell}(\pi') - \widehat{D}_{\bar{\pi}_\ell}(\pi) > 8\epsilon_\ell \}$
19:    **if** $|\Pi_{\ell+1}| = 1$ **then return** $\pi \in \Pi_{\ell+1}$
20: **end for**
21: **return** any $\pi \in \Pi_{\ell+1}$
---

and from the definition of this procedure given in Wagenmaker and Pacchiano [43], we see that each iteration of Franke-Wolfe has computational complexity polynomial in problem parameters. We omit several technical details from Algorithm 1 for simplicity, but present the full definition in Algorithm 2.

## 6   When is $\rho_\Pi$ Sufficient?

Our results so far show that $\rho_\Pi$ is not in general sufficient for tabular RL. In this section, we consider several special cases where it *is* sufficient.

**Tabular Contextual Bandits.** The tabular contextual bandit setting is the special case of the RL setting with $H = 1$ and where the initial action does not affect the next-state transition. Theorem 2.2 of Li et al. [30] show that if the rewards distributions $\nu(s,a)$ are Gaussian for each $(s,a)$, where here $s$ denotes the context, any $(0, \delta)$-PAC algorithm requires at least $\rho_\Pi$ samples. Crucially, however, they assume that the context distribution—in this case corresponding to the initial transition $P_1$—is known. Their algorithm makes explicit use of this fact, using this to estimate the value of $\phi^\pi$. The following result shows that knowing the context distribution is not critical—we can achieve a complexity of $\mathcal{O}(\rho_\Pi)$ without this prior knowledge.

**Corollary 1.** *For the setting of tabular contextual bandits, there exists an algorithm such that with probability at least $1 - 2\delta$, as long as $\Pi$ contains only deterministic policies, it finds an $\epsilon$-optimal*

*policy and terminates after collecting at most the following number of samples:*

$$\inf_{\pi_{\exp}} \max_{\pi \in \Pi} \frac{\|\phi^\star - \phi^\pi\|^2_{\Lambda(\pi_{\exp})^{-1}}}{\max\{\epsilon^2, \Delta(\pi)^2\}} \cdot \beta^2 \log \frac{1}{\Delta_{\min} \vee \epsilon} + \frac{C_{\text{poly}}}{\max\{\epsilon^{5/3}, \Delta_{\min}^{5/3}\}},$$

*for $C_{\text{poly}} = \text{poly}(|\mathcal{S}|, A, \log 1/\delta, \log 1/(\Delta_{\min} \vee \epsilon), \log |\Pi|)$ and $\beta = C\sqrt{\log(\frac{S|\Pi|}{\delta} \cdot \frac{1}{\Delta_{\min} \vee \epsilon})}$.*

The theorem is proved in Appendix D, and follows from the application of our algorithm PERP to the contextual bandit problem. The key intuition behind this result is that, in the contextual case:

$$U(\pi, \bar{\pi}) = \mathbb{E}_{s \sim P_1}[(r_1(s, \pi_1(s)) - r_1(s, \bar{\pi}_1(s)))^2] \leq \mathbb{E}_{s \sim P_1}[\mathbb{I}\{\pi_1(s) \neq \bar{\pi}_1(s)\}].$$

It is then possible to show that, since $\pi_{\exp}$ only has choices of which actions are taken (and cannot affect the context distribution), this can be further bounded by $\inf_{\pi_{\exp}} \|\phi^\pi - \phi^{\bar{\pi}}\|^2_{\Lambda(\pi_{\exp})^{-1}}$. This is not true in the full MDP case, where our choice of exploration policy in $\pi_{\exp}$ could make $\inf_{\pi_{\exp}} \|\phi^\pi - \phi^{\bar{\pi}}\|^2_{\Lambda(\pi_{\exp})^{-1}}$ significantly smaller than $U(\pi, \bar{\pi})$ (as is the case in Lemma 2). Hence, we observe that the cost of learning the contexts is dominated by that of learning the rewards in the case of contextual bandits. This is the opposite of tabular RL, where our complexity from Theorem 1 is unchanged (as seen in Section 4.2) even if we knew the reward distribution. This shows that there is a distinct separation between instance-optimal learning in tabular RL vs contextual bandits.

**MDPs with Action-Independent Transitions.** In the special case of MDPs where the transitions do not depend on the actions selected, the complexity simplifies to $\mathcal{O}(\rho_\Pi)$. Note that this exactly matches (up to lower order terms) the lower bound from [2].

**Corollary 2.** *Assume that all $P_h$ are such that $P_h(s'|s, a) = P_h(s'|s, a')$ for all $(a, a') \in \mathcal{A}$. Then, with probability at least $1 - 2\delta$, PERP (Algorithm 2) finds an $\epsilon$-optimal policy and terminates after collecting at most the following number of episodes:*

$$\sum_{h=1}^{H} \inf_{\pi_{\exp}} \max_{\pi \in \Pi} \frac{\|\phi_h^\star - \phi_h^\pi\|^2_{\Lambda_h(\pi_{\exp})^{-1}}}{\max\{\epsilon^2, \Delta(\pi)^2\}} \cdot \iota H^4 \beta^2 + \frac{C_{\text{poly}}}{\max\{\epsilon^{5/3}, \Delta_{\min}^{5/3}\}}$$

*for $C_{\text{poly}}, \beta$ as defined in Theorem 1.*

The intuition for Corollary 2 is similar to that of Corollary 1, and proved in Appendix E.

## 7  Discussion

In this paper, we performed a fine-grained study of the instance-dependent complexity of tabular RL. We proposed a new off-policy estimator that estimates the value relative to a reference policy. We leveraged this insight to close the instance-dependent contextual bandits problem and obtained the tightest known upper bound for tabular MDPs.

**Limitations and Future work** One limitation of the present work is that PERP, in it's current form, would be too computationally expensive to run for most practical applications; enumerating the policy set $\Pi$ is often intractable, but works in contextual bandits have avoided this issue by only relying on argmax oracles over this set [1, 30]; an interesting direction of future work would be to extend this technique to tabular RL. Extending the results from this paper to obtain refined instance-dependent bounds for linear MDPs and general function approximation is an exciting direction as well.

The new estimator and its improved sample complexity raise additional theoretical questions. Our upper bound has unfortunate low order terms; can these be removed? Can one show that $\frac{U(\pi, \bar{\pi})}{\max\{\Delta(\pi)^2, \epsilon^2\}}$ is unavoidable for all MDPs in general, thereby matching our upper bound? As discussed above, a few works have proven gap-dependent regret upper bounds, but we are unaware of any matching lower bounds besides over restricted classes of MDPs; can our estimator involving the differences result in even tighter instance-dependent regret bounds for MDPs?

## Acknowledgments

AN and LJR are supported in part by ONR YIP N000142012571 and NSF CAREER 1844729. AN was supported, in part by the Amazon Hub Fellowship at the University of Washington. KJ and AW were funded in part by NSF CAREER 2141511 and NSF TRIPODS 2023239.

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

# Contents

| Notation | Description |
|---|---|
| $\mathcal{S}$ | State space |
| $\mathcal{A}$ | Action space |
| $H$ | Horizon |
| $P_h$ | Transition matrix at stage $h$ |
| $\nu_h$ | Distribution over reward at stage $h$ |
| $r_h(s,a)$ | Expected reward at stage $h$ for state $s$ and action $a$ |
| $\pi$ | Policy |
| $\Pi$ | Set of candidate policies |
| $\pi_h(s)$ | Distribution over actions for policy $\pi$ at state $s$ and stage $h$ |
| $w_h^\pi$ | State visitation vector at step $h$ for policy $\pi$ |
| $\boldsymbol{\pi}_h$ | Policy matrix for policy $\pi$ at step $h$ |
| $\phi_h^\pi$ | State-action visitation vector for policy $\pi$ at step $h$ |
| $\Lambda_h(\pi)$ | Expected covariance matrix at timestep $h$ for policy $\pi$ |
| $Q_h^\pi(s,a)$ | Q-value function for policy $\pi$ at state $s$, action $a$, and step $h$ |
| $V_h^\pi(s)$ | Value function for policy $\pi$ at state $s$ and step $h$ |
| $V^\pi$ | Value of policy $\pi$ |
| $\pi^\star$ | Optimal policy within $\Pi$ |
| $\Delta(\pi)$ | Suboptimality of policy $\pi$ |
| $W_h^\star(s)$ | Maximum probability of reaching state $s$ at step $h$ over all policies |
| $\mathcal{C}$ | Context space (for contextual bandits) |
| $\mu^\star$ | Context distribution (for contextual bandits) |
| $\theta^\star$ | Reward parameters (for contextual bandits) |
| $\rho_\Pi$ | Complexity measure based on feature differences |
| $\bar{\pi}_\ell$ | Reference policy |
| $\delta_h^\pi$ | Difference in state visitation between policy $\pi$ and reference policy at step $h$ |
| $D_{\bar{\pi}_\ell}(\pi)$ | Difference in value between policy $\pi$ and reference policy |
| $U_h(\pi,\pi')$ | Expected squared difference in Q-values between policies $\pi$ and $\pi'$ at step $h$ |
| $\mathcal{S}_\ell^{\mathrm{keep}}$ | Set of reachable states at epoch $\ell$ |
| $\epsilon_{\mathrm{unif}}^\ell$ | Minimum reachability threshold at epoch $\ell$ |
| $\epsilon_{\mathrm{exp}}^\ell$ | Tolerance for experiment design at epoch $\ell$ |
| $\beta_\ell$ | Confidence parameter at epoch $\ell$ |
| $n_\ell, K_{\mathrm{unif}}^\ell$ | Number of samples and minimum exploration at epoch $\ell$ |
| $\mathfrak{D}_{\ell,h}^{\mathrm{ED}}$ | Dataset collected during exploration in PERP |
| $\mathfrak{D}_\ell^{\mathrm{ref}}$ | Dataset collected from reference policy |

Table 1: Table of notation used in the paper

# A  Understanding the origins of $U(\pi, \bar{\pi})$

This section is inspired by the exposition of Soare et al. [37] for justifying the sample complexity of linear bandits. Fix a reference policy $\bar{\pi}$ and some (stochastic) logging policy $\mu$. For $K \in \mathbb{N}$ to be determined later, roll out $\bar{\pi}$ $K$ times and compute the empirical state visitations $\widehat{w}_h^{\bar{\pi}}(s) = \frac{1}{K}\sum_{k=1}^K \sum_{s,h} \mathbf{1}\{s_h^k = s\}$. Also roll out $\mu$ $K$ times and compute the empirical transition probabilities $\widehat{P}_h(s'|s,a) = \frac{\sum_{k=1}^K \mathbf{1}\{(s_h^k, a_h^k, s_{h+1}^k) = (s,a,s')\}}{\sum_{k=1}^K \mathbf{1}\{(s_h^k, a_h^k) = (s,a)\}}$. For any $\pi \neq \bar{\pi}$, use $\{\widehat{P}_h(s'|s,a)\}_{s,a,s',h}$ to compute $\widehat{w}_h^\pi(s)$. With $\delta_{h+1}^\pi := w_{h+1}^\pi - w_{h+1}^{\bar{\pi}} = P_h\boldsymbol{\pi}_h w_h^\pi - P_h\bar{\boldsymbol{\pi}}_h w_h^{\bar{\pi}} = P_h\boldsymbol{\pi}_h\delta_h^\pi + P_h(\boldsymbol{\pi}_h - \bar{\boldsymbol{\pi}}_h)w_h^{\bar{\pi}}$ set

$$D(\pi) = V_0^\pi - V_0^{\bar{\pi}} = \sum_{h=1}^H \langle r_h, \boldsymbol{\pi}_h w_h^\pi - \bar{\boldsymbol{\pi}}_h w_h^{\bar{\pi}}\rangle = \sum_{h=1}^H \langle r_h, \boldsymbol{\pi}_h\delta_h^\pi\rangle + \langle r_h, (\boldsymbol{\pi}_h - \bar{\boldsymbol{\pi}}_h)w_h^{\bar{\pi}}\rangle$$

and also define the empirical counterparts $\widehat{\delta}_{h+1}^\pi := \widehat{P}_h\boldsymbol{\pi}_h\widehat{\delta}_h^\pi + \widehat{P}_h(\boldsymbol{\pi}_h - \bar{\boldsymbol{\pi}}_h)\widehat{w}_h^{\bar{\pi}}$ with

$$\widehat{D}(\pi) = \sum_{h=1}^H \langle r_h, \boldsymbol{\pi}_h\widehat{\delta}_h^\pi\rangle + \langle r_h, (\boldsymbol{\pi}_h - \bar{\boldsymbol{\pi}}_h)\widehat{w}_h^{\bar{\pi}}\rangle.$$

If $\widehat{\pi} = \arg\max_{\pi \in \Pi} \widehat{D}(\pi)$, how large must $K$ be to ensure that $\widehat{\pi} = \pi^\star := \arg\max_{\pi \in \Pi} D(\pi) = \arg\max_{\pi \in \Pi} V_0^\pi$?

Assume at time $h = 0$ all policies are initialized arbitrarily in some state $s_0$ so that $\widehat{P}_0(s'|s_0, a)$ simply defines the initial empirical state distribution at time $h = 1$. Let $\widehat{w}_0^\pi(s_0) = w_0^\pi(s_0) = 1$ We can then unroll the recursion for $h = 0, \ldots, H-1$

$$\widehat{\delta}_{h+1}^\pi - \delta_{h+1}^\pi = \widehat{P}_h \boldsymbol{\pi}_h \widehat{\delta}_h^\pi + \widehat{P}_h(\boldsymbol{\pi}_h - \bar{\boldsymbol{\pi}}_h)\widehat{w}_h^{\bar{\pi}} - \delta_{h+1}^\pi$$

$$= (\widehat{P}_h - P_h)\boldsymbol{\pi}_h \delta_h^\pi + (\widehat{P}_h - P_h)(\boldsymbol{\pi}_h - \bar{\boldsymbol{\pi}}_h)w_h^{\bar{\pi}} + P_h(\boldsymbol{\pi}_h - \bar{\boldsymbol{\pi}}_h)(\widehat{w}_h^{\bar{\pi}} - w_h^{\bar{\pi}}) + P_h \boldsymbol{\pi}_h(\widehat{\delta}_h^\pi - \delta_h^\pi)$$

$$+ \underbrace{(\widehat{P}_h - P_h)\boldsymbol{\pi}_h(\widehat{\delta}_h^\pi - \delta_h^\pi) + (\widehat{P}_h - P_h)(\boldsymbol{\pi}_h - \bar{\boldsymbol{\pi}}_h)(\widehat{w}_h^{\bar{\pi}} - w_h^{\bar{\pi}})}_{\text{Low order terms} \approx 0}$$

$$\approx (\widehat{P}_h - P_h)(\phi_k^\pi - \phi_k^{\bar{\pi}}) + P_h(\boldsymbol{\pi}_h - \bar{\boldsymbol{\pi}}_h)(\widehat{w}_h^{\bar{\pi}} - w_h^{\bar{\pi}}) + P_h \boldsymbol{\pi}_h(\widehat{\delta}_h^\pi - \delta_h^\pi)$$

$$\approx \sum_{i=0}^{h} \Big( \prod_{j=h-i+1}^{h} P_j \boldsymbol{\pi}_j \Big)\Big((\widehat{P}_{h-i} - P_{h-i})(\phi_{h-i}^\pi - \phi_{h-i}^{\bar{\pi}}) + P_{h-i}(\boldsymbol{\pi}_{h-i} - \bar{\boldsymbol{\pi}}_{h-i})(\widehat{w}_{h-i}^{\bar{\pi}} - w_{h-i}^{\bar{\pi}})\Big)$$

$$= \sum_{k=0}^{h} \Big( \prod_{j=k+1}^{h} P_j \boldsymbol{\pi}_j \Big)\Big((\widehat{P}_k - P_k)(\phi_k^\pi - \phi_k^{\bar{\pi}}) + P_k(\boldsymbol{\pi}_k - \bar{\boldsymbol{\pi}}_k)(\widehat{w}_k^{\bar{\pi}} - w_k^{\bar{\pi}})\Big)$$

where we recall $\phi_k^\pi = \boldsymbol{\pi}_k w_k^\pi$. If $\epsilon_{k+1} := (\widehat{P}_k - P_k)(\boldsymbol{\pi}_h w_k^\pi - \bar{\boldsymbol{\pi}} w_k^{\bar{\pi}}) + P_k(\boldsymbol{\pi}_k - \bar{\boldsymbol{\pi}}_k)(\widehat{w}_k^{\bar{\pi}} - w_k^{\bar{\pi}})$ then

$$\sum_{h=1}^{H} \langle r_h, \boldsymbol{\pi}_h(\widehat{\delta}_h^\pi - \delta_h^\pi)\rangle = \sum_{h=1}^{H}\sum_{k=0}^{h-1} \langle r_h, \boldsymbol{\pi}_h\Big( \prod_{j=k+1}^{h-1} P_j \boldsymbol{\pi}_j \Big)\epsilon_{k+1}\rangle$$

$$= \sum_{k=0}^{H-1}\sum_{h=k+1}^{H} \langle r_h, \boldsymbol{\pi}_h\Big( \prod_{j=k+1}^{h-1} P_j \boldsymbol{\pi}_j \Big)\epsilon_{k+1}\rangle = \sum_{k=0}^{H-1} \langle V_{k+1}^\pi, \epsilon_{k+1}\rangle$$

$$= \sum_{k=0}^{H-1} \langle V_{k+1}^\pi, (\widehat{P}_k - P_k)(\phi_k^\pi - \phi_k^{\bar{\pi}}) + P_k(\boldsymbol{\pi}_k - \bar{\boldsymbol{\pi}}_k)(\widehat{w}_k^{\bar{\pi}} - w_k^{\bar{\pi}})\rangle.$$

Finally, we use these calculations to compute the deviation

$$\widehat{D}(\pi) - D(\pi) = \sum_{h=1}^{H} \langle r_h, \boldsymbol{\pi}_h(\widehat{\delta}_h^\pi - \delta_h^\pi)\rangle + \langle r_h, (\boldsymbol{\pi}_h - \bar{\boldsymbol{\pi}}_h)(\widehat{w}_h^{\bar{\pi}} - w_h^{\bar{\pi}})\rangle$$

$$= \sum_{h=0}^{H-1} \langle V_{h+1}^\pi, (\widehat{P}_h - P_h)(\phi_h^\pi - \phi_h^{\bar{\pi}})\rangle + \langle r_h + P_h^\top V_{h+1}^\pi, (\boldsymbol{\pi}_h - \bar{\boldsymbol{\pi}}_h)(\widehat{w}_h^{\bar{\pi}} - w_h^{\bar{\pi}})\rangle$$

$$= \sum_{h=0}^{H-1} \langle V_{h+1}^\pi, (\widehat{P}_h - P_h)(\phi_h^\pi - \phi_h^{\bar{\pi}})\rangle + \langle Q_h^\pi, (\boldsymbol{\pi}_h - \bar{\boldsymbol{\pi}}_h)(\widehat{w}_h^{\bar{\pi}} - w_h^{\bar{\pi}})\rangle$$

$$= \sum_{h=0}^{H-1}\sum_{s,a,s'} V_{h+1}^\pi(s')(\widehat{P}_h(s'|s,a) - P_h(s'|s,a))(\phi_h^\pi(s,a) - \phi_h^{\bar{\pi}}(s,a))$$

$$+ \sum_{h=0}^{H-1}\sum_{s} \big(Q_h^\pi(s, \pi_h(s)) - Q_h^\pi(s, \bar{\pi}_h(s))\big)(\widehat{w}_h^{\bar{\pi}}(s) - w_h^{\bar{\pi}}(s))$$

$$\lesssim \sqrt{\sum_{h=0}^{H-1}\sum_{s,a,s'} V_{h+1}^\pi(s')^2 \frac{P_h(s'|s,a)}{K\mu_h(s,a)}(\phi_h^\pi(s,a) - \phi_h^{\bar{\pi}}(s,a))^2}$$

$$+ \sqrt{\sum_{h=0}^{H-1}\sum_{s} \big(Q_h^\pi(s, \pi_h(s)) - Q_h^\pi(s, \bar{\pi}_h(s))\big)^2 \frac{w_h^{\bar{\pi}}(s)}{K}}.$$

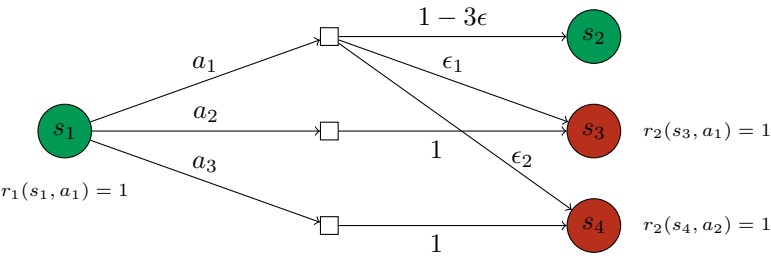

Figure 2: A motivating example for differences. All rewards other than the ones specified in the figure are 0.

Applying $\sum_{s'} V_{h+1}^\pi(s')^2 P_h(s'|s,a) \leq H^2$, we observe that if

$$K \geq \min_{\mu,\bar{\pi}} \max_\pi H^2 \sum_{h=1}^{H-1} \frac{\sum_{s,a}(\phi_h^\pi(s,a) - \phi_h^{\bar{\pi}}(s,a))^2/\mu_h(s,a)}{\Delta(\pi)^2}$$

$$+ \sum_{h=1}^{H-1} \frac{\sum_s \left(Q_h^\pi(s,\pi_h(s)) - Q_h^\pi(s,\bar{\pi}_h(s))\right)^2 w_h^{\bar{\pi}}(s)}{\Delta(\pi)^2}$$

and we employ the minimizers $\mu,\bar{\pi}$ to collect data, then $\widehat{D}(\pi) - D(\pi) < \Delta(\pi)$ and $\widehat{\pi} = \arg\max_{\pi\in\Pi} \widehat{D}(\pi) = \arg\max_{\pi\in\Pi} D(\pi)$. Notice that up to $H$ and $\log$ factors, this is precisely the sample complexity of our algorithm. A natural candidate for $\bar{\pi}$ is $\pi^\star$ so that the first term matches the lower bound of [2].

On the other hand, suppose we used the data from the logging policy $\mu$ to compute the empirical state visitations $\widehat{w}_h^\pi$ for all $\pi \in \Pi$ and set $\widehat{\pi} = \arg\max_{\pi\in\Pi} \sum_{h=1}^H \langle r_h, \boldsymbol{\pi}\widehat{w}_h^\pi\rangle =: \widehat{V}_0^\pi$. Using the same techniques as above, it is straightforward to show that if

$$\widehat{w}_{h+1}^\pi - w_{h+1}^\pi = \widehat{P}_h\boldsymbol{\pi}_h\widehat{w}_h^\pi - P_h\boldsymbol{\pi}_h w_h^\pi$$

$$= (\widehat{P}_h - P_h + P_h)\boldsymbol{\pi}_h(\widehat{w}_h^\pi - w_h^\pi + w_h^\pi) - P_h\boldsymbol{\pi}_h w_h^\pi$$

$$= (\widehat{P}_h - P_h)\boldsymbol{\pi}_h w_h^\pi + P_h\boldsymbol{\pi}_h(\widehat{w}_h^\pi - w_h^\pi) + \underbrace{(\widehat{P}_h - P_h)\boldsymbol{\pi}_h(\widehat{w}_h^\pi - w_h^\pi)}_{\text{Low order terms} \approx 0}$$

$$\approx \sum_{i=0}^h \Big(\prod_{j=h-i+1}^h P_j\boldsymbol{\pi}_j\Big)(\widehat{P}_{h-i} - P_{h-i})\boldsymbol{\pi}_{h-i}w_{h-i}^\pi$$

$$= \sum_{k=0}^h \Big(\prod_{j=k+1}^h P_j\boldsymbol{\pi}_j\Big)(\widehat{P}_k - P_k)\boldsymbol{\pi}_k w_k^\pi$$

and we employ the minimizer $\mu$ to collect data, then $\widehat{V}_0^\pi - V_0^\pi \leq \Delta(\pi)$ and $\widehat{\pi} = \arg\max_{\pi\in\Pi} \widehat{V}_0^\pi = \arg\max_{\pi\in\Pi} V_0^\pi$.

# B   Tabular MDPs: Comparison with Prior Work and Lower Bounds

**Illustrative Family of MDP Instances**   Recall the family of MDP instances in the introduction (visualized in Figure 2 for ease of reference). The family of MDPs is parameterized by $\epsilon, \epsilon_1, \epsilon_2 > 0$, with $H = 2$, $\mathcal{S} = \{s_1, s_2, s_3, s_4\}$, and $\mathcal{A} = \{a_1, a_2, a_3\}$, which start in state $s_0$ and are defined as:

$$P_1(s_2 \mid s_1, a_1) = 1 - 3\epsilon, \quad P_1(s_3 \mid s_1, a_1) = \epsilon_1, \quad P_1(s_4 \mid s_1, a_1) = \epsilon_2$$
$$P_1(s_3 \mid s_1, a_2) = P_1(s_4 \mid s_1, a_3) = 1.$$

We define the reward function so that all rewards are 0 except $r_1(s_1, a_1) = r_2(s_3, a_1) = r_2(s_4, a_2) = 1$ for all $a$.

Let $\mathcal{M}$ denote the MDP above with $\epsilon_1 = 2\epsilon, \epsilon_2 = \epsilon$, and $\mathcal{M}'$ the MDP above with $\epsilon_1 = \epsilon, \epsilon_2 = 2\epsilon$.

Let $\Pi = \{\pi_1, \pi_2\}$ denote some set of policies. Let $\pi_1$ denote the policy which always plays $a_1$, and $\pi_2$ the policy which plays $a_1$ at green states and $a_2$ at red states i.e $\pi_2(s_1) = \pi_2(s_2) = a_1$ and $\pi_2(s_3) = \pi_2(s_4) = a_2$.

Now note that $V_0^{\mathcal{M},\pi_1} = 1 + 2\epsilon$, $V_0^{\mathcal{M},\pi_2} = 1 + \epsilon$, $V_0^{\mathcal{M}',\pi_1} = 1 + \epsilon$, and $V_0^{\mathcal{M}',\pi_2} = 1 + 2\epsilon$.

## B.1 Comparison with complexities from prior work

The lemma below shows that the upper bound presented in Theorem 1 is smaller than that of PEDEL from Theorem 1 of [42] for all MDP instances.

**Lemma 4.** *For any MDP instance and policy set* $\Pi$*, we have that*

1. $\inf_{\pi_{\exp}} \max_{\pi \in \Pi} \dfrac{\|\phi_h^\pi\|_{\Lambda_h(\pi_{\exp})^{-1}}^2}{\max\{\epsilon^2, \Delta(\pi)^2, \Delta_{\min}^2\}} \geq \dfrac{1}{\max\{\epsilon^2, \Delta(\pi)^2, \Delta_{\min}^2\}}$

2.

$$H^4 \sum_{h=1}^{H} \inf_{\pi_{\exp}} \max_{\pi \in \Pi} \frac{\|\phi_h^\star - \phi_h^\pi\|_{\Lambda_h(\pi_{\exp})^{-1}}^2}{\max\{\epsilon^2, \Delta(\pi)^2, \Delta_{\min}^2\}} \leq 4H^4 \sum_{h=1}^{H} \inf_{\pi_{\exp}} \max_{\pi \in \Pi} \frac{\|\phi_h^\pi\|_{\Lambda_h(\pi_{\exp})^{-1}}^2}{\max\{\epsilon^2, \Delta(\pi)^2, \Delta_{\min}^2\}}$$

3. $\dfrac{HU(\pi,\pi^\star)}{\max\{\epsilon^2, \Delta(\pi)^2, \Delta_{\min}^2\}} \leq H^4 \sum_{h=1}^{H} \inf_{\pi_{\exp}} \max_{\pi \in \Pi} \dfrac{\|\phi_h^\pi\|_{\Lambda_h(\pi_{\exp})^{-1}}^2}{\max\{\epsilon^2, \Delta(\pi)^2, \Delta_{\min}^2\}}$

*Proof.* **Proof of Claim 1.** Note that

$$\|\phi_h^\pi\|_{\Lambda_h(\pi_{\exp})^{-1}}^2 = \sum_{s,a} \frac{\phi_h^\pi(s,a)^2}{\phi_h^{\pi_{\exp}}(s,a)} \geq \inf_{\lambda \in \Delta_{SA}} \sum_{s,a} \frac{\phi_h^\pi(s,a)^2}{\lambda_{s,a}}$$

In order to solve this optimization problem, we can consider the KKT conditions. We can verify from stationarity that at optimality, $\lambda_{s,a} = \frac{\phi_h^\pi(s,a)}{\sqrt{\beta}}$ for some constant $\beta > 0$. But since $\lambda_{s,a}$ must live in the simplex $\Delta_{SA}$, and since $\phi_h^\pi(s,a)$ is itself a distribution over $\mathcal{S} \times \mathcal{A}$, it follows that $\beta = 1$ must be true. Plugging this optimal value into the above, we obtain that

$$\|\phi_h^\pi\|_{\Lambda_h(\pi_{\exp})^{-1}}^2 \geq \inf_{\lambda \in \Delta_{SA}} \sum_{s,a} \frac{\phi_h^\pi(s,a)^2}{\lambda_{s,a}} = 1$$

Then,

$$\inf_{\pi_{\exp}} \max_{\pi \in \Pi} \frac{\|\phi_h^\pi\|_{\Lambda_h(\pi_{\exp})^{-1}}^2}{\max\{\epsilon^2, \Delta(\pi)^2, \Delta_{\min}^2\}} \geq \frac{1}{\max\{\epsilon^2, \Delta(\pi)^2, \Delta_{\min}^2\}}$$

directly follows from the above.

**Proof of Claim 2.** From the triangle inequality,

$$\inf_{\pi_{\exp}} \max_{\pi \in \Pi} \frac{\|\phi_h^\star - \phi_h^\pi\|_{\Lambda_h(\pi_{\exp})^{-1}}^2}{\max\{\epsilon^2, \Delta(\pi)^2, \Delta_{\min}^2\}}$$

$$\leq 2 \inf_{\pi_{\exp}} \max_{\pi \in \Pi} \left( \frac{\|\phi_h^\star\|_{\Lambda_h(\pi_{\exp})^{-1}}^2}{\max\{\epsilon^2, \Delta(\pi)^2, \Delta_{\min}^2\}} + \frac{\|\phi_h^\pi\|_{\Lambda_h(\pi_{\exp})^{-1}}^2}{\max\{\epsilon^2, \Delta(\pi)^2, \Delta_{\min}^2\}} \right)$$

$$\leq 2 \inf_{\pi_{\exp}} \max_{\pi \in \Pi} \left( \frac{\|\phi_h^\star\|_{\Lambda_h(\pi_{\exp})^{-1}}^2}{\max\{\epsilon^2, \Delta(\pi^\star)^2, \Delta_{\min}^2\}} + \frac{\|\phi_h^\pi\|_{\Lambda_h(\pi_{\exp})^{-1}}^2}{\max\{\epsilon^2, \Delta(\pi)^2, \Delta_{\min}^2\}} \right)$$

$$\leq 4 \inf_{\pi_{\exp}} \max_{\pi \in \Pi} \frac{\|\phi_h^\pi\|_{\Lambda_h(\pi_{\exp})^{-1}}^2}{\max\{\epsilon^2, \Delta(\pi)^2, \Delta_{\min}^2\}}$$

where we have used that $\Delta(\pi) \geq \Delta(\pi^\star)$ for all $\pi$. Plugging this bound into the expression from (2) from the Lemma statement completes the proof.

**Proof of Claim 3.** We have that

$$HU(\pi, \pi^\star) = H \sum_{h=1}^{H} \mathbb{E}_{s_h \sim w_h^{\pi^\star}} [(Q_h^\pi(s_h, \pi_h(s)) - Q_h^\pi(s_h, \pi_h^\star(s)))^2] \leq H \sum_{h=1}^{H} H^2 \leq H^4$$

Then,

$$\frac{HU(\pi, \pi^\star)}{\max\{\epsilon^2, \Delta(\pi)^2, \Delta_{\min}^2\}} \leq \frac{H^4}{\max\{\epsilon^2, \Delta(\pi)^2, \Delta_{\min}^2\}}$$

$$\leq H^4 \sum_{h=1}^{H} \inf_{\pi_{\exp}} \max_{\pi \in \Pi} \frac{\|\phi_h^\pi\|_{\Lambda_h(\pi_{\exp})^{-1}}^2}{\max\{\epsilon^2, \Delta(\pi)^2, \Delta_{\min}^2\}}$$

Where the final inequality follows from Claim 1 above. $\square$

The lemma below shows that there are some instances where the complexity from Theorem 1 is strictly smaller in terms of $\epsilon$ dependence than that from Theorem 1 from [42] for PEDEL.

**Lemma 5.** *On MDP $\mathcal{M}$ defined above, we have:*

1. $\sum_{h=1}^{H} \inf_{\pi_{\exp}} \max_{\pi \in \Pi} \frac{\|\phi_h^\star - \phi_h^\pi\|_{\Lambda_h(\pi_{\exp})^{-1}}^2}{\max\{\epsilon^2, \Delta(\pi)^2, \Delta_{\min}^2\}} \leq 15$

2. $\max_{\pi \in \Pi} \frac{HU(\pi, \pi^\star)}{\max\{\epsilon^2, \Delta(\pi)^2, \Delta_{\min}^2\}} = \frac{3H}{\epsilon}$

3. $\sum_{h=1}^{H} \inf_{\pi_{\exp}} \max_{\pi \in \Pi} \frac{\|\phi_h^\pi\|_{\Lambda_h(\pi_{\exp})^{-1}}^2}{\max\{\epsilon^2, \Delta(\pi)^2, \Delta_{\min}^2\}} \geq \frac{H}{\epsilon^2}$

*Proof.* **Proof of 1.** In this case we have that $\pi^* = \pi_1$, and the only other $\pi$ of interest is $\pi_2$. Note that $\pi_1$ and $\pi_2$ differ only at state $s_3$ and $s_4$ at $h = 2$. Let $\pi_{\exp}$ be the policy that plays actions uniformly at random. Then, we have

$$\sum_{h=1}^{H} \inf_{\pi_{\exp}} \max_{\pi \in \Pi} \frac{\|\phi_h^\star - \phi_h^\pi\|_{\Lambda_h(\pi_{\exp})^{-1}}^2}{\max\{\epsilon^2, \Delta(\pi)^2, \Delta_{\min}^2\}} \leq \inf_{\pi_{\exp}} \frac{\|\phi_2^{\pi_1} - \phi_2^{\pi_2}\|_{\Lambda_h(\pi_{\exp})^{-1}}^2}{\epsilon^2}$$

$$= \frac{1}{\epsilon^2} \left( \frac{w_2^{\pi_1}(s_3)^2}{w_2^{\pi_{\exp}}(s_3)} + \frac{w_2^{\pi_1}(s_4)^2}{w_2^{\pi_{\exp}}(s_4)} \right)$$

$$\leq \frac{1}{\epsilon^2} \left( \frac{4\epsilon^2}{1/3} + \frac{\epsilon^2}{1/3} \right)$$

$$= 15.$$

**Proof of 2.** Note that

$$\max_{\pi \in \Pi} \frac{HU(\pi, \pi^\star)}{\max\{\epsilon^2, \Delta(\pi)^2, \Delta_{\min}^2\}} = \frac{HU(\pi_2, \pi_1)}{\epsilon^2}.$$

Then,

$$U(\pi_2, \pi_1) = \sum_{h=1}^{H} \mathbb{E}_{s \sim w_h^{\pi_1}} [(Q_h^{\pi_1}(s, \pi_{1,h}(s)) - Q_h^{\pi_1}(s, \pi_{2,h}(s)))^2]$$

$$= \mathbb{E}_{s \sim w_2^{\pi_1}} [(Q_2^{\pi_1}(s, \pi_{1,2}(s)) - Q_2^{\pi_1}(s, \pi_{2,2}(s)))^2]$$

$$= 2\epsilon + \epsilon = 3\epsilon.$$

Combining these proves the result.

**Proof of 3.** By Claim 1 in Lemma 4, the stated result then follows by recognizing that $\max\{\epsilon^2, \Delta(\pi)^2, \Delta_{\min}^2\} \leq \epsilon^2$.

$\square$

## B.2 Lower bound

**Lemma 6.** *On MDP $\mathcal{M}$ defined above, any $(\epsilon, \delta)$-PAC algorithm must collect*

$$\mathbb{E}^{\mathcal{M}}[\tau] \geq \frac{1}{\epsilon} \cdot \log \frac{1}{2.4\delta}.$$

*samples.*

*Proof.* Consider $\Pi, \mathcal{M}$, and $\mathcal{M}'$ defined above. Let $\mathcal{E}$ denote the event $\{\widehat{\pi} = \pi_1\}$. By the above observations, we have that $\pi_1$ is $\epsilon$-optimal on $\mathcal{M}$ while $\pi_2$ is not, and that $\pi_2$ is $\epsilon$-optimal on $\mathcal{M}'$ while $\pi_1$ is not. Then by the definition of an $(\epsilon, \delta)$-PAC algorithm, $\mathbb{P}^{\mathcal{M}}[\mathcal{E}] \geq 1 - \delta$ and $\mathbb{P}^{\mathcal{M}'}[\mathcal{E}] \leq \delta$.

Let $\gamma_h(s, a)$ denote the distribution of $(r_h, s_{h+1})$ given $(s, a, h)$ on $\mathcal{M}$, and $\gamma'_h(s, a)$ is the same on $\mathcal{M}'$. Then, letting $\nu_h \leftarrow \gamma_h, \nu'_h \leftarrow \gamma'_h$ and otherwise adopting the same notation as in Lemma F.1 of [46], we have from Lemma F.1 of [46] that:

$$\sum_{s,a,h} \mathbb{E}^{\mathcal{M}}[N_h^\tau(s, a)] \text{KL}(\gamma_h(s, a), \gamma'_h(s, a)) \geq \sup_{\mathcal{E}' \in \mathcal{F}_\tau} d(\mathbb{P}^{\mathcal{M}}[\mathcal{E}'], \mathbb{P}^{\mathcal{M}'}[\mathcal{E}'])$$

$$\geq d(\mathbb{P}^{\mathcal{M}}[\mathcal{E}], \mathbb{P}^{\mathcal{M}'}[\mathcal{E}])$$

$$\geq \log \frac{1}{2.4\delta}$$

where the last inequality follows from [23].

Note that $\mathcal{M}$ and $\mathcal{M}'$ differ only at $(s_1, a_1)$, so

$$\sum_{s,a,h} \mathbb{E}^{\mathcal{M}}[N_h^\tau(s, a)] \text{KL}(\gamma_h(s, a), \gamma'_h(s, a)) = \mathbb{E}^{\mathcal{M}}[N_1^\tau(s_1, a_1)] \text{KL}(\gamma_1(s_1, a_1), \gamma'_1(s_1, a_1)).$$

Furthermore, we see that

$$\text{KL}(\gamma_1(s_1, a_1), \gamma'_1(s_1, a_1)) = 2\epsilon \log \frac{2\epsilon}{\epsilon} + \epsilon \log \frac{\epsilon}{2\epsilon} \leq \epsilon.$$

So it follows that we must have

$$\mathbb{E}^{\mathcal{M}}[N_1^\tau(s_1, a_1)] \geq \frac{1}{\epsilon} \cdot \log \frac{1}{2.4\delta}.$$

Noting that $\mathbb{E}^{\mathcal{M}}[N_1^\tau(s_1, a_1)] \leq \mathbb{E}^{\mathcal{M}}[\tau]$ completes the proof. $\qquad\square$

# C   Tabular MDP Upper Bound

## C.1   Notation

**Covariance matrices.**   We use

$$\Lambda_h(\pi_{\exp}) = \mathbb{E}_{\pi_{\exp}}[e_{s_h a_h} e_{s_h a_h}^\top]$$

to denote the expected covariance matrix and $\widehat{\Lambda}_{\ell,h}$ to denote the empirical covariance matrix collected from $\mathfrak{D}_{\ell,h}^{\text{ED}}$.

**State visitations.**   Let $\delta_{\ell,h}^\pi(s') := w_h^\pi(s') - w_h^{\bar{\pi}_\ell}(s')$, for $\bar{\pi}_\ell$ the reference policy, $\delta_{\ell,h}^\pi$ the vectorization of $\delta_{\ell,h}^\pi(s')$, and $w_h^\pi(s) = \mathbb{P}_\pi[s_h = s]$ the visitation probability, and $W_h^\star(s) = \sup_\pi w_h^\pi(s)$. Then, we can recursively define

$$\delta_{\ell,h+1}^\pi = P_h(\boldsymbol{\pi}_h - \bar{\boldsymbol{\pi}}_{\ell,h}) w_{\ell,h}^{\bar{\pi}} + P_h \boldsymbol{\pi}_h \delta_{\ell,h}^\pi. \tag{C.1}$$

Similarly,

$$\widetilde{\delta}_{\ell,h+1}^\pi = M_h \left( P_h(\boldsymbol{\pi}_h - \bar{\boldsymbol{\pi}}_{\ell,h}) \widehat{w}_{\ell,h}^{\bar{\pi}} + P_h \boldsymbol{\pi}_h \widetilde{\delta}_{\ell,h}^\pi \right). \tag{C.2}$$

And

$$\widehat{\delta}_{\ell,h+1}^\pi = M_h \left( \widehat{P}_{\ell,h}(\boldsymbol{\pi}_h - \bar{\boldsymbol{\pi}}_{\ell,h}) \widehat{w}_{\ell,h}^{\bar{\pi}} + \widehat{P}_{\ell,h} \boldsymbol{\pi}_h \widehat{\delta}_{\ell,h}^\pi \right). \tag{C.3}$$

---

**Algorithm 2** PERP: Policy Elimination with Reference Policy

---

**Require:** tolerance $\epsilon$, confidence $\delta$, policies $\Pi$

1: $\Pi_1 \leftarrow \Pi$, $\widehat{P}_0 \leftarrow$ arbitrary transition matrix
2: **for** $\ell = 1, 2, 3, \ldots, \lceil \log_2 \frac{16}{\epsilon} \rceil$ **do**
3:       Set $\epsilon_\ell \leftarrow 2^{-\ell}$, $\epsilon_{\text{unif}}^\ell \leftarrow \frac{\epsilon_\ell}{64 S^{3/2} H^2}$, $K_{\text{unif}}^\ell \leftarrow \frac{\epsilon_\ell^{-2/3}}{\epsilon_{\text{unif}}^\ell}$
4:       $\mathcal{S}_\ell^{\text{keep}} = \text{PRUNE}(\epsilon_{\text{unif}}^\ell, \delta/3\ell^2)$ (Algorithm 5)    // Prune states that are hard to reach
5:       Use $\{\widehat{P}_{\ell-1,h}\}_{h=1}^H$ to compute $\widehat{U}_{\ell-1,h}(\pi, \pi')$ for all $(\pi, \pi') \in \Pi_\ell$ // Compute new reference policy
6:       Choose $\bar{\pi}_\ell \leftarrow \min_{\bar{\pi} \in \Pi_\ell} \max_{\pi \in \Pi_\ell} \sum_{h=1}^H \widehat{U}_{\ell-1,h}(\pi, \bar{\pi})$
7:       Collect the following number of episodes from $\bar{\pi}_\ell$ and store in dataset $\mathfrak{D}_\ell^{\text{ref}}$

$$\bar{n}_\ell = \max_{\pi \in \Pi_\ell} c \cdot \frac{H\widehat{U}_{\ell-1}(\pi, \bar{\pi}_\ell) + H^4 S^{3/2} \sqrt{A} \log \frac{SAH\ell^2}{\delta} \cdot \epsilon_\ell^{1/3} + S^2 H^4 \epsilon_{\text{unif}}^\ell}{\epsilon_\ell^2} \cdot \log \frac{60 H \ell^2 |\Pi_\ell|}{\delta}$$

8:       Compute $\{\widehat{w}_{\ell,h}^{\bar{\pi}}(s)\}_{h=1}^H$ using empirical state visitation frequencies in $\mathfrak{D}_\ell^{\text{ref}}$
9:       Initialize $\widehat{\delta}_1^\pi \leftarrow 0$                                  // Exploration via experiment design
10:      **for** $h = 1, \ldots, H$ **do**
11:          Define $M_{\ell,h} \in \mathbb{R}^{SA \times SA}$ as $M_{\ell,h} \leftarrow \text{diag}(\alpha_{s_1,a_1} \ldots \alpha_{s_S,a_A})$, where $\alpha_{s,a} = \mathbf{1}(s \in \mathcal{S}_{\ell,h}^{\text{keep}})$.
12:          $\Phi^\ell \leftarrow \left\{ M_{\ell,h} \left( (\boldsymbol{\pi}_h - \bar{\boldsymbol{\pi}}_{\ell,h}) \widehat{w}_{\ell,h}^{\bar{\pi}} + \boldsymbol{\pi}_h \widehat{\delta}_{\ell,h}^\pi \right) : \pi \in \Pi_\ell \right\}$
13:          $\epsilon_{\text{exp}}^\ell \leftarrow \epsilon_\ell^2 / H^4 \beta_\ell^2$ for $\beta_\ell \leftarrow \left( \sqrt{2 \log \left( \frac{60 S H^2 \ell^2 |\Pi_\ell|}{\delta} \right)} + \frac{4}{3} \sqrt{\frac{SA}{\epsilon_{\text{unif}}^\ell K_{\text{unif}}^\ell} \log \left( \frac{60 H^2 \ell^2 |\Pi_\ell|}{\delta} \right)} \right)$
14:          Run $\mathfrak{D}_{\ell,h}^{\text{ED}} \leftarrow \text{OPTCOV} \left( \Phi^\ell, \epsilon_{\text{exp}}^\ell, \frac{\delta}{6H\ell^2}, \epsilon_{\text{unif}}^\ell, K_{\text{unif}}^\ell, \mathcal{S}_{\ell,h}^{\text{keep}}, h \right)$ (Algorithm 3)
15:          Use $\mathfrak{D}_{\ell,h}^{\text{ED}}$ to compute $\widehat{P}_{\ell,h}(s'|s,a) \leftarrow \frac{N_{\ell,h}(s',s,a)}{N_{\ell,h}(s,a)}$ if $N_{\ell,h}(s,a) > 0$, $\text{unif}(\mathcal{S})$ otherwise, and $\widehat{r}_{\ell,h}(s,a) = \frac{1}{N_{\ell,h}(s,a)} \sum_{(s',a',r',s'') \in \mathfrak{D}_{\ell,h}^{\text{ED}}} r' \cdot \mathbb{I}\{(s,a) = (s',a')\}$ if $N_{\ell,h}(s,a) > 0$, 0 otherwise
16:          Compute $\widehat{\delta}_{\ell,h+1}^\pi \leftarrow M_{\ell,h}(\widehat{P}_{\ell,h}(\boldsymbol{\pi}_h - \bar{\boldsymbol{\pi}}_{\ell,h}) \widehat{w}_{\ell,h}^{\bar{\pi}} + \widehat{P}_{\ell,h} \boldsymbol{\pi}_h \widehat{\delta}_{\ell,h}^\pi)$
17:      **end for**
18:      Compute $\widehat{D}_{\bar{\pi}_\ell}(\pi) \leftarrow \sum_h \langle \widehat{r}_{\ell,h}, \boldsymbol{\pi}_h \widehat{\delta}_{\ell,h} \rangle + \sum_h \langle \widehat{r}_{\ell,h}, (\boldsymbol{\pi}_h - \bar{\boldsymbol{\pi}}_{\ell,h}) \widehat{w}_{\ell,h}^{\bar{\pi}} \rangle$
19:      Update $\Pi_{\ell+1} = \Pi_\ell \backslash \{\pi \in \Pi_\ell : \max_{\pi'} \widehat{D}_{\bar{\pi}_\ell}(\pi') - \widehat{D}_{\bar{\pi}_\ell}(\pi) > 8\epsilon_\ell \}$
20:      **if** $|\Pi_{\ell+1}| = 1$ **then return** $\pi \in \Pi_{\ell+1}$
21: **end for**
22: **return** any $\pi \in \Pi_{\ell+1}$

---

**Value functions.** Note that we can express the value function as:

$$V_h^\pi = \sum_{k=h}^H \left( \prod_{j=h+1}^k P_j \boldsymbol{\pi}_j \right)^\top \boldsymbol{\pi}_k^\top r_k$$

On the "pruned" MDP, define

$$\widetilde{r}_{\ell,h} = M_{\ell,h} r_h,$$

and

$$\widetilde{V}_{\ell,h} := \sum_{k=h}^H \left( \prod_{j=h+1}^k M_{\ell,j+1} P_j \boldsymbol{\pi}_j \right)^\top \boldsymbol{\pi}_k^\top \widetilde{r}_{\ell,k}.$$

**Reward difference term.** Define

$$U_h(\pi, \pi') := \mathbb{E}_{\pi'}[(Q_h^\pi(s_h, \pi_h(s)) - Q_h^\pi(s_h, \pi_h'(s)))^2]$$

and $U(\pi, \pi') := \sum_{h=1}^H U_h(\pi, \pi')$. Additionally, define

$$\widehat{U}_{\ell,h}(\pi, \pi') := \mathbb{E}_{\pi',\ell}[(\widehat{Q}^\pi_{\ell,h}(s_h, \pi_h(s)) - \widehat{Q}^\pi_{\ell,h}(s_h, \pi'_h(s)))^2]$$

where $\mathbb{E}_{\pi',\ell}$ denotes the expectation induced playing $\pi'$ on the MDP with transitions $\widehat{P}_\ell$, and $\widehat{Q}^\pi_{\ell,h}$ denotes the $Q$-function for policy $\pi$ on this same MDP. Let $\widehat{U}_\ell(\pi, \pi') := \sum_{h=1}^H \widehat{U}_{\ell,h}(\pi, \pi')$.

## C.2 Technical Results

**Lemma 7.** *Let* $\mathfrak{D} = \{(s_1, a_1, s'_1), \ldots (s_T, a_T, s'_T)\}$ *be any dataset of transitions collected from level* $h$. *Let* $\widehat{P} \in \mathbb{R}^{S \times SA}$ *denote the empirical transition matrix with* $[\widehat{P}]_{s',sa} = \frac{N(s'|s,a)}{N(s,a)}$ *if* $N(s,a) > 0$, *and 0 otherwise, for* $N(s' \mid s, a) = \sum_t \mathbb{I}\{(s_t, a_t, s'_t) = (s, a, s')\}$ *and* $N(s, a) = \sum_t \mathbb{I}\{(s_t, a_t) = (s, a)\}$. *Consider any* $v \in [0, 1]^S$ *and* $u \in \mathbb{R}^{SA}$ *and assume that* $N(s, a) > \underline{\lambda} > 0$ *for all* $(s, a) \in \mathrm{support}(u)$. *Then, for* $P$ *the true transition matrix, we have that with probability at least* $1 - \delta$:

$$\left| v^\top (P - \widehat{P}) u \right| \le \sqrt{\sum_{s,a} \frac{[u]^2_{s,a}}{N(s,a)}} \cdot \left( \sqrt{2 \log \left( \frac{1}{\delta} \right)} + \frac{4}{3\sqrt{\underline{\lambda}}} \log \left( \frac{1}{\delta} \right) \right).$$

*Proof.* First write

$$v^\top (P - \widehat{P}) u = \sum_{s'} \sum_{s,a} v_{s'} \left( P(s' \mid s, a) - \frac{N(s' \mid s, a)}{N(s, a)} \right) u_{sa}$$

$$= \sum_t \sum_{s'} \frac{v_{s'} \left( P(s' \mid s_t, a_t) - \mathbb{I}\{s'_t = s'\} \right) u_{s_t a_t}}{N(s_t, a_t)}$$

where the second equality follows from some simple manipulations. Note that, for any $t$, we have

$$\mathbb{E}\left[ \frac{v_{s'} \left( P(s' \mid s_t, a_t) - \mathbb{I}\{s'_t = s'\} \right) u_{s_t a_t}}{N(s_t, a_t)} \mid s_t, a_t \right] = 0$$

and can bound

$$\left| \sum_{s'} \frac{v_{s'} \left( P(s' \mid s_t, a_t) - \mathbb{I}\{s'_t = s'\} \right) u_{s_t a_t}}{N(s_t, a_t)} \right| \le \frac{2 u_{s_t a_t}}{N(s_t, a_t)} \le \frac{2}{\sqrt{\underline{\lambda}}} \cdot \frac{u_{s_t a_t}}{\sqrt{N(s_t, a_t)}}$$

$$\le \frac{2}{\sqrt{\underline{\lambda}}} \cdot \sqrt{\sum_{s,a} \frac{u^2_{sa}}{N(s, a)}}$$

where we have used the fact that $N(s, a) \ge \underline{\lambda}$ for $(s, a) \in \mathrm{support}(u)$, and since $v$ has entries in $[0, 1]$ and $P(s' \mid s_t, a_t)$ and $\mathbb{I}\{s'_t = s'\}$ are valid distributions, so $\sum_{s'} v_{s'} (P(s' \mid s_t, a_t) - \mathbb{I}\{s'_t = s'\}) \in [-1, 1]$. Furthermore, we have that

$$\mathbb{E}_{s'_t}\left[ \left( \sum_{s'} \frac{v_{s'} \left( P(s' \mid s_t, a_t) - \mathbb{I}\{s'_t = s'\} \right) u_{s_t a_t}}{N(s_t, a_t)} \right)^2 \right] \le \mathbb{E}_{s'_t}\left[ \left( \frac{u_{s_t a_t}}{N(s_t, a_t)} \right)^2 \right] = \left( \frac{u_{s_t a_t}}{N(s_t, a_t)} \right)^2$$

where we have again used that $\sum_{s'} v_{s'} (P(s' \mid s_t, a_t) - \mathbb{I}\{s'_t = s'\}) \in [-1, 1]$.

By Bernstein's inequality, we therefore have that with probability at least $1 - \delta$:

$$\left| v^\top (P - \widehat{P}) u \right| \le \sqrt{2 \sum_t \left( \frac{u_{s_t a_t}}{N(s_t, a_t)} \right)^2 \cdot \log \frac{2}{\delta}} + \frac{4}{3\sqrt{\underline{\lambda}}} \cdot \sqrt{\sum_t \frac{u^2_{s_t a_t}}{N(s_t, a_t)}} \cdot \log \frac{2}{\delta}$$

$$= \left( \sqrt{2 \log \frac{2}{\delta}} + \frac{4}{3\sqrt{\underline{\lambda}}} \log \frac{2}{\delta} \right) \cdot \sqrt{\sum_{s,a} \frac{u^2_{sa}}{N(s, a)}}.$$

$\square$

**Lemma 8.** *Let $\mathfrak{D} = \{(s_1, a_1, r_1), \ldots (s_T, a_T, r_T)\}$ be any dataset of state-action-reward tuples collected from level $h$. Let $\widehat{r} \in \mathbb{R}^{SA}$ denote the empirical reward estimation with $[\widehat{r}]_{sa} = \frac{1}{N(s,a)} \cdot \sum_{t=1}^{T} r_t \cdot \mathbb{I}\{(s_t, a_t) = (s, a)\}$ if $N(s, a) > 0$, and 0 otherwise, for $N(s, a) = \sum_t \mathbb{I}\{(s_t, a_t) = (s, a)\}$. Consider any $u \in \mathbb{R}^{SA}$ and assume that $N(s, a) > \underline{\lambda} > 0$ for all $(s, a) \in \mathrm{support}(u)$. Then, for $r$ the true reward mean, we have that with probability at least $1 - \delta$:*

$$\left| (r - \widehat{r})^\top u \right| \le \sqrt{\sum_{s,a} \frac{[u]_{s,a}^2}{N(s,a)}} \cdot \left( \sqrt{2 \log\left(\frac{1}{\delta}\right)} + \frac{4}{3\sqrt{\underline{\lambda}}} \log\left(\frac{1}{\delta}\right) \right).$$

*Proof.* First write

$$(r - \widehat{r})^\top u = \sum_t \frac{(r(s_t, a_t) - r_t)\, u_{s_t a_t}}{N(s_t, a_t)}.$$

Note that, for any $t$, we have

$$\mathbb{E}\left[ \frac{(r(s_t, a_t) - r_t)\, u_{s_t a_t}}{N(s_t, a_t)} \mid s_t, a_t \right] = 0$$

and can bound

$$\left| \frac{(r(s_t, a_t) - r_t)\, u_{s_t a_t}}{N(s_t, a_t)} \right| \le \frac{u_{s_t a_t}}{N(s_t, a_t)} \le \frac{1}{\sqrt{\underline{\lambda}}} \cdot \frac{u_{s_t a_t}}{\sqrt{N(s_t, a_t)}} \le \frac{1}{\sqrt{\underline{\lambda}}} \cdot \sqrt{\sum_{s,a} \frac{u_{sa}^2}{N(s,a)}}$$

where we have used the fact that $N(s, a) \ge \underline{\lambda}$ for $(s, a) \in \mathrm{support}(u)$, and since we assume our rewards are in $[0, 1]$. Furthermore, we have that

$$\mathbb{E}_{r_t}\left[ \left( \frac{(r(s_t, a_t) - r_t)\, u_{s_t a_t}}{N(s_t, a_t)} \right)^2 \right] \le \mathbb{E}_{r_t}\left[ \left( \frac{u_{s_t a_t}}{N(s_t, a_t)} \right)^2 \right] = \left( \frac{u_{s_t a_t}}{N(s_t, a_t)} \right)^2.$$

By Bernstein's inequality, we therefore have that with probability at least $1 - \delta$:

$$\left| (r - \widehat{r})^\top u \right| \le \sqrt{2 \sum_t \left( \frac{u_{s_t a_t}}{N(s_t, a_t)} \right)^2 \cdot \log\frac{2}{\delta}} + \frac{4}{3\sqrt{\underline{\lambda}}} \cdot \sqrt{\sum_t \frac{u_{s_t a_t}^2}{N(s_t, a_t)}} \cdot \log\frac{2}{\delta}$$

$$= \left( \sqrt{2 \log\frac{2}{\delta}} + \frac{4}{3\sqrt{\underline{\lambda}}} \log\frac{2}{\delta} \right) \cdot \sqrt{\sum_{s,a} \frac{u_{sa}^2}{N(s,a)}}.$$

$\square$

**Lemma 9.** *Let $u \in \mathbb{R}^S$ be any vector such that $\forall s, |u_s| \le M$. Then, for any $(\ell, h)$, the following holds with probability $(1 - \delta)$:*

$$\left| \mathbb{E}_{s \sim w_{\ell,h}^{\bar{\pi}}}[u_s] - \mathbb{E}_{s \sim \widehat{w}_{\ell,h}^{\bar{\pi}}}[u_s] \right| \le \sqrt{\frac{2 \mathbb{E}_{s \sim w_{\ell,h}^{\bar{\pi}}}[u_s^2]}{\bar{n}_\ell} \log\left(\frac{2}{\delta}\right)} + \frac{2M}{3\bar{n}_\ell} \log\left(\frac{2}{\delta}\right)$$

*Proof.* The left side of the inequality above takes the form of the deviation between an empirical and true mean of the random variable $u_s$. Hence, the result follows directly from Bernstein's inequality since we know $|u_s| \le M$ is bounded. $\square$

**Lemma 10.** *Assume that $A$ and $B$ are matrices with entries in $[0, 1]$ and whose rows sum to a value $\le 1$. Then $AB$ also satisfies this.*

*Proof.* To see this, consider the $i$th row of $AB$, and note that the sum of the elements in this row can be written as, for $a_i^\top$ the $i$th row of $A$, and $b_j$ the $j$th column of $B$:

$$\sum_j a_i^\top b_j = \sum_k \sum_j a_{ik} b_{jk} = \sum_k a_{ik} \left( \sum_j b_{jk} \right).$$

Now note that $\sum_j b_{jk}$ is the sum across the $k$th row of $B$, so this is $\le 1$ by assumption. Furthermore, $\sum_k a_{ik} \le 1$ for the same reason. Thus, the $i$th row of $AB$ sums to a value $\le 1$. Furthermore, it is easy to see $a_i^\top b_j \le 1$ for each $j$. Thus, $AB$ has values in $[0, 1]$ and rows that sum to a value $\le 1$. $\square$

**Lemma 11.** *We have that $\|\Pi_{h=i}^j M_{h+1} P_h \boldsymbol{\pi}_h\|_2, \|\Pi_{h=i}^j P_h \boldsymbol{\pi}_h\|_2 \le \sqrt{S}$ for any $i, j, h$.*

*Proof.* By definition $P_h \boldsymbol{\pi}_h$ is a transition matrix—each row has values in $[0, 1]$ and sums to 1—and $M_{h+1}$ is diagonal with diagonal elements either 0 or 1. Thus, each matrix $M_h P_h \boldsymbol{\pi}_h$ has values in $[0, 1]$ and rows that sum to a value $\le 1$, so Lemma 10 implies that $\Pi_{h=i}^j M_{h+1} P_h \boldsymbol{\pi}_h$ does as well. Denote $A := \|\Pi_{h=i}^j M_h P_h \boldsymbol{\pi}_h\|_2$. We can then bound

$$\|\Pi_{h=i}^j M_{h+1} P_h \boldsymbol{\pi}_h\|_2^2 = \|A\|_2^2 \le \|A\|_{\mathrm{F}}^2 = \sum_i \sum_j A_{ij}^2 \le \sum_i 1 \le S,$$

which proves the result. The bound on $\|\Pi_{h=i}^j P_h \boldsymbol{\pi}_h\|_2$ follows from the same argument. $\square$

**Lemma 12.** *We have*

$$\widetilde{\delta}_{\ell,h+1}^{\pi} - \widehat{\delta}_{\ell,h+1}^{\pi}$$
$$= \sum_{i=0}^{h-2} \left( \prod_{j=h-i+1}^{h} M_{\ell,j+1} P_j \boldsymbol{\pi}_j \right) (P_{h-i} - \widehat{P}_{\ell,h-i}) M_{\ell,h-i} \left[ (\boldsymbol{\pi}_{h-i} - \bar{\boldsymbol{\pi}}_{\ell,h-i}) \widehat{w}_{\ell,h-i}^{\bar{\pi}} + \boldsymbol{\pi}_{h-i} \widehat{\delta}_{\ell,h-i}^{\pi} \right].$$

*Proof.* This follows immediately from the definition of $\widetilde{\delta}_{\ell,h+1}^{\pi}, \widehat{\delta}_{\ell,h+1}^{\pi}$, and simple manipulations. $\square$

### C.3 Concentration Arguments and Good Events

**Lemma 13.** *Let $\mathcal{E}_{\mathrm{prune}}^{\ell}$ be the event for which the call to PRUNE in epoch $\ell$ in Algorithm 2 will terminate after running for at most*

$$\mathrm{poly}(S, A, H, \log \frac{SAH\ell}{\delta \epsilon_\ell}) \cdot \frac{1}{\epsilon_{\mathrm{unif}}^{\ell}}$$

*episodes and will return a set $\mathcal{S}_\ell^{\mathrm{keep}}$ such that, for every $(s, h) \in \mathcal{S}_\ell^{\mathrm{keep}}$, we have $W_h^{\star}(s) \ge \epsilon_{\mathrm{unif}}^{\ell}$, and, if $(s, h) \notin \mathcal{S}_\ell^{\mathrm{keep}}$, then $W_h^{\star}(s) \le 32 \epsilon_{\mathrm{unif}}^{\ell}$. Then $\mathbb{P}(\mathcal{E}_{\mathrm{prune}}^{\ell}) \ge 1 - \frac{\delta}{3\ell^2}$.*

*Proof.* From Lemma 38, this event follows directly with probability $(1 - \frac{\delta}{3\ell^2})$. $\square$

**Lemma 14.** *Let $\mathcal{E}_{\mathrm{exp}}^{\ell,h}$ be the event for which:*

*1. The exploration procedure in Algorithm 3 will produce $\mathfrak{D}_{\ell,h}^{\mathrm{ED}}$ such that*

$$\max_{\pi \in \Pi_\ell} \|M_{\ell,h}((\boldsymbol{\pi}_h - \bar{\boldsymbol{\pi}}_{\ell,h}) \widehat{w}_{\ell,h}^{\bar{\pi}} + \boldsymbol{\pi}_h \widehat{\delta}_{\ell,h}^{\pi})\|_{\widehat{\Lambda}_{\ell,h}^{-1}}^2 \le \epsilon_{\mathrm{exp}}^{\ell} \quad \text{for} \quad \widehat{\Lambda}_{\ell,h} = \sum_{(s,a) \in \mathfrak{D}_{\ell,h}^{\mathrm{ED}}} e_{sa} e_{sa}^{\top}, \quad \text{(C.4)}$$

*and will collect at most*

$$C \cdot \frac{\inf_{\pi_{\mathrm{exp}}} \max_{\pi \in \Pi_\ell} \|M_{\ell,h}((\boldsymbol{\pi}_h - \bar{\boldsymbol{\pi}}_{\ell,h}) \widehat{w}_{\ell,h}^{\bar{\pi}} + \boldsymbol{\pi}_h \widehat{\delta}_{\ell,h}^{\pi})\|_{\Lambda_h(\pi_{\mathrm{exp}})^{-1}}^2}{\epsilon_{\mathrm{exp}}^{\ell}} + \frac{C_{\mathrm{fw}}^{\ell}}{(\epsilon_{\mathrm{exp}}^{\ell})^{4/5}}$$
$$+ \frac{C_{\mathrm{fw}}^{\ell}}{\epsilon_{\mathrm{unif}}^{\ell}} + \log(C_{\mathrm{fw}}^{\ell}) \cdot K_{\mathrm{unif}}^{\ell}$$

*episodes.*

*2. For each $s \in \mathcal{S}_\ell^{\mathrm{keep}}$, we have that $\sum_{(s',a') \in \mathfrak{D}_{\ell,h}^{\mathrm{ED}}} \mathbb{I}\{(s', a') = (s, a)\} \ge \frac{K_{\mathrm{unif}}^{\ell} \epsilon_{\mathrm{unif}}^{\ell}}{SA}$ for any $a \in \mathcal{A}$.*

*Above, $C$ is a universal constant and $C_{\mathrm{fw}}^{\ell} = \mathrm{poly}(S, A, H, \log \ell/\delta, \log 1/\epsilon, \log |\Pi|)$. Then $\mathbb{P}[(\mathcal{E}_{\mathrm{exp}}^{\ell,h})^c \cap \mathcal{E}_{\mathrm{prune}}^{\ell} \cap \bar{\mathcal{E}}_{\mathrm{est}}^{\ell} \cap (\cap_{h' \le h-1} \mathcal{E}_{\mathrm{est}}^{\ell,h'}) \cap (\cap_{h' \le h-1} \mathcal{E}_{\mathrm{exp}}^{\ell,h'})] \le \frac{\delta}{6H\ell^2}$.*

*Proof.* Since the event $\mathcal{E}_{\text{prune}}^\ell$ holds, for each $s \in \mathcal{S}_\ell^{\text{keep}}$ we have $W_h^\star(s) \geq \epsilon_{\text{unif}}^\ell$. Now, observe that, for $s \in \mathcal{S}_\ell^{\text{keep}}$ and any $a$:

$$|[(\boldsymbol{\pi}_h - \bar{\boldsymbol{\pi}}_{\ell,h'})\widehat{w}_{\ell,h}^{\bar{\pi}} + \boldsymbol{\pi}_h \widehat{\delta}_{\ell,h}^\pi]_{(s,a)}|$$

$$\leq [\widehat{w}_{\ell,h}^{\bar{\pi}}]_s + |[\widehat{\delta}_{\ell,h}^\pi]_s| \leq [w_{\ell,h}^{\bar{\pi}}]_s + |[\delta_{\ell,h}^\pi]_s| + |[\widehat{w}_{\ell,h}^{\bar{\pi}} - w_{\ell,h}^{\bar{\pi}}]_{(s)}| + |[\delta_{\ell,h}^\pi]_s - |[\widehat{\delta}_{\ell,h}^\pi]_s||.$$

By construction, we have $[w_{\ell,h}^{\bar{\pi}}]_s, |[\delta_{\ell,h}^\pi]_s| \leq W_h^\star(s)$. By Lemma 19, on $\bar{\mathcal{E}}_{\text{est}}^\ell$, we can bound $|[\widehat{w}_{\ell,h}^{\bar{\pi}} - w_{\ell,h}^{\bar{\pi}}]_{(s)}| \leq \sqrt{8S\epsilon_\ell^{5/3}}$. By Lemma 18, on $\mathcal{E}_{\text{prune}}^\ell \cap (\cap_{h' \leq h-1} \mathcal{E}_{\text{est}}^{\ell,h'}) \cap (\cap_{h' \leq h-1} \mathcal{E}_{\text{exp}}^{\ell,h'})$, we can bound

$$|[\delta_{\ell,h}^\pi]_s - |[\widehat{\delta}_{\ell,h}^\pi]_s|| \leq \sqrt{SH\beta_\ell \epsilon_{\text{exp}}^\ell} + SH(\sqrt{8\epsilon_\ell^{5/3}} + 32\epsilon_{\text{unif}}^\ell).$$

Altogether then, we have

$$|[(\boldsymbol{\pi}_h - \bar{\boldsymbol{\pi}}_{\ell,h'})\widehat{w}_{\ell,h}^{\bar{\pi}} + \boldsymbol{\pi}_h \widehat{\delta}_{\ell,h}^\pi]_{(s,a)}|$$

$$\leq 2W_h^\star(s) + \sqrt{SH\beta_\ell \epsilon_{\text{exp}}^\ell} + SH(\sqrt{8\epsilon_\ell^{5/3}} + 32\epsilon_{\text{unif}}^\ell) + \sqrt{8S\epsilon_\ell^{5/3}}.$$

By our choice of $\epsilon_{\text{exp}}^\ell$ and $\epsilon_{\text{unif}}^\ell$, we can bound all of this as

$$\leq C_\phi \cdot (W_h^\star(s) + \sqrt{K_{\text{unif}}^\ell \epsilon_{\text{unif}}^\ell \epsilon_{\text{exp}}^\ell})$$

for $C_\phi = cSH\beta_\ell$. This is the condition required by Theorem 2, so the result follows from Theorem 2. □

**Lemma 15.** *Let $\mathcal{E}_{\text{est}}^{\ell,h}$ be the event at epoch $\ell$ for step $h$ on which:*

*(1) For all $\pi \in \Pi_\ell$, $h' \leq h$:*

$$\left| \left\langle \boldsymbol{\pi}_h^\top \widetilde{r}_{\ell,h}, \left(\prod_{i=h'+1}^h M_{\ell,i+1} P_i \boldsymbol{\pi}_i\right)(P_{h'} - \widehat{P}_{\ell,h'}) M_{\ell,h'} \left[(\boldsymbol{\pi}_{h'} - \bar{\boldsymbol{\pi}}_{\ell,h'})\widehat{w}_{\ell,h'}^{\bar{\pi}} + \boldsymbol{\pi}_{h'} \widehat{\delta}_{\ell,h'}^\pi\right] \right\rangle \right|$$

$$\leq \beta_\ell \sqrt{\sum_{s,a} \frac{\left[M_{\ell,h'}\left((\boldsymbol{\pi}_{h'} - \bar{\boldsymbol{\pi}}_{\ell,h'})\widehat{w}_{\ell,h'}^{\bar{\pi}} + \boldsymbol{\pi}_{h'} \widehat{\delta}_{\ell,h'}^\pi\right)\right]_{(s,a)}^2}{N_{\ell,h'}(s,a)}}.$$

*(2) For all canonical vectors $e_{s'}$ in $\mathbb{R}^S$, $\pi \in \Pi_\ell$, and $h' \leq h$,*

$$\left| \left\langle e_{s'}, \left(\prod_{i=h'+1}^h M_{\ell,i+1} P_i \boldsymbol{\pi}_i\right)(P_{h'} - \widehat{P}_{\ell,h'}) M_{\ell,h'} \left[(\boldsymbol{\pi}_{h'} - \bar{\boldsymbol{\pi}}_{\ell,h'})\widehat{w}_{\ell,h'}^{\bar{\pi}} + \boldsymbol{\pi}_{h'} \widehat{\delta}_{\ell,h'}^\pi\right] \right\rangle \right|$$

$$\leq \beta_\ell \sqrt{\sum_{s,a} \frac{[M_{\ell,h'}((\boldsymbol{\pi}_{h'} - \bar{\boldsymbol{\pi}}_{\ell,h'})\widehat{w}_{\ell,h'}^{\bar{\pi}} + \boldsymbol{\pi}_{h'} \widehat{\delta}_{\ell,h'}^\pi)]_{s,a}^2}{N_{\ell,h'}(s,a)}}.$$

*(3) For each $(s,a)$, we have*

$$\sum_{s'} |\widehat{P}_{\ell,h}(s' \mid s,a) - P_h(s' \mid s,a)| \leq S\sqrt{\frac{\log \frac{48S^2 AH\ell^2}{\delta}}{N_{\ell,h}(s,a)}}.$$

*(4) For each $\pi \in \Pi_\ell$,*

$$|\langle \widehat{r}_{\ell,h} - \widetilde{r}_{\ell,h}, \boldsymbol{\pi}_h \widehat{\delta}_{\ell,h}^\pi + (\boldsymbol{\pi}_h - \bar{\boldsymbol{\pi}}_{\ell,h})\widehat{w}_{\ell,h}^{\bar{\pi}}\rangle|$$

$$\leq \beta_\ell \sqrt{\sum_{s,a} \frac{\left[M_{\ell,h}\left((\boldsymbol{\pi}_h - \bar{\boldsymbol{\pi}}_{\ell,h'})\widehat{w}_{\ell,h}^{\bar{\pi}} + \boldsymbol{\pi}_h \widehat{\delta}_{\ell,h}^\pi\right)\right]_{(s,a)}^2}{N_{\ell,h}(s,a)}}.$$

*Then $\mathbb{P}[(\mathcal{E}_{\text{est}}^{\ell,h})^c \cap \mathcal{E}_{\text{prune}}^\ell \cap (\cap_{h' \leq h} \mathcal{E}_{\text{exp}}^{\ell,h})] \leq \frac{\delta}{6H\ell^2}$.*

*Proof.* We prove each of the events sequentially.

**Proof of Event (1).** Consider any fixed choice of $(\pi, h')$. By Lemma 10 and since our rewards are in $[0,1]$, we have that $\left(\prod_{i=h'+1}^{h} M_{\ell,i+1} P_i \boldsymbol{\pi}_i\right)^{\top} \boldsymbol{\pi}_h^{\top} \widetilde{r}_{\ell,h}$ is a vector in $[0,1]$. Let $v \leftarrow \left(\prod_{i=h'+1}^{h} M_{\ell,i+1} P_i \boldsymbol{\pi}_i\right)^{\top} \boldsymbol{\pi}_h^{\top} \widetilde{r}_{\ell,h}$ and $u \leftarrow M_{\ell,h'}\left[(\boldsymbol{\pi}_{h'} - \bar{\boldsymbol{\pi}}_{\ell,h'}) \widehat{w}_{\ell,h'}^{\bar{\pi}} + \boldsymbol{\pi}_{h'} \widehat{\delta}_{\ell,h'}^{\pi}\right]$. Note that by construction we have that $u_{sa} = 0$ for $s \notin \mathcal{S}_{\ell,h'}^{\mathrm{keep}}$, and so on $\mathcal{E}_{\exp}^{\ell,h'}$, we have $N_{\ell,h'}(s,a) \geq \frac{K_{\mathrm{unif}}^{\ell} \epsilon_{\mathrm{unif}}^{\ell}}{2SA}$ for all $(s,a) \in \mathrm{support}(u)$. On $\mathcal{E}_{\mathrm{prune}}^{\ell} \cap \mathcal{E}_{\exp}^{\ell,h'}$, we can then apply Lemma 7 with $u$ and $v$ as defined above to get that the bound fails with probability at most $\frac{\delta}{30H^2\ell^2|\Pi_\ell|}$. Union bounding over $h'$ and $\pi$ we get that the stated result fails with probability at most $\frac{\delta}{30H\ell^2}$.

**Proof of Event (2).** Choose

$$v = e_i^{\top} \left(\prod_{i=h'+1}^{h} M_{\ell,i} P_i \boldsymbol{\pi}_i\right) \quad \text{and} \quad u = M_{h',\ell}\left((\boldsymbol{\pi}_{h'} - \bar{\boldsymbol{\pi}}_{\ell,h'}) w_{\ell,h'}^{\bar{\pi}} + \boldsymbol{\pi}_{h'} \widehat{\delta}_{\ell,h'}^{\pi}\right).$$

Note that by construction of $w_{\ell,h'}^{\bar{\pi}}$ and $\widehat{\delta}_{\ell,h'}^{\pi}$ we have that $u_{sa} = 0$ for $s \notin \mathcal{S}_{\ell,h'}^{\mathrm{keep}}$, and so on $\mathcal{E}_{\exp}^{\ell,h'}$, we have $N_{\ell,h'}(s,a) \geq \frac{K_{\mathrm{unif}}^{\ell} \epsilon_{\mathrm{unif}}^{\ell}}{2SA}$ for all $(s,a) \in \mathrm{support}(u)$. Furthermore, we have that $v \in [0,1]^S$ by Lemma 10. Then, the event follows by invoking Lemma 7.

**Proof of Event (3).** By Hoeffding's inequality, for any $(s,a)$, we have, with probability at least $1 - \frac{\delta}{24S^2AH\ell^2}$:

$$|\widehat{P}_{\ell,h}(s' \mid s,a) - P_h(s' \mid s,a)| \leq \sqrt{\frac{\log \frac{24S^2AH\ell^2}{\delta}}{N_{\ell,h}(s,a)}}.$$

Thus, we have that with probability at least $1 - \frac{\delta}{24SAH\ell^2}$:

$$\sum_{s'} |\widehat{P}_{\ell,h}(s' \mid s,a) - P_h(s' \mid s,a)| \leq S\sqrt{\frac{\log \frac{24S^2AH\ell^2}{\delta}}{N_{\ell,h}(s,a)}}.$$

Union bounding over all $(s,a)$, we obtain that this holds with probability at least $1 - \frac{\delta}{24H\ell^2}$.

**Proof of Event (4).** Note first that $\langle \widehat{r}_{\ell,h} - \widetilde{r}_{\ell,h}, \boldsymbol{\pi}_h \widehat{\delta}_{\ell,h}^{\pi} + (\boldsymbol{\pi}_h - \bar{\boldsymbol{\pi}}_{\ell,h}) \widehat{w}_{\ell,h}^{\bar{\pi}}\rangle = \langle \widehat{r}_{\ell,h} - \widetilde{r}_{\ell,h}, M_{\ell,h}(\boldsymbol{\pi}_h \widehat{\delta}_{\ell,h}^{\pi} + (\boldsymbol{\pi}_h - \bar{\boldsymbol{\pi}}_{\ell,h}) \widehat{w}_{\ell,h}^{\bar{\pi}})\rangle$. The result then follows on $\mathcal{E}_{\mathrm{prune}}^{\ell}$ by a direct application of Lemma 8.

The final result then holds by a union bound. $\qquad\square$

**Lemma 16.** *Let $\bar{\mathcal{E}}_{\mathrm{est}}^{\ell}$ denote the event that at epoch $\ell$ and for each $h$:*

*(1) For all $\pi \in \Pi_\ell$ and $h \in [H]$, we have*

$$\left|\langle P_h^{\top} M_{\ell,h+1} \widetilde{V}_{\ell,h+1} + r_h, (\boldsymbol{\pi}_h - \bar{\boldsymbol{\pi}}_{\ell,h})(w_{\ell,h}^{\bar{\pi}} - \widehat{w}_{\ell,h}^{\bar{\pi}})\rangle\right| \leq \frac{2H}{3\bar{n}_\ell} \log \frac{60H\ell^2|\Pi_\ell|}{\delta}$$

$$+ \sqrt{\frac{2\mathbb{E}_{s \sim w_{\ell,h}^{\bar{\pi}_\ell}}[\langle P_h^{\top} M_{\ell,h+1} \widetilde{V}_{\ell,h+1}^{\pi} + r_h, (\boldsymbol{\pi}_h - \bar{\boldsymbol{\pi}}_{\ell,h}) e_s\rangle^2]}{\bar{n}_\ell}} \cdot \log \frac{60H\ell^2|\Pi_\ell|}{\delta}.$$

*(2) For all canonical vectors $e_s \in \mathbb{R}^S$,*

$$|\langle e_s, \widehat{w}_{\ell,h}^{\bar{\pi}} - w_{\ell,h}^{\bar{\pi}}\rangle| \leq \sqrt{\frac{2\log\left(\frac{30H\ell^2 S}{\delta}\right)}{\bar{n}_\ell}} + \frac{2\log\left(\frac{30H\ell^2 S}{\delta}\right)}{\bar{n}_\ell}.$$

*Then $\mathbb{P}[(\bar{\mathcal{E}}_{\mathrm{est}}^{\ell})^c] \leq \frac{\delta}{15\ell^2}$.*

*Proof.* **Proof of Event (1).** Consider a fixed choice of $\pi$, and let $u_s^\pi = \left\langle P_h^\top \widetilde{V}_{\ell,h+1}^\pi + r_h, (\boldsymbol{\pi}_h - \bar{\boldsymbol{\pi}}_{\ell,h})e_s \right\rangle$, and note that $|u_s^\pi| \leq H$ for all $s$. Lemma 9 then gives that with probability at least $1 - \frac{\delta}{30H\ell^2|\Pi_\ell|}$ we have

$$\left| \langle P_h^\top M_{\ell,h+1}\widetilde{V}_{\ell,h+1} + r_h, (\boldsymbol{\pi}_h - \bar{\boldsymbol{\pi}}_{\ell,h})(w_{\ell,h}^{\bar{\pi}} - \widehat{w}_{\ell,h}^{\bar{\pi}}) \rangle \right|$$

$$\leq \sqrt{\frac{2\mathbb{E}_{s\sim w_{\ell,h}^{\bar{\pi}_\ell}}[\langle P_h^\top M_{\ell,h+1}\widetilde{V}_{\ell,h+1}^\pi + r_h, (\boldsymbol{\pi}_h - \bar{\boldsymbol{\pi}}_{\ell,h})e_s \rangle^2]}{\bar{n}_\ell} \cdot \log \frac{60H\ell^2|\Pi_\ell|}{\delta}} + \frac{2H}{3\bar{n}_\ell}\log\frac{60H\ell^2|\Pi_\ell|}{\delta}.$$

**Proof of Event (2).** For a fixed choice of $s \in [S]$, the event follows from Lemma 9 with $u = e_s$ with probability $1 - \delta$, where $\delta = \frac{\delta}{30H\ell^2 S}$. Once we take the union bound over all $s \in [S]$, then the event follows with probability $1 - \frac{\delta}{30H\ell^2}$.

The result then holds by union bounding over each of these for all $h$. $\qquad\square$

**Lemma 17.** *On $\mathcal{E}_{\mathrm{prune}}^\ell$, for all $h$ and $\pi$ we have*

$$\delta_{\ell,h+1}^\pi - \widetilde{\delta}_{\ell,h+1}^\pi$$
$$= \sum_{i=0}^{h-2}\left(\prod_{j=h-i+1}^{h} M_{\ell,j+1}P_j\boldsymbol{\pi}_j\right)M_{\ell,h-i+1}P_{h-i}(\boldsymbol{\pi}_{h-i} - \bar{\boldsymbol{\pi}}_{h-i})(w_{\ell,h-i}^{\bar{\pi}_\ell} - \widehat{w}_{\ell,h-i}^{\bar{\pi}_\ell}) + \Delta_{\ell,h+1}^\pi$$

*for some $\Delta_{\ell,h}^\pi \in \mathbb{R}^S$ with $\|\Delta_{\ell,h}^\pi\|_2 \leq 32SH\epsilon_{\mathrm{unif}}^\ell$. Furthermore, for any $\pi$ and any $i, k$ satisfying $0 \leq i \leq k \leq H$, we have*

$$\left\|\left(\prod_{j=i}^{k} M_{\ell,j+1}P_j\boldsymbol{\pi}_j - \prod_{j=i}^{k} P_j\boldsymbol{\pi}_j\right)w_i^\pi\right\|_2 \leq 32SH\epsilon_{\mathrm{unif}}^\ell.$$

*Proof.* By definition, we have that

$$\delta_{\ell,h+1}^\pi - \widetilde{\delta}_{\ell,h+1}^\pi$$
$$= P_h(\boldsymbol{\pi}_h - \bar{\boldsymbol{\pi}}_{\ell,h})w_{\ell,h}^{\bar{\pi}_\ell} + P_h\boldsymbol{\pi}_h\delta_{\ell,h}^\pi - M_{\ell,h+1}P_h(\boldsymbol{\pi}_h - \bar{\boldsymbol{\pi}}_{\ell,h})\widehat{w}_{\ell,h}^{\bar{\pi}} - M_{\ell,h+1}P_h\boldsymbol{\pi}_h\widetilde{\delta}_{\ell,h}^\pi$$
$$= (I - M_{\ell,h+1})P_h(\boldsymbol{\pi}_h - \bar{\boldsymbol{\pi}}_{\ell,h})w_{\ell,h}^{\bar{\pi}_\ell} + M_{\ell,h+1}P_h(\boldsymbol{\pi}_h - \bar{\boldsymbol{\pi}}_{\ell,h})(w_{\ell,h}^{\bar{\pi}_\ell} - \widehat{w}_{\ell,h}^{\bar{\pi}})$$
$$\quad + (I - M_{\ell,h+1})P_h\boldsymbol{\pi}_h\delta_{\ell,h}^\pi + M_{\ell,h+1}P_h\boldsymbol{\pi}_h(\delta_{\ell,h}^\pi - \widetilde{\delta}_{\ell,h}^\pi)$$

$$\vdots$$

$$= \sum_{i=0}^{h-2}\left(\prod_{j=h-i+1}^{h} M_{\ell,j+1}P_j\boldsymbol{\pi}_j\right)\Bigg[(I - M_{\ell,h-i+1})P_{h-i}(\boldsymbol{\pi}_{h-i} - \bar{\boldsymbol{\pi}}_{h-i})w_{\ell,h-i}^{\bar{\pi}_\ell}$$

$$\quad + M_{\ell,h-i+1}P_{h-i}(\boldsymbol{\pi}_{h-i} - \bar{\boldsymbol{\pi}}_{h-i})(w_{\ell,h-i}^{\bar{\pi}_\ell} - \widehat{w}_{\ell,h-i}^{\bar{\pi}_\ell}) + (I - M_{\ell,h-i+1})P_{h-i}\boldsymbol{\pi}_{h-i}\delta_{\ell,h-i}^\pi\Bigg].$$

Note that $[P_{h-i}(\boldsymbol{\pi}_{h-i} - \bar{\boldsymbol{\pi}}_{h-i})w_{\ell,h'}^{\bar{\pi}_\ell}]_s \leq W_{h-i+1}^\star(s)$, and similarly $[P_{h-i}\boldsymbol{\pi}_{h-i}\delta_{\ell,h-i}^\pi]_s \leq W_{h-i+1}^\star(s)$. On the event $\mathcal{E}_{\mathrm{prune}}^\ell$, we have that if $[M_{\ell,h-i+1}]_{s,s} = 0$, then $W_{h-i+1}^\star(s) \leq 32\epsilon_{\mathrm{unif}}^\ell$. It follows from this that every non-zero element in $(I - M_{\ell,h-i+1})P_{h-i}(\boldsymbol{\pi}_{h-i} - \bar{\boldsymbol{\pi}}_{h-i})w_{\ell,h-i}^{\bar{\pi}_\ell}$ and $(I - M_{\ell,h-i+1})P_{h-i}\boldsymbol{\pi}_{h-i}\delta_{\ell,h-i}^\pi$ is bounded by $32\epsilon_{\mathrm{unif}}^\ell$, so:

$$\|(I - M_{\ell,h-i+1})P_{h-i}(\boldsymbol{\pi}_{h-i} - \bar{\boldsymbol{\pi}}_{h-i})w_{\ell,h-i}^{\bar{\pi}_\ell}\|_2 \leq 32\sqrt{S}\epsilon_{\mathrm{unif}}^\ell \text{ and}$$

$$\|(I - M_{\ell,h-i+1})P_{h-i}\boldsymbol{\pi}_{h-i}\delta_{\ell,h-i}^\pi\|_2 \leq 32\sqrt{S}\epsilon_{\mathrm{unif}}^\ell.$$

By Lemma 11, we can bound

$$\|\prod_{j=h-i+1}^{h} M_{\ell,j+1}P_j\boldsymbol{\pi}_j\|_2 \leq \sqrt{S}.$$

Combining these gives the result.

We now prove the second part of the result. Denote $A_j := M_{\ell,j+1} P_j \boldsymbol{\pi}_j$ and $B_j := P_j \boldsymbol{\pi}_j$. Then

$$\prod_{j=i}^{k} M_{\ell,j+1} P_j \boldsymbol{\pi}_j - \prod_{j=i}^{k} P_j \boldsymbol{\pi}_j = \prod_{j=i}^{k} A_j - \prod_{j=i}^{k} B_j$$

$$= A_k \left( \prod_{j=i}^{k-1} A_j - \prod_{j=i}^{k-1} B_j \right) + (A_k - B_k) \prod_{j=i}^{k-1} B_j$$

$$\vdots$$

$$= \sum_{s=i}^{k} \left( \prod_{j=s+1}^{k} A_j \right) (A_s - B_s) \left( \prod_{j'=i}^{s-1} B_{j'} \right).$$

By Lemma 11 we have $\| \prod_{j=s+1}^{k} A_j \|_2 \leq \sqrt{S}$. Furthermore, note that $\prod_{j'=i}^{s-1} B_{j'} w_i^\pi = w_s^\pi$. So it follows that

$$\left\| \left( \prod_{j=i}^{k} M_{\ell,j+1} P_j \boldsymbol{\pi}_j - \prod_{j=i}^{k} P_j \boldsymbol{\pi}_j \right) w_i^\pi \right\|_2 \leq \sum_{s=i}^{k} \sqrt{S} \| (A_s - B_s) w_s^\pi \|_2.$$

By the same argument as above, we can bound $\| (A_s - B_s) w_s^\pi \|_2 \leq 32 \sqrt{S} \epsilon_{\text{unif}}^\ell$. $\qquad\square$

**Lemma 18.** *On the event $\mathcal{E}_{\text{prune}}^\ell \cap (\cap_{h' \leq h} \mathcal{E}_{\text{est}}^{\ell,h'}) \cap (\cap_{h' \leq h} \mathcal{E}_{\text{exp}}^{\ell,h'})$, we have, for all $\pi \in \Pi_\ell$:*

$$\| \widehat{\delta}_{\ell,h+1}^\pi - \delta_{\ell,h+1}^\pi \|_2 \leq \sqrt{SH\beta_\ell \epsilon_{\text{exp}}^\ell} + SH(\sqrt{8\epsilon_\ell^{5/3}} + 32\epsilon_{\text{unif}}^\ell).$$

*Proof.* We can write

$$\| \widehat{\delta}_{\ell,h+1}^\pi - \delta_{\ell,h+1}^\pi \|_2 \leq \| \widehat{\delta}_{\ell,h+1}^\pi - \widetilde{\delta}_{\ell,h+1}^\pi \|_2 + \| \widetilde{\delta}_{\ell,h+1}^\pi - \delta_{\ell,h+1}^\pi \|_2.$$

From Lemma 12 we have

$$\widetilde{\delta}_{\ell,h+1}^\pi - \widehat{\delta}_{\ell,h+1}^\pi$$

$$= \sum_{i=0}^{h-2} \left( \prod_{j=h-i+1}^{h} M_{\ell,j+1} P_j \boldsymbol{\pi}_j \right) (P_{h-i} - \widehat{P}_{\ell,h-i}) M_{\ell,h-i} \left[ (\boldsymbol{\pi}_{h-i} - \bar{\boldsymbol{\pi}}_{\ell,h-i}) \widehat{w}_{\ell,h-i}^{\bar{\pi}} + \boldsymbol{\pi}_{h-i} \widehat{\delta}_{\ell,h-i}^\pi \right].$$

From Event (2) of $\mathcal{E}_{\text{est}}^{\ell,h}$ in Lemma 15, we have that for all canonical vectors $e_s$ and $\pi \in \Pi_\ell$:

$$\left\langle e_s, \left( \prod_{j=h-i+1}^{h} M_{\ell,j+1} P_j \boldsymbol{\pi}_j \right) (P_{h-i} - \widehat{P}_{\ell,h-i}) M_{\ell,h-i} \left[ (\boldsymbol{\pi}_{h-i} - \bar{\boldsymbol{\pi}}_{\ell,h-i}) \widehat{w}_{\ell,h-i}^{\bar{\pi}} + \boldsymbol{\pi}_{h-i} \widehat{\delta}_{\ell,h-i}^\pi \right] \right\rangle$$

$$\leq \beta_\ell \sqrt{\sum_{s,a} \frac{[M_{\ell,h-i}((\boldsymbol{\pi}_{h-i} - \bar{\boldsymbol{\pi}}_{\ell,h-i}) \widehat{w}_{\ell,h-i}^{\bar{\pi}} + \boldsymbol{\pi}_{h-i} \widehat{\delta}_{\ell,h-i}^\pi)]_{s,a}^2}{N_{\ell,h-i}(s,a)}}.$$

Now, summing over the bound above for all canonical vectors, and applying this for each $i$, it follows that

$$\| \widehat{\delta}_{\ell,h+1}^\pi - \widetilde{\delta}_{\ell,h+1}^\pi \|_2^2 \leq S\beta_\ell^2 \sum_{h'=1}^{h} \sum_{s,a} \frac{[M_{\ell,h'}((\boldsymbol{\pi}_{h'} - \bar{\boldsymbol{\pi}}_{\ell,h'}) \widehat{w}_{\ell,h'}^{\bar{\pi}} + \boldsymbol{\pi}_{h'} \widehat{\delta}_{\ell,h'}^\pi)]_{s,a}^2}{N_{\ell,h'}(s,a)} \leq SH\beta_\ell \epsilon_{\text{exp}}^\ell$$

where the last inequality holds on $\cap_{h' \leq h} \mathcal{E}_{\text{exp}}^{\ell,h'}$.

We now turn to bounding $\|\widetilde{\delta}^{\pi}_{\ell,h+1} - \delta^{\pi}_{\ell,h+1}\|_2$. By Lemma 17 we have

$$\delta^{\pi}_{\ell,h+1} - \widetilde{\delta}^{\pi}_{\ell,h+1}$$
$$= \sum_{i=0}^{h-2}\left(\prod_{j=h-i+1}^{h} M_{\ell,j+1}P_j\boldsymbol{\pi}_j\right)M_{\ell,h-i+1}P_{h-i}(\boldsymbol{\pi}_{h-i} - \bar{\boldsymbol{\pi}}_{h-i})(w^{\bar{\pi}_\ell}_{\ell,h-i} - \widehat{w}^{\bar{\pi}_\ell}_{\ell,h-i}) + \Delta^{\pi}_{\ell,h+1}$$

for some $\Delta^{\pi}_{\ell,h} \in \mathbb{R}^S$ with $\|\Delta^{\pi}_{\ell,h}\|_2 \leq 32SH\epsilon^\ell_{\text{unif}}$. Furthermore, on $\mathcal{E}^{\ell,h-i}_{\text{est}}$, by Lemma 19 we can bound

$$\|w^{\bar{\pi}_\ell}_{\ell,h-i} - \widehat{w}^{\bar{\pi}_\ell}_{\ell,h-i}\|_2 \leq \sqrt{8S\epsilon^{5/3}_\ell}.$$

Combining this with Lemma 11 gives the result. $\qquad\square$

**Lemma 19.** *On event $\bar{\mathcal{E}}^\ell_{\text{est}}$ we have:*

$$\|\widehat{w}^{\bar{\pi}}_{\ell,h} - w^{\bar{\pi}}_{\ell,h}\|^2_2 \leq 8S\epsilon^{5/3}_\ell.$$

*Proof.* From Event (2) of Lemma 16, we have that for all canonical vectors $e_i \in \mathbb{R}^S$:

$$|\langle e_i, \widehat{w}^{\bar{\pi}}_{\ell,h} - w^{\bar{\pi}}_{\ell,h}\rangle| \leq \sqrt{\frac{2\log\left(\frac{30H\ell^2S}{\delta}\right)}{\bar{n}_\ell}} + \frac{2\log\left(\frac{30H\ell^2S}{\delta}\right)}{\bar{n}_\ell}.$$

Then, combining these bounds together for all $s$:

$$\|\widehat{w}^{\bar{\pi}}_{\ell,h} - w^{\bar{\pi}}_{\ell,h}\|^2_2 \leq \frac{4S\log\left(\frac{30H\ell^2S}{\delta}\right)}{\bar{n}_\ell} + \frac{4S\log^2\left(\frac{30H\ell^2S}{\delta}\right)}{\bar{n}^2_\ell} \leq 4S\epsilon^{5/3}_\ell + 4S\epsilon^{10/3}_\ell \leq 8S\epsilon^{5/3}_\ell,$$

where the last inequality follows from our choice of $\bar{n}_\ell$ in Algorithm 2. $\qquad\square$

**Lemma 20.** *Let $\mathcal{E}_{\text{good}} := (\cap^\infty_{\ell=1}\mathcal{E}^\ell_{\text{prune}}) \cap (\cap^\infty_{\ell=1}\bar{\mathcal{E}}^\ell_{\text{est}}) \cap (\cap^\infty_{\ell=1}\cap_{h\in[H]}\mathcal{E}^{\ell,h}_{\text{est}}) \cap (\cap^\infty_{\ell=1}\cap_{h\in[H]}\mathcal{E}^{\ell,h}_{\text{exp}}).$ Then $\mathbb{P}[\mathcal{E}_{\text{good}}] \geq 1 - 2\delta.$*

*Proof.* By a union bound and basic set manipulations, we have:

$$\mathbb{P}[\mathcal{E}^c_{\text{good}}] \leq \sum_{\ell=1}^{\infty}\mathbb{P}[(\mathcal{E}^\ell_{\text{prune}})^c] + \sum_{\ell=1}^{\infty}\mathbb{P}[(\bar{\mathcal{E}}^\ell_{\text{est}})^c]$$
$$+ \sum_{\ell=1}^{\infty}\sum_{h=1}^{H}\mathbb{P}[(\mathcal{E}^{\ell,h}_{\text{exp}})^c \cap \mathcal{E}^\ell_{\text{prune}} \cap \bar{\mathcal{E}}^\ell_{\text{est}} \cap (\cap_{h'\leq h-1}\mathcal{E}^{\ell,h'}_{\text{est}}) \cap (\cap_{h'\leq h-1}\mathcal{E}^{\ell,h'}_{\text{exp}})]$$
$$+ \sum_{\ell=1}^{\infty}\sum_{h=1}^{H}\mathbb{P}[(\mathcal{E}^{\ell,h}_{\text{est}})^c \cap \mathcal{E}^\ell_{\text{prune}} \cap (\cap_{h'\leq h}\mathcal{E}^{\ell,h}_{\text{exp}})].$$

By Lemma 13, we have $\mathbb{P}[(\mathcal{E}^\ell_{\text{prune}})^c] \leq \delta/3\ell^2$. By By Lemma 16, we have $\mathbb{P}[(\bar{\mathcal{E}}^\ell_{\text{est}})^c] \leq \frac{\delta}{15\ell^2}$. By Lemma 14, we have $\mathbb{P}[(\mathcal{E}^{\ell,h}_{\text{exp}})^c \cap \mathcal{E}^\ell_{\text{prune}} \cap \bar{\mathcal{E}}^\ell_{\text{est}} \cap (\cap_{h'\leq h-1}\mathcal{E}^{\ell,h'}_{\text{est}}) \cap (\cap_{h'\leq h-1}\mathcal{E}^{\ell,h'}_{\text{exp}})] \leq \frac{\delta}{6H\ell^2}$. By Lemma 15 we have $\mathbb{P}[(\mathcal{E}^{\ell,h}_{\text{est}})^c \cap \mathcal{E}^\ell_{\text{prune}} \cap (\cap_{h'\leq h}\mathcal{E}^{\ell,h}_{\text{exp}})] \leq \frac{\delta}{6H\ell^2}$. Putting this together we can bound the above as

$$\leq \sum_{\ell=1}^{\infty}(\frac{\delta}{3\ell^2} + \frac{\delta}{15\ell^2}) + \sum_{\ell=1}^{\infty}\sum_{h=1}^{H}\frac{2\delta}{6H\ell^2} \leq 2\delta.$$

$\qquad\square$

## C.4 Estimation of Reference Policy and Values

**Lemma 21.** *On $\mathcal{E}_{\text{good}}$ we have that:*

$$\left|\sum_{h=1}^{H}\langle \widetilde{r}_{\ell,h}, \boldsymbol{\pi}_h(\widetilde{\delta}_{\ell,h}^{\pi} - \widehat{\delta}_{\ell,h}^{\pi})\rangle\right| \leq \epsilon_\ell \quad and \quad \sum_{h=1}^{H}|\langle\widehat{r}_{\ell,h}-\widetilde{r}_{\ell,h}, \boldsymbol{\pi}_h\widehat{\delta}_{\ell,h}^{\pi}+(\boldsymbol{\pi}_h-\bar{\boldsymbol{\pi}}_{\ell,h})\widehat{w}_{\ell,h}^{\bar{\pi}}\rangle| \leq \epsilon_\ell. \quad \text{(C.5)}$$

*Proof.* From Lemma 12 we have:

$$\widetilde{\delta}_{\ell,h+1}^{\pi} - \widehat{\delta}_{\ell,h+1}^{\pi}$$
$$= \sum_{i=0}^{h-2}\left(\prod_{j=h-i+1}^{h}M_{\ell,j+1}P_j\boldsymbol{\pi}_j\right)(P_{h-i} - \widehat{P}_{\ell,h-i})M_{\ell,h-i}\left[(\boldsymbol{\pi}_{h-i} - \bar{\boldsymbol{\pi}}_{\ell,h-i})\widehat{w}_{\ell,h-i}^{\bar{\pi}} + \boldsymbol{\pi}_{h-i}\widehat{\delta}_{\ell,h-i}^{\pi}\right].$$

A sufficient condition for (C.5) is that, for each $i$:

$$\left|\left\langle \boldsymbol{\pi}_h^{\top}\widetilde{r}_{\ell,h}, \left(\prod_{j=h-i+1}^{h}M_{\ell,j+1}P_j\boldsymbol{\pi}_j\right)(P_{h-i} - \widehat{P}_{\ell,h-i})\right.\right.$$
$$\left.\left. M_{\ell,h-i}\left[(\boldsymbol{\pi}_{h-i} - \bar{\boldsymbol{\pi}}_{\ell,h-i})\widehat{w}_{\ell,h-i}^{\bar{\pi}} + \boldsymbol{\pi}_{h-i}\widehat{\delta}_{\ell,h-i}^{\pi}\right]\right\rangle\right| \leq \epsilon_\ell.$$

On $\mathcal{E}_{\text{good}}$, and in particular $\mathcal{E}_{\text{est}}^{\ell,h}$ (Lemma 15), we can bound the left-hand side of this as:

$$\leq \beta_\ell\sqrt{\sum_{s,a}\frac{\left[M_{\ell,h-i}\left((\boldsymbol{\pi}_{h-i} - \bar{\boldsymbol{\pi}}_{\ell,h-i})\widehat{w}_{\ell,h-i}^{\bar{\pi}} + \boldsymbol{\pi}_{h-i}\widehat{\delta}_{\ell,h-i}^{\pi}\right)\right]_{(s,a)}^2}{N_{\ell,h-i}(s,a)}}$$
$$\leq \beta_\ell\sqrt{\epsilon_\ell^2/H^4\beta_\ell^2}$$
$$\leq \epsilon_\ell/H^2$$

where the second inequality holds on $\mathcal{E}_{\text{good}}$ (in particular $\mathcal{E}_{\text{exp}}^{\ell,h-i}$). This proves the first inequality.

On $\mathcal{E}_{\text{est}}^{\ell,h}$ we can also bound

$$|\langle\widehat{r}_{\ell,h} - \widetilde{r}_{\ell,h}, \boldsymbol{\pi}_h\widehat{\delta}_{\ell,h}^{\pi} + (\boldsymbol{\pi}_h - \bar{\boldsymbol{\pi}}_{\ell,h})\widehat{w}_{\ell,h}^{\bar{\pi}}\rangle|$$
$$\leq \beta_\ell\sqrt{\sum_{s,a}\frac{\left[M_{\ell,h}\left((\boldsymbol{\pi}_h - \bar{\boldsymbol{\pi}}_{\ell,h'})\widehat{w}_{\ell,h}^{\bar{\pi}} + \boldsymbol{\pi}_h\widehat{\delta}_{\ell,h}^{\pi}\right)\right]_{(s,a)}^2}{N_{\ell,h}(s,a)}}$$
$$\leq \epsilon_\ell/H^2.$$

This proves the second inequality.

$\square$

**Lemma 22.** *On event $\mathcal{E}_{\text{good}}$, for any timestep $h$, policies $\pi, \pi'$, and action $a$, we have:*

$$\mathbb{E}_{\pi'}[|\widehat{Q}_{\ell,h}^{\pi}(s_h, a) - Q_h^{\pi}(s_h, a)|] \leq H^2S^{3/2}\sqrt{A\log\frac{24S^2AH\ell^2}{\delta}} \cdot \epsilon_\ell^{1/3} + 64H^2S\epsilon_{\text{unif}}^{\ell}. \quad \text{(C.6)}$$

*Proof.* By Lemma E.15 of [10], we have that:

$$\widehat{Q}_{\ell,h}^{\pi}(s, a) - Q_h^{\pi}(s, a)$$
$$= \mathbb{E}_{\pi}\left[\sum_{h'=h}^{H}\sum_{s'}(\widehat{P}_{\ell,h'}(s' \mid s_{h'}, a_{h'}) - P_h(s' \mid s_{h'}, a_{h'}))\widehat{V}_{\ell,h'+1}^{\pi}(s_{h'}) \mid s_h = s, a_h = a\right].$$

On $\mathcal{E}_{\text{good}}$, in particular $\mathcal{E}_{\text{est}}^{\ell,h'}$, we can bound, for $s \in \mathcal{S}_{\ell,h'}^{\text{keep}}$ and any $a$:

$$\left| \sum_{s'} (\widehat{P}_{\ell,h'}(s' \mid s, a) - P_h(s' \mid s, a)) \widehat{V}_{\ell,h'+1}^{\pi}(s') \right|$$

$$\leq SH\sqrt{\frac{\log \frac{24S^2 AH\ell^2}{\delta}}{N_{\ell,h'}(s,a)}} \leq SH\sqrt{\frac{SA\log \frac{24S^2 AH\ell^2}{\delta}}{K_{\text{unif}}^{\ell} \epsilon_{\text{unif}}^{\ell}}}$$

and where the last inequality follows on $\mathcal{E}_{\text{exp}}^{\ell,h'}$. By our choice of $K_{\text{unif}}^{\ell}$ and $\epsilon_{\text{unif}}^{\ell}$, we can further bound this as

$$\leq SH\sqrt{SA\log \frac{24S^2 AH\ell^2}{\delta}} \cdot \epsilon_{\ell}^{1/3}.$$

For $s \notin \mathcal{S}_{\ell,h'}^{\text{keep}}$, we can bound $|\sum_{s'}(\widehat{P}_{\ell,h'}(s' \mid s, a) - P_h(s' \mid s, a))\widehat{V}_{\ell,h'}^{\pi}(s_{h'})| \leq 2H$. We therefore have that

$$\mathbb{E}_{\pi'}[|\widehat{Q}_{\ell,h}^{\pi}(s_h, a) - Q_h^{\pi}(s_h, a)|]$$

$$\leq \mathbb{E}_{\pi'}\left[ \mathbb{E}_{\pi}\left[ \sum_{h'=h}^{H} SH\sqrt{SA\log \frac{24S^2 AH\ell^2}{\delta}} \cdot \epsilon_{\ell}^{1/3} \cdot \mathbb{I}\{s_{h'} \in \mathcal{S}_{\ell,h'}^{\text{keep}}\} \right.\right.$$

$$\left.\left. + 2H\mathbb{I}\{s_{h'} \notin \mathcal{S}_{\ell,h'}^{\text{keep}}\} \mid s_h = s, a_h = a \right]\right]$$

$$= \sum_{h'=h}^{H} \mathbb{E}_{\widetilde{\pi}}\left[ SH\sqrt{SA\log \frac{24S^2 AH\ell^2}{\delta}} \cdot \epsilon_{\ell}^{1/3} \cdot \mathbb{I}\{s_{h'} \in \mathcal{S}_{\ell,h'}^{\text{keep}}\} + 2H\mathbb{I}\{s_{h'} \notin \mathcal{S}_{\ell,h'}^{\text{keep}}\} \right]$$

$$\leq H^2 S^{3/2}\sqrt{A\log \frac{24S^2 AH\ell^2}{\delta}} \cdot \epsilon_{\ell}^{1/3} + 64H^2 S\epsilon_{\text{unif}}^{\ell},$$

where the last inequality follows by definition of $\mathcal{S}_{\ell,h'}^{\text{keep}}$, and $\pi'$ is the policy which plays $\bar{\pi}_{\ell}$ for the first $h$ steps and then plays $\pi$. This proves the result. $\qquad\square$

**Lemma 23.** *On event $\mathcal{E}_{\text{good}}$, for all $h$ and any $\pi$ and $\pi'$, we have that*

$$|\widehat{U}_{\ell,h}(\pi, \pi') - U_h(\pi, \pi')| \leq 9H^3 S^{3/2}\sqrt{A\log \frac{24S^2 AH\ell^2}{\delta}} \cdot \epsilon_{\ell}^{1/3} + 576H^3 S\epsilon_{\text{unif}}^{\ell}.$$

*Proof.* We have

$$\widehat{U}_{\ell,h}(\pi, \pi') = \mathbb{E}_{\pi',\ell}\left[ \left( \widehat{Q}_{\ell,h}^{\pi}(s_h, \pi_h(s_h)) - \widehat{Q}_{\ell,h}^{\pi}(s_h, \pi_h'(s_h)) \right)^2 \right]$$

where $\mathbb{E}_{\pi',\ell}$ denotes the expectation induced playing policy $\pi'$ on the MDP with transition $\widehat{P}_{\ell}$. We can think of this as simply a value function for policy $\pi$ on the reward $\check{r}_h(s, a) = \left( \widehat{Q}_{\ell,h}^{\pi}(s, \pi_h(s)) - \widehat{Q}_{\ell,h}^{\pi}(s, a) \right)^2$. Let $\check{V}$ denote the value function on this reward on $\widehat{P}_{\ell}$, and note that $\check{V}_h(s) \in [0, H^2]$ for all $(s, h)$. By Lemma E.15 of [10], we then have that

$$\left| \widehat{U}_{\ell,h}(\pi, \pi') - \mathbb{E}_{\pi'}\left[ \left( \widehat{Q}_{\ell,h}^{\pi}(s_h, \pi_h(s_h)) - \widehat{Q}_{\ell,h}^{\pi}(s_h, \pi_h'(s_h)) \right)^2 \right] \right|$$

$$= \mathbb{E}_{\pi'}\left[ \sum_{h=1}^{H} \sum_{s'} (\widehat{P}_{\ell,h}(s' \mid s_h, a_h) - P_h(s' \mid s_h, a_h)) \check{V}_{h+1}(s') \right]$$

$$\leq H^2 \sum_{h=1}^{H} \mathbb{E}_{\pi'}\left[ \sum_{s'} |\widehat{P}_{\ell,h}(s' \mid s_h, a_h) - P_h(s' \mid s_h, a_h)| \right].$$

Note that we always have $\sum_{s'} |\widehat{P}_{\ell,h}(s' \mid s_h, a_h) - P_h(s' \mid s_h, a_h)| \le 2$. Furthermore, on $\mathcal{E}_{\text{good}}$ we also have $\sum_{s'} |\widehat{P}_{\ell,h}(s' \mid s_h, a_h) - P_h(s' \mid s_h, a_h)| \le S\sqrt{\frac{\log \frac{24S^2 AH\ell^2}{\delta}}{N_{\ell,h}(s_h, a_h)}}$. We can therefore bound the above as

$$
\le H^2 \sum_{h=1}^{H} \mathbb{E}_{\pi'} \left[ \min\left\{ 2, S\sqrt{\frac{\log \frac{24S^2 AH\ell^2}{\delta}}{N_{\ell,h}(s_h, a_h)}} \right\} \right]
$$

$$
\le H^2 \sum_{h=1}^{H} \mathbb{E}_{\pi'} \left[ 2 \cdot \mathbb{I}\{s_h \notin \mathcal{S}_{\ell,h}^{\text{keep}}\} + S\sqrt{\frac{\log \frac{24S^2 AH\ell^2}{\delta}}{N_{\ell,h}(s_h, a_h)}} \cdot \mathbb{I}\{s_h \in \mathcal{S}_{\ell,h}^{\text{keep}}\} \right].
$$

For $s \in \mathcal{S}_{\ell,h}^{\text{keep}}$, on $\mathcal{E}_{\text{good}}$ we have $N_{\ell,h}(s_h, a_h) \ge \frac{K_{\text{unif}}^{\ell} \epsilon_{\text{unif}}^{\ell}}{SA} = \epsilon_{\ell}^{2/3}/SA$, and we also have for $s_h \notin \mathcal{S}_{\ell,h}^{\text{keep}}$ that $W_h^{\star}(s) \le 32\epsilon_{\text{unif}}^{\ell}$. Putting this together we can bound the above as

$$
\le H^2 \sum_{h=1}^{H} \left[ 64S\epsilon_{\text{unif}}^{\ell} + S\sqrt{SA \log \frac{24S^2 AH\ell^2}{\delta}} \cdot \epsilon_{\ell}^{1/3} \right]
$$

$$
\le 64SH^3 \epsilon_{\text{unif}}^{\ell} + H^3 S^{3/2} \sqrt{A \log \frac{24S^2 AH\ell^2}{\delta}} \cdot \epsilon_{\ell}^{1/3}.
$$

Furthermore,

$$
\left| \mathbb{E}_{\pi'} \left[ \left( \widehat{Q}_{\ell,h}^{\pi}(s, \pi_h(s)) - \widehat{Q}_{\ell,h}^{\pi}(s, \pi_h'(s)) \right)^2 \right] - \mathbb{E}_{\pi'} \left[ (Q_h^{\pi}(s, \pi_h(s)) - Q_h^{\pi}(s, \pi_h'(s)))^2 \right] \right|
$$

$$
= \left| \mathbb{E}_{\pi'} \left[ \left( \widehat{Q}_{\ell,h}^{\pi}(s, \pi_h(s)) - Q_h^{\pi}(s, \pi_h(s)) + Q_h^{\pi}(s, \pi_h'(s)) - \widehat{Q}_{\ell,h}^{\pi}(s, \pi_h'(s)) \right)^2 \right] \right.
$$

$$
+ \mathbb{E}_{\pi'} \left[ \left( \widehat{Q}_{\ell,h}^{\pi}(s, \pi_h(s)) - Q_h^{\pi}(s, \pi_h(s)) + Q_h^{\pi}(s, \pi_h'(s)) - \widehat{Q}_{\ell,h}^{\pi}(s, \pi_h'(s)) \right) \right.
$$

$$
\left. \left. (Q_h^{\pi}(s, \pi_h(s)) - Q_h^{\pi}(s, \pi_h'(s))) \right] \right|
$$

$$
\le 4H \mathbb{E}_{\pi'}[|\widehat{Q}_{\ell,h}^{\pi}(s, \pi_h(s)) - Q_h^{\pi}(s, \pi_h(s))|] + 4H \mathbb{E}_{\pi'}[|Q_h^{\pi}(s, \pi_h'(s)) - \widehat{Q}_{\ell,h}^{\pi}(s, \pi_h'(s))|]
$$

$$
\le 8H^3 S^{3/2} \sqrt{A \log \frac{24S^2 AH\ell^2}{\delta}} \cdot \epsilon_{\ell}^{1/3} + 512H^3 S\epsilon_{\text{unif}}^{\ell}
$$

where the final inequality follows from Lemma 22. Combining this with the above bound completes the argument. $\qquad\square$

**Lemma 24.** *On event $\mathcal{E}_{\text{good}}$, for all epochs $\ell$, we have that*

$$
\left| \sum_{h=1}^{H} \langle \widetilde{r}_{\ell,h}, \boldsymbol{\pi}_h(\delta_{\ell,h}^{\pi} - \widetilde{\delta}_{\ell,h}^{\pi}) \rangle + \langle \widetilde{r}_{\ell,h}, (\boldsymbol{\pi}_h - \bar{\boldsymbol{\pi}}_{\ell,h})(w_{\ell,h}^{\bar{\pi}} - \widehat{w}_{\ell,h}^{\bar{\pi}}) \rangle \right| \le \epsilon_{\ell}. \qquad (\text{C.7})
$$

*Proof.* We first bound $|\langle M_{\ell,h} r_h, \boldsymbol{\pi}_h(\delta_{\ell,h}^{\pi} - \widetilde{\delta}_{\ell,h}^{\pi}) \rangle|$. By Lemma 17 we have that

$$
\delta_{\ell,h+1}^{\pi} - \widetilde{\delta}_{\ell,h+1}^{\pi}
$$

$$
= \sum_{i=0}^{h-2} \left( \prod_{j=h-i+1}^{h} M_{\ell,j+1} P_j \boldsymbol{\pi}_j \right) M_{\ell,h-i+1} P_{h-i} (\boldsymbol{\pi}_{h-i} - \bar{\boldsymbol{\pi}}_{\ell,h-i})(w_{h-i}^{\bar{\pi}} - \widehat{w}_{\ell,h-i}^{\bar{\pi}}) + \Delta_{\ell,h+1}^{\pi}
$$

for some $\Delta_{\ell,h}^{\pi} \in \mathbb{R}^S$ with $\|\Delta_{\ell,h}^{\pi}\|_2 \leq 32SH\epsilon_{\text{unif}}^{\ell}$. Furthermore, note that

$$
\sum_{h=1}^{H} \sum_{i=0}^{h-2} \left\langle \widetilde{r}_{\ell,h}, \boldsymbol{\pi}_h \left( \prod_{j=h-i+1}^{h} M_{\ell,j+1} P_j \boldsymbol{\pi}_j \right) M_{\ell,h-i+1} P_{h-i} (\boldsymbol{\pi}_{h-i} - \bar{\boldsymbol{\pi}}_{\ell,h-i})(w_{h-i}^{\bar{\pi}} - \widehat{w}_{\ell,h-i}^{\bar{\pi}}) \right\rangle
$$

$$
= \sum_{h=1}^{H} \sum_{k=2}^{h} \left\langle \widetilde{r}_{\ell,h}, \boldsymbol{\pi}_h \left( \prod_{j=k+1}^{h} M_{\ell,j+1} P_j \boldsymbol{\pi}_j \right) M_{\ell,k+1} P_k (\boldsymbol{\pi}_k - \bar{\boldsymbol{\pi}}_{\ell,k})(w_k^{\bar{\pi}} - \widehat{w}_{\ell,k}^{\bar{\pi}}) \right\rangle
$$

$$
= \sum_{k=2}^{H} \sum_{h=k}^{H} \left\langle \widetilde{r}_{\ell,h}, \boldsymbol{\pi}_h \left( \prod_{j=k+1}^{h} M_{\ell,j+1} P_j \boldsymbol{\pi}_j \right) M_{\ell,k+1} P_k (\boldsymbol{\pi}_k - \bar{\boldsymbol{\pi}}_{\ell,k})(w_k^{\bar{\pi}} - \widehat{w}_{\ell,k}^{\bar{\pi}}) \right\rangle
$$

$$
= \sum_{k=2}^{H} \langle P_k^{\top} M_{\ell,k+1} \widetilde{V}_{\ell,k+1}, (\boldsymbol{\pi}_k - \bar{\boldsymbol{\pi}}_{\ell,k})(w_k^{\bar{\pi}} - \widehat{w}_{\ell,k}^{\bar{\pi}}) \rangle.
$$

It follows that

$$
\sum_{h=1}^{H} \langle \widetilde{r}_{\ell,h}, \boldsymbol{\pi}_h (\delta_{\ell,h}^{\pi} - \widetilde{\delta}_{\ell,h}^{\pi}) \rangle + \langle \widetilde{r}_{\ell,h}, (\boldsymbol{\pi}_h - \bar{\boldsymbol{\pi}}_{\ell,h})(w_h^{\bar{\pi}} - \widehat{w}_{\ell,h}^{\bar{\pi}}) \rangle
$$

$$
= \sum_{h=2}^{H} \langle P_h^{\top} M_{\ell,h+1} \widetilde{V}_{\ell,h+1} + \widetilde{r}_{\ell,h}, (\boldsymbol{\pi}_h - \bar{\boldsymbol{\pi}}_{\ell,h})(w_{\ell,h}^{\bar{\pi}} - \widehat{w}_{\ell,h}^{\bar{\pi}}) \rangle + \Delta
$$

for some $\Delta$ satisfying $|\Delta| \leq 32S^{3/2}H^2\epsilon_{\text{unif}}^{\ell}$. On $\mathcal{E}_{\text{good}}$ (specifically $\bar{\mathcal{E}}_{\text{est}}^{\ell}$), we can bound

$$
\sum_{h=2}^{H} |\langle P_h^{\top} M_{\ell,h+1} \widetilde{V}_{\ell,h+1} + \widetilde{r}_{\ell,h}, (\boldsymbol{\pi}_h - \bar{\boldsymbol{\pi}}_{\ell,h})(w_{\ell,h}^{\bar{\pi}} - \widehat{w}_{\ell,h}^{\bar{\pi}}) \rangle|
$$

$$
\leq \sum_{h=2}^{H} \sqrt{\frac{2\mathbb{E}_{s \sim w_{\ell,h}^{\bar{\pi}}}[\langle P_h^{\top} M_{\ell,h+1} \widetilde{V}_{\ell,h+1}^{\pi} + \widetilde{r}_{\ell,h}, (\boldsymbol{\pi}_h - \bar{\boldsymbol{\pi}}_{\ell,h})e_s \rangle^2]}{\bar{n}_{\ell}} \cdot \log \frac{60H\ell^2 |\Pi_{\ell}|}{\delta}}
$$

$$
+ \frac{2H}{3\bar{n}_{\ell}} \log \frac{60H^2\ell^2 |\Pi_{\ell}|}{\delta}
$$

We can also bound

$$
\mathbb{E}_{s \sim w_{\ell,h}^{\bar{\pi}}}[\langle P_h^{\top} M_{\ell,h+1} \widetilde{V}_{\ell,h+1}^{\pi} + \widetilde{r}_{\ell,h}, (\boldsymbol{\pi}_h - \bar{\boldsymbol{\pi}}_{\ell,h})e_s \rangle^2]
$$

$$
\leq 2\mathbb{E}_{s \sim w_{\ell,h}^{\bar{\pi}}}[\langle P_h^{\top} V_{h+1}^{\pi} + r_h, (\boldsymbol{\pi}_h - \bar{\boldsymbol{\pi}}_{\ell,h})e_s \rangle^2] + 2H\mathbb{E}_{s \sim w_{\ell,h}^{\bar{\pi}}}[|[\boldsymbol{\pi}_h^{\top} P_h^{\top} (M_{\ell,h+1} \widetilde{V}_{\ell,h+1}^{\pi} - V_{h+1}^{\pi})]_s|]
$$

$$
+ 2H\mathbb{E}_{s \sim w_{\ell,h}^{\bar{\pi}}}[|[\bar{\boldsymbol{\pi}}_{\ell,h}^{\top} P_h^{\top} (M_{\ell,h+1} \widetilde{V}_{\ell,h+1}^{\pi} - V_{h+1}^{\pi})]_s|] + 4\mathbb{E}_{s \sim w_{\ell,h}^{\bar{\pi}}}[\sup_a |r_h(s,a) - \widetilde{r}_{\ell,h}(s,a)|]
$$

Furthermore,

$$
\mathbb{E}_{s \sim w_{\ell,h}^{\bar{\pi}}}[|[\boldsymbol{\pi}_h^{\top} P_h^{\top} (M_{\ell,h+1} \widetilde{V}_{\ell,h+1}^{\pi} - V_{h+1}^{\pi})]_s|]
$$

$$
= \sum_s |[\boldsymbol{\pi}_h^{\top} P_h^{\top} (M_{\ell,h+1} \widetilde{V}_{\ell,h+1}^{\pi} - V_{h+1}^{\pi})]_s| w_{\ell,h}^{\bar{\pi}}(s)
$$

$$
\leq \sqrt{S} \|(M_{\ell,h+1} \widetilde{V}_{\ell,h+1}^{\pi} - V_{h+1}^{\pi})^{\top} P_h \boldsymbol{\pi}_h w_{\ell,h}^{\bar{\pi}}\|_2
$$

$$
\leq \sqrt{S} \|(\widetilde{V}_{\ell,h+1}^{\pi} - V_{h+1}^{\pi})^{\top} P_h \boldsymbol{\pi}_h w_{\ell,h}^{\bar{\pi}}\|_2 + \sqrt{S} \|(M_{\ell,h+1} \widetilde{V}_{\ell,h+1}^{\pi} - \widetilde{V}_{\ell,h+1}^{\pi})^{\top} P_h \boldsymbol{\pi}_h w_{\ell,h}^{\bar{\pi}}\|_2
$$

$$
\leq 64S^2 H^2 \epsilon_{\text{unif}}^{\ell}
$$

where the last inequality follows from the definition of $\widetilde{V}$ and Lemma 17. A similar bound can be shown for $\mathbb{E}_{s \sim w_{\ell,h}^{\bar{\pi}}}[|[\bar{\boldsymbol{\pi}}_{\ell,h}^{\top} P_h^{\top} (M_{\ell,h+1} \widetilde{V}_{\ell,h+1}^{\pi} - V_{h+1}^{\pi})]_s|]$. In addition, by definition of $\widetilde{r}_{\ell,h}$ we have

$$
\mathbb{E}_{s \sim w_{\ell,h}^{\bar{\pi}}}[\sup_a |r_h(s,a) - \widetilde{r}_{\ell,h}(s,a)|] \leq \mathbb{E}_{s \sim w_{\ell,h}^{\bar{\pi}}}[\mathbb{I}\{s \notin \mathcal{S}_{\ell,h}^{\text{keep}}\}] \leq 32S\epsilon_{\text{unif}}^{\ell}.
$$

Thus, we have

$$\sum_{h=2}^{H} |\langle P_h^\top M_{\ell,h+1} \widetilde{V}_{\ell,h+1} + r_h, (\boldsymbol{\pi}_h - \bar{\boldsymbol{\pi}}_{\ell,h})(w_{\ell,h}^{\bar{\pi}} - \widehat{w}_{\ell,h}^{\bar{\pi}})\rangle|$$

$$\leq \sum_{h=2}^{H} \sqrt{\frac{2\mathbb{E}_{s \sim w_{\ell,h}^{\bar{\pi}}}[\langle P_h^\top M_{\ell,h+1} \widetilde{V}_{\ell,h+1}^{\pi} + r_h, (\boldsymbol{\pi}_h - \bar{\boldsymbol{\pi}}_{\ell,h}) e_s \rangle^2]}{\bar{n}_\ell} \cdot \log \frac{60 H \ell^2 |\Pi_\ell|}{\delta}}$$

$$+ \frac{2H}{3\bar{n}_\ell} \log \frac{60 H^2 \ell^2 |\Pi_\ell|}{\delta}$$

$$\leq \sum_{h=2}^{H} \sqrt{\frac{4 U_h(\pi, \bar{\pi}_\ell) + 384 S^2 H^3 \epsilon_{\mathrm{unif}}^\ell}{\bar{n}_\ell} \cdot \log \frac{60 H \ell^2 |\Pi_\ell|}{\delta}} + \frac{2H}{3\bar{n}_\ell} \log \frac{60 H^2 \ell^2 |\Pi_\ell|}{\delta}.$$

By Lemma 23 and Jensen's inequality, this can be further bounded as

$$\leq \sum_{h=2}^{H} c \sqrt{\frac{\widehat{U}_{\ell-1,h}(\pi, \bar{\pi}_\ell) + S^{3/2} H^3 \sqrt{A \log \frac{24 S^2 A H \ell^2}{\delta}} \cdot \epsilon_\ell^{1/3} + S^2 H^3 \epsilon_{\mathrm{unif}}^\ell}{\bar{n}_\ell} \cdot \log \frac{60 H \ell^2 |\Pi_\ell|}{\delta}}$$

$$+ \frac{2H}{3\bar{n}_\ell} \log \frac{60 H^2 \ell^2 |\Pi_\ell|}{\delta}$$

$$\leq c \sqrt{\frac{H \widehat{U}_{\ell-1}(\pi, \bar{\pi}_\ell) + S^{3/2} H^4 \sqrt{A \log \frac{24 S^2 A H \ell^2}{\delta}} \cdot \epsilon_\ell^{1/3} + S^2 H^4 \epsilon_{\mathrm{unif}}^\ell}{\bar{n}_\ell} \cdot \log \frac{60 H \ell^2 |\Pi_\ell|}{\delta}}$$

$$+ \frac{2H}{3\bar{n}_\ell} \log \frac{60 H^2 \ell^2 |\Pi_\ell|}{\delta}.$$

The result then follows from this, our choice of $\bar{n}_\ell$ and $\epsilon_{\mathrm{unif}}^\ell$, and the bound on $\Delta$ above. $\qquad\square$

**Lemma 25.** *On $\mathcal{E}_{\mathrm{good}}$, we can bound*

$$\frac{\inf_{\pi_{\mathrm{exp}}} \max_{\pi \in \Pi_\ell} \|M_{\ell,h}((\boldsymbol{\pi}_h - \bar{\boldsymbol{\pi}}_{\ell,h}) \widehat{w}_{\ell,h}^{\bar{\pi}} + \boldsymbol{\pi}_h \widehat{\delta}_{\ell,h}^{\pi})\|_{\Lambda_h(\pi_{\mathrm{exp}})^{-1}}^2}{\epsilon_{\mathrm{exp}}^\ell}$$

$$\leq \frac{\inf_{\pi_{\mathrm{exp}}} \max_{\pi \in \Pi_\ell} 4\|\bar{\boldsymbol{\pi}}_{\ell,h} w_{\ell,h}^{\bar{\pi}} - \boldsymbol{\pi}_h w_h^{\pi}\|_{\Lambda_h(\pi_{\mathrm{exp}})^{-1}}^2}{\epsilon_{\mathrm{exp}}^\ell}$$

$$+ \frac{(8 S^2 A + 32 S^3 A H^2) \epsilon_\ell^{5/3} + 2 S^2 A H \beta_\ell \epsilon_{\mathrm{exp}}^\ell + 4096 S^3 A H^2 (\epsilon_{\mathrm{unif}}^\ell)^2}{\epsilon_{\mathrm{unif}}^\ell \epsilon_{\mathrm{exp}}^\ell}.$$

*Proof.* We can bound:

$$\inf_{\pi_{\mathrm{exp}}} \max_{\pi \in \Pi_\ell} \|M_{\ell,h}((\boldsymbol{\pi}_h - \bar{\boldsymbol{\pi}}_{\ell,h}) \widehat{w}_{\ell,h}^{\bar{\pi}} + \boldsymbol{\pi}_h \widehat{\delta}_{\ell,h}^{\pi})\|_{\Lambda_h(\pi_{\mathrm{exp}})^{-1}}^2$$

$$\leq \inf_{\pi_{\mathrm{exp}}} \max_{\pi \in \Pi_\ell} 4\|M_{\ell,h}((\boldsymbol{\pi}_h - \bar{\boldsymbol{\pi}}_{\ell,h}) w_{\ell,h}^{\bar{\pi}} + \boldsymbol{\pi}_h \delta_{\ell,h}^{\pi})\|_{\Lambda_h(\pi_{\mathrm{exp}})^{-1}}^2$$

$$+ \inf_{\pi_{\mathrm{exp}}'} \max_{\pi \in \Pi_\ell} \Big[ 8\|M_{\ell,h}(\boldsymbol{\pi}_h - \bar{\boldsymbol{\pi}}_{\ell,h})(w_{\ell,h}^{\bar{\pi}} - \widehat{w}_{\ell,h}^{\bar{\pi}})\|_{\Lambda_h(\pi_{\mathrm{exp}}')^{-1}}^2$$

$$+ 8\|M_{\ell,h} \boldsymbol{\pi}_h (\delta_{\ell,h}^{\pi} - \widehat{\delta}_{\ell,h}^{\pi})\|_{\Lambda_h(\pi_{\mathrm{exp}}')^{-1}}^2 \Big].$$

We can write

$$\|M_{\ell,h}(\boldsymbol{\pi}_h - \bar{\boldsymbol{\pi}}_{\ell,h})(w^{\bar{\pi}}_{\ell,h} - \widehat{w}^{\bar{\pi}}_{\ell,h})\|^2_{\Lambda_h(\pi'_{\exp})^{-1}}$$

$$= \sum_{s,a} \frac{(\boldsymbol{\pi}_h(a \mid s) - \bar{\boldsymbol{\pi}}_{\ell,h}(a \mid s))^2(w^{\bar{\pi}}_{\ell,h}(s) - \widehat{w}^{\bar{\pi}}_{\ell,h}(s))^2}{[\Lambda_h(\pi'_{\exp})]_{sa,sa}} \cdot \mathbb{I}\{(s,a) \in \mathcal{S}^{\text{keep}}_{\ell,h}\}$$

$$\leq \sum_{s,a} \frac{(w^{\bar{\pi}}_{\ell,h}(s) - \widehat{w}^{\bar{\pi}}_{\ell,h}(s))^2}{[\Lambda_h(\pi'_{\exp})]_{sa,sa}} \cdot \mathbb{I}\{(s,a) \in \mathcal{S}^{\text{keep}}_{\ell,h}\}.$$

On $\mathcal{E}_{\text{good}}$, for each $(s,a) \in \mathcal{S}^{\text{keep}}_{\ell,h}$ we have $W^\star_h(s) \geq \epsilon^\ell_{\text{unif}}$. Let $\pi^{sh}$ denote the policy which achieves $w^{\pi^{sh}}_h(s) = W^\star_h(s)$, and then plays actions uniformly at random at $(s,h)$. Let $\pi'_{\exp} = \text{unif}(\{\pi^{sh}\}_s)$. Then we have $[\Lambda_h(\pi'_{\exp})]_{sa,sa} \geq W^\star_h(s)/SA \geq \epsilon^\ell_{\text{unif}}/SA$ for each $(s,a) \in \mathcal{S}^{\text{keep}}_{\ell,h}$, so we can bound the above as

$$\leq \frac{SA}{\epsilon^\ell_{\text{unif}}} \sum_{s,a} (w^{\bar{\pi}}_{\ell,h}(s) - \widehat{w}^{\bar{\pi}}_{\ell,h}(s))^2 = \frac{SA}{\epsilon^\ell_{\text{unif}}} \|w^{\bar{\pi}}_{\ell,h} - \widehat{w}^{\bar{\pi}}_{\ell,h}\|^2_2 \leq \frac{8S^2 A \epsilon^{5/3}_\ell}{\epsilon^\ell_{\text{unif}}},$$

where the last inequality follows from Lemma 19.

We can obtain a bound on $\|M_{\ell,h}\boldsymbol{\pi}_h(\delta^\pi_{\ell,h} - \widehat{\delta}^\pi_{\ell,h})\|^2_{\Lambda_h(\pi'_{\exp})^{-1}}$ using a similar argument but now applying Lemma 18 to get that:

$$\|M_{\ell,h}\boldsymbol{\pi}_h(\delta^\pi_{\ell,h} - \widehat{\delta}^\pi_{\ell,h})\|^2_{\Lambda_h(\pi'_{\exp})^{-1}} \leq \frac{2S^2 AH \beta_\ell \epsilon^\ell_{\exp}}{\epsilon^\ell_{\text{unif}}} + \frac{32S^3 AH^2 \epsilon^{5/3}_\ell}{\epsilon^\ell_{\text{unif}}} + 4096 S^3 AH^2 \epsilon^\ell_{\text{unif}}.$$

Finally, note that

$$\|M_{\ell,h}((\boldsymbol{\pi}_h - \bar{\boldsymbol{\pi}}_{\ell,h})w^{\bar{\pi}}_{\ell,h} + \boldsymbol{\pi}_h \delta^\pi_{\ell,h})\|^2_{\Lambda_h(\pi_{\exp})^{-1}} = \|M_{\ell,h}(\bar{\boldsymbol{\pi}}_{\ell,h}w^{\bar{\pi}}_{\ell,h} + \boldsymbol{\pi}_h w^\pi_h)\|^2_{\Lambda_h(\pi_{\exp})^{-1}}$$

$$\leq \|\bar{\boldsymbol{\pi}}_{\ell,h}w^{\bar{\pi}}_{\ell,h} - \boldsymbol{\pi}_h w^\pi_h\|^2_{\Lambda_h(\pi_{\exp})^{-1}}$$

where the equality holds by definition, and the inequality by simply manipulations. Combining these bounds gives the result. $\square$

### C.5 Correctness and Sample Complexity

**Lemma 26.** *On the event* $\mathcal{E}_{\text{good}}$, *for all* $\pi \in \Pi_{\ell+1}$, *we have* $V^\star_0(\Pi) - V^\pi_0 \leq 16\epsilon_\ell$, *and* $\pi^\star \in \Pi_\ell$.

*Proof.* Recall $D_{\bar{\pi}_\ell}(\pi) = V^\pi_0 - V^{\bar{\pi}_\ell}_0$. For $\pi \in \Pi_\ell$, we have

$$|\widehat{D}_{\bar{\pi}_\ell}(\pi) - D_{\bar{\pi}_\ell}(\pi)|$$

$$= \left| \sum_{h=1}^H \left[ \langle \widehat{r}_{\ell,h}, \boldsymbol{\pi}_h \widehat{\delta}^\pi_{\ell,h} + (\boldsymbol{\pi}_h - \bar{\boldsymbol{\pi}}_{\ell,h})\widehat{w}^{\bar{\pi}}_{\ell,h} \rangle - \langle r_h, \boldsymbol{\pi}_h \delta^\pi_{\ell,h} + (\boldsymbol{\pi}_h - \bar{\boldsymbol{\pi}}_{\ell,h})w^{\bar{\pi}}_{\ell,h} \rangle \right] \right|$$

$$\leq \underbrace{\sum_{h=1}^H |\langle \widehat{r}_{\ell,h} - \widetilde{r}_{\ell,h}, \boldsymbol{\pi}_h \widehat{\delta}^\pi_{\ell,h} + (\boldsymbol{\pi}_h - \bar{\boldsymbol{\pi}}_{\ell,h})\widehat{w}^{\bar{\pi}}_{\ell,h} \rangle|}_{(a)} + \underbrace{\sum_{h=1}^H |\langle \widetilde{r}_{\ell,h}, \boldsymbol{\pi}_h (\widetilde{\delta}^\pi_{\ell,h} - \widehat{\delta}^\pi_{\ell,h}) \rangle|}_{(b)}$$

$$+ \underbrace{\left| \sum_{h=1}^H \langle \widetilde{r}_{\ell,h}, \boldsymbol{\pi}_h (\delta^\pi_{\ell,h} - \widetilde{\delta}^\pi_{\ell,h}) \rangle + \langle r_h, (\boldsymbol{\pi}_h - \bar{\boldsymbol{\pi}}_{\ell,h})(w^{\bar{\pi}}_{\ell,h} - \widehat{w}^{\bar{\pi}}_{\ell,h}) \rangle \right|}_{(c)}$$

$$+ \underbrace{\sum_{h=1}^H |\langle \widetilde{r}_{\ell,h} - r_h, \boldsymbol{\pi}_h \delta^\pi_{\ell,h} + (\boldsymbol{\pi}_h - \bar{\boldsymbol{\pi}}_{\ell,h})w^{\bar{\pi}}_{\ell,h} \rangle|}_{(d)}.$$

By Lemma 21, on $\mathcal{E}_{\text{good}}$ we have $(a) \leq \epsilon_\ell$ and $(b) \leq \epsilon_\ell$, and by Lemma 24, $(c) \leq \epsilon_\ell$. To bound $(d)$, we note that $\boldsymbol{\pi}_h \delta^\pi_{\ell,h} + (\boldsymbol{\pi}_h - \bar{\boldsymbol{\pi}}_{\ell,h}) w^\pi_{\ell,h} = \boldsymbol{\pi}_h w^\pi_h - \bar{\boldsymbol{\pi}}_{\ell,h} w^{\bar{\pi}}_{\ell,h}$, and so, on $\mathcal{E}_{\text{good}}$ and by definition of $\widetilde{r}_{\ell,h}$,

$$(d) \leq \sum_{h=1}^{H} \sum_{s \notin \mathcal{S}^{\text{keep}}_{\ell,h}} (w^\pi_h(s) + w^{\bar{\pi}}_{\ell,h}(s)) \leq 64 HS \epsilon^\ell_{\text{unif}} \leq \epsilon_\ell.$$

Note that we only eliminate policy $\pi \in \Pi_\ell$ at round $\ell$ if $\max_{\pi'} \widehat{D}_{\bar{\pi}_\ell}(\pi') - \widehat{D}_{\bar{\pi}_\ell}(\pi) > 8\epsilon_\ell$. Assume that $\pi^\star \in \Pi_\ell$. By what we have just shown, if policy $\pi$ is eliminated, we then have

$$8\epsilon_\ell < \max_{\pi' \in \Pi_\ell} D_{\bar{\pi}_\ell}(\pi') - D_{\bar{\pi}_\ell}(\pi) + 8\epsilon_\ell = V^\star_0 - V^\pi_0 + 8\epsilon_\ell \implies V^\pi_0 < V^\star_0.$$

It follows that $\pi^\star$ will not be eliminated at round $\ell$, as long as $\pi^\star \in \Pi_\ell$. By a simple inductive argument, since $\pi^\star \in \Pi_0$, it follows that on $\mathcal{E}_{\text{good}}$, $\pi^\star \in \Pi_\ell$ for all $\ell$.

Furthermore, for each $\pi \in \Pi_{\ell+1}$, we have $\max_{\pi'} \widehat{D}_{\bar{\pi}_\ell}(\pi') - \widehat{D}_{\bar{\pi}_\ell}(\pi) \leq 8\epsilon_\ell$. Which, again by what we have just shown, implies that

$$8\epsilon_\ell \geq \max_{\pi' \in \Pi_\ell} D_{\bar{\pi}_\ell}(\pi') - D_{\bar{\pi}_\ell}(\pi) - 8\epsilon_\ell = V^\star_0 - V^\pi_0 - 8\epsilon_\ell \implies V^\star_0 - V^\pi_0 \leq 16\epsilon_\ell.$$

$\square$

**Theorem 1.** *There exists an algorithm (Algorithm 1) which, with probability at least $1 - 2\delta$, finds an $\epsilon$-optimal policy and terminates after collecting at most*

$$\sum_{h=1}^{H} \inf_{\pi_{\text{exp}}} \max_{\pi \in \Pi} \frac{H^4 \|\phi^\star_h - \phi^\pi_h\|^2_{\Lambda_h(\pi_{\text{exp}})^{-1}}}{\max\{\epsilon^2, \Delta(\pi)^2\}} \cdot \iota\beta^2 + \max_{\pi \in \Pi} \frac{HU(\pi, \pi^\star)}{\max\{\epsilon^2, \Delta(\pi)^2\}} \log \frac{H|\Pi|\iota}{\delta} + \frac{C_{\text{poly}}}{\max\{\epsilon^{\frac{5}{3}}, \Delta^{\frac{5}{3}}_{\min}\}}$$

*episodes, for $C_{\text{poly}} := \text{poly}(S, A, H, \log 1/\delta, \iota, \log |\Pi|), \beta := C\sqrt{\log(\frac{SH|\Pi|}{\delta} \cdot \frac{1}{\Delta_{\min} \vee \epsilon})}$ and $\iota := \log \frac{1}{\Delta_{\min} \vee \epsilon}$.*

*Proof.* First, by Lemma 20, we have that $\mathbb{P}[\mathcal{E}_{\text{good}}] \geq 1 - 2\delta$. For the remainder of the proof we assume we are on $\mathcal{E}_{\text{good}}$.

By Lemma 26, we have that on $\mathcal{E}_{\text{good}}$, for every $\pi \in \Pi_{\ell+1}$, $V^\star_0 - V^\pi_0 \leq 16\epsilon_\ell$, and that $\pi^\star \in \Pi_\ell$ for all $\ell$. It follows that, since we run for $\ell_\epsilon = \lceil \log_2 16/\epsilon \rceil$ epochs, when we terminate each policy $\pi \in \Pi_{\ell_\epsilon}$ satisfies $V^\star_0 - V^\pi_0 \leq 16\epsilon_{\ell_\epsilon} = 16 \cdot 2^{-\ell_\epsilon} \leq \epsilon$. Furthermore, if we terminate early on Line 20, then we know that $|\Pi_{\ell+1}| = 1$, and since $\pi^\star \in \Pi_{\ell+1}$, we have that the algorithm returns $\pi^\star$. Thus, the policy returned by Algorithm 2 is always $\epsilon$-optimal.

It therefore remains to bound the sample complexity of Algorithm 2. At round $\ell$ of Algorithm 2, we collect $\bar{n}_\ell$ samples plus the number of samples collected from OPTCOV. On $\mathcal{E}_{\text{good}}$, we have that the

number of samples collected by OPTCOV at round $\ell$ step $h$ is bounded by

$$C \cdot \frac{\inf_{\pi_{\exp}} \max_{\pi \in \Pi_\ell} \|M_h^\ell((\boldsymbol{\pi}_h - \bar{\boldsymbol{\pi}}_{\ell,h})\widehat{w}_{\ell,h}^{\bar{\pi}} + \boldsymbol{\pi}_h \widehat{\delta}_{\ell,h}^\pi)\|_{\Lambda_h(\pi_{\exp})^{-1}}^2}{\epsilon_{\exp}^\ell}$$

$$+ \frac{C_{\mathrm{fw}}^\ell}{(\epsilon_{\exp}^\ell)^{4/5}} + \frac{C_{\mathrm{fw}}^\ell}{\epsilon_{\mathrm{unif}}^\ell} + \log(C_{\mathrm{fw}}^\ell) \cdot K_{\mathrm{unif}}^\ell$$

$$\overset{(a)}{\leq} C \cdot \frac{\inf_{\pi_{\exp}} \max_{\pi \in \Pi_\ell} \|\bar{\boldsymbol{\pi}}_{\ell,h} w_{\ell,h}^{\bar{\pi}} - \boldsymbol{\pi}_h w_h^\pi\|_{\Lambda_h(\pi_{\exp})^{-1}}^2}{\epsilon_{\exp}^\ell} + \frac{C_{\mathrm{fw}}^\ell}{(\epsilon_{\exp}^\ell)^{4/5}} + \frac{C_{\mathrm{fw}}^\ell}{\epsilon_{\mathrm{unif}}^\ell} + \log(C_{\mathrm{fw}}^\ell) \cdot K_{\mathrm{unif}}^\ell$$

$$+ \frac{(8S^2A + 32S^3AH^2)\epsilon_\ell^{5/3} + 2S^2AH\beta_\ell \epsilon_{\exp}^\ell + 4096S^3AH^2(\epsilon_{\mathrm{unif}}^\ell)^2}{\epsilon_{\mathrm{unif}}^\ell \epsilon_{\exp}^\ell}$$

$$\overset{(b)}{\leq} C \cdot \frac{\inf_{\pi_{\exp}} \max_{\pi \in \Pi_\ell} \|\bar{\boldsymbol{\pi}}_{\ell,h} w_{\ell,h}^{\bar{\pi}} - \boldsymbol{\pi}_h w_h^\pi\|_{\Lambda_h(\pi_{\exp})^{-1}}^2}{\epsilon_\ell^2} \cdot H^4\beta_\ell^2 + \frac{C_{\mathrm{poly}}^\ell}{\epsilon_\ell^{5/3}}$$

$$\overset{(c)}{\leq} C \cdot \frac{\inf_{\pi_{\exp}} \max_{\pi \in \Pi_\ell} \|\boldsymbol{\pi}_h^\star w_h^{\pi^\star} - \boldsymbol{\pi}_h w_h^\pi\|_{\Lambda_h(\pi_{\exp})^{-1}}^2}{\epsilon_\ell^2} \cdot H^4\beta_\ell^2 + \frac{C_{\mathrm{poly}}^\ell}{\epsilon_\ell^{5/3}}$$

$$\overset{(d)}{\leq} C \cdot \inf_{\pi_{\exp}} \max_{\pi \in \Pi} \frac{\|\boldsymbol{\pi}_h^\star w_h^{\pi^\star} - \boldsymbol{\pi}_h w_h^\pi\|_{\Lambda_h(\pi_{\exp})^{-1}}^2}{\max\{\epsilon_\ell^2, \Delta(\pi)^2\}} \cdot H^4\beta_\ell^2 + \frac{C_{\mathrm{poly}}^\ell}{\epsilon_\ell^{5/3}}$$

where the initial bound holds from Lemma 14, the $(a)$ follows from Lemma 25, and $(b)$ follows plugging in our choice of $\epsilon_{\mathrm{unif}}^\ell$ and $\epsilon_{\exp}^\ell$, and with $C_{\mathrm{poly}}^\ell = \mathrm{poly}(S, A, H, \log \ell/\delta, \log 1/\epsilon, \log |\Pi|)$, $(c)$ holds by the triangle inequality and since $\bar{\pi}_\ell \in \Pi_\ell$, and $(d)$ holds because, for all $\pi \in \Pi_\ell$, we have $\Delta(\pi) \leq 32\epsilon_\ell$. Furthermore, we can bound $\bar{n}_\ell$ as

$$\bar{n}_\ell = \min_{\bar{\pi} \in \Pi_\ell} \max_{\pi \in \Pi_\ell} c \cdot \frac{H\widehat{U}_{\ell-1}(\pi, \bar{\pi}) + H^4 S^{3/2}\sqrt{A}\log\frac{SAH\ell^2}{\delta} \cdot \epsilon_\ell^{1/3} + S^2 H^4 \epsilon_{\mathrm{unif}}^\ell}{\epsilon_\ell^2} \cdot \log\frac{60H\ell^2|\Pi_\ell|}{\delta}$$

$$\overset{(a)}{\leq} \min_{\bar{\pi} \in \Pi_\ell} \max_{\pi \in \Pi_\ell} c \cdot \frac{HU(\pi, \bar{\pi}) + H^4 S^{3/2}\sqrt{A}\log\frac{SAH\ell^2}{\delta} \cdot \epsilon_\ell^{1/3} + S^2 H^4 \epsilon_{\mathrm{unif}}^\ell}{\epsilon_\ell^2} \cdot \log\frac{60H\ell^2|\Pi_\ell|}{\delta}$$

$$\overset{(b)}{\leq} \max_{\pi \in \Pi} c \cdot \frac{HU(\pi, \pi^\star)}{\max\{\epsilon_\ell^2, \Delta(\pi)^2\}} \cdot \log\frac{60H\ell^2|\Pi_\ell|}{\delta} + \frac{C_{\mathrm{poly}}^\ell}{\epsilon_\ell^{5/3}}$$

where $(a)$ follows from Lemma 23, and $(b)$ since $\pi^\star \in \Pi_\ell$, and by a similar argument as above.

Thus, if we run for a total of $L$ rounds, the sample complexity is bounded as

$$\sum_{\ell=1}^{L} \left( C \cdot \sum_{h=1}^{H} \inf_{\pi_{\exp}} \max_{\pi \in \Pi} \frac{\|\boldsymbol{\pi}_h^\star w_h^{\pi^\star} - \boldsymbol{\pi}_h w_h^\pi\|_{\Lambda_h(\pi_{\exp})^{-1}}^2}{\max\{\epsilon_\ell^2, \Delta(\pi)^2\}} \cdot H^4\beta_\ell^2 \right.$$

$$\left. + \max_{\pi \in \Pi} c \cdot \frac{HU(\pi, \pi^\star)}{\max\{\epsilon_\ell^2, \Delta(\pi)^2\}} \cdot \log\frac{60H\ell^2|\Pi_\ell|}{\delta} \right) + \frac{LC_{\mathrm{poly}}^L}{\epsilon_L^{5/3}}.$$

By construction, we have that $L \leq \lceil \log_2 16/\epsilon \rceil$. However, we terminate early if $|\Pi_{\ell+1}| = 1$, and since each $\pi \in \Pi_{\ell+1}$ satisfies $\Delta(\pi) \leq \epsilon_\ell$, it follows that we will have $|\Pi_{\ell+1}| = 1$ once $\epsilon_\ell < \Delta_{\min}$, which will occur for $\ell \geq \lceil \log_2 \frac{1}{\Delta_{\min}} \rceil + 1$. Thus, we can bound

$$L \leq \min\{\lceil \log_2 16/\epsilon \rceil, \lceil \log_2 1/\Delta_{\min} \rceil + 1\},$$

and so for all $\epsilon_\ell, \ell \leq L$, we have $\epsilon_\ell \geq c \cdot \max\{\epsilon, \Delta_{\min}\}$. Plugging this into the above gives the final complexity.

$\square$

# D   Tabular Contextual Bandits: Upper Bound

**Setting and notation.** We study stochastic tabular contextual bandits, denoted by the tuple $(\mathcal{C}, \mathcal{A}, \mu^\star, \nu)$. At each episode, a context $c \sim \mu^\star$ arrives, the agent chooses an action $a \in \mathcal{A}$,

and receives reward $r(c, a) \sim \nu(c, a)$ in $\mathbb{R}$. Note that this is a special case of the Tabular MDP when $H = 1$. In this setting, we use the terminology "contexts" instead of "states" to highlight that the agent has no impact on these. The vector $\mu^\star$ plays the same role as the state visitation vectors $w_h^\pi$ previously, except this is now policy-independent. The notation for policy matrix $\boldsymbol{\pi}$, values $V^\pi$, features $\phi^\pi(c, a)$ are inherited directly from the general case.

Define $\theta^\star \in R^{|\mathcal{C}|A}$ as the vector of reward means, so that $[\theta^\star]_{(c,a)} = \mathbb{E}_\nu[r(c, a)]$. Then, we can write the value of $\pi$ as:

$$\mathbb{E}_{\nu, \mu^\star}[r(c, \pi(c))] = \sum_{c,a} \theta^\star_{c,a}[\mu^\star]_c[\pi(c)]_a = (\theta^\star)^\top \boldsymbol{\pi}\mu^\star$$

For any $(\theta, \mu)$ define $\mathsf{OPT}(\theta, \mu) := \arg\max_{\pi \in \Pi} \theta^\top \boldsymbol{\pi}\mu$, where $\theta$ is any hypothetical vector of reward-means and $\mu \in \Delta_{|\mathcal{C}|}$ is a hypothetical context distribution.

Recall that we use $\boldsymbol{\pi} \in \mathbb{R}^{|\mathcal{C}|A \times |\mathcal{C}|}$ to refer to the policy matrix. The vector $\boldsymbol{\pi}\mu \in \mathbb{R}^{|\mathcal{C}|A}$ contains context-action visitations for policy $\pi$ under context distribution $\mu$. Define function $G(\mu, \pi) = \mathbb{E}_{\mu, \pi}[(\boldsymbol{\pi}\mu)(\boldsymbol{\pi}\mu)^\top]$ which returns the expected covariance matrix of policy $\pi$ under context distribution $\mu$. For shorthand, we refer to $\hat{A}(\pi) = G(\hat{\mu}_\ell, \pi_{\exp})$ and $A(\pi) = G(\mu^\star, \pi_{\exp})$ for any $\pi$.

**Lemma 27.** *Define the experimental design objective*

$$F(\pi_{\exp}, \mu, \pi, \pi') = \|(\boldsymbol{\pi}' - \boldsymbol{\pi})\mu\|^2_{G(\mu, \pi_{\exp})^{-1}}.$$

*Then, for any $\mu \in \Delta_\mathcal{C}$,*

$$\min_{\pi_{\exp}} \max_{\pi, \pi' \in \Pi_\ell} F(\pi_{\exp}, \mu, \pi, \pi') = \max_{\pi, \pi' \in \Pi_\ell} \min_{\pi_{\exp}} F(\pi_{\exp}, \mu, \pi, \pi')$$

*Proof.* We can rewrite the maximization problem to be over the simplex $\Delta_{\Pi_\ell \times \Pi_\ell}$ instead:

$$\min_{\pi_{\exp}} \max_{\lambda \in \Delta_{\Pi_\ell \times \Pi_\ell}} \sum_{\pi, \pi' \in \Pi_\ell \times \Pi_\ell} \lambda_{\pi, \pi'} F(\pi_{\exp}, \mu, \pi, \pi') \tag{D.1}$$

This does not change the objective value. To see this, note that for any selection $(\pi_1, \pi_2)$ in the original problem, the same objective value can be obtained by setting $\lambda = e_{\pi_1, \pi_2}$; hence, the modification to the optimization cannot reduce the value. Further if $F(\pi_{\exp}, \mu, \pi, \pi')$ is maximized by $(\pi_1, \pi_2)$, setting $\lambda$ as anything other than $e_{\pi_1, \pi_2}$ cannot increase the objective value.

Now, note that both the minimization and maximization problems are over simplices, which are compact and convex sets. The objective is linear in the maximization variable, and hence concave. The objective can be rewritten as

$$\sum_{c \in \mathcal{S}} \sum_{a \in \mathcal{A}} \frac{(\boldsymbol{\pi} - \boldsymbol{\pi}')^\top e_{a,c} e_{a,c}^\top (\boldsymbol{\pi} - \boldsymbol{\pi}')}{p_{c,a}}.$$

Here, $p_{c,a}$ as the probability that $\pi_{\exp}$ plays action $a$, given that we are in context $c$. From this representation, we can clearly see that the objective is convex in each $p_{c,a}$. Hence, since we are optimizing over finite-dimensional spaces ($|\mathcal{A}|$ and $|\mathcal{C}|$ are finite), Von Neumann's minimax theorem applies and the proof is complete. $\square$

**Lemma 28.** *For the contextual bandit problem, define the experimental design objective*

$$F(\pi_{\exp}, \mu, \pi, \pi') = \|(\boldsymbol{\pi}' - \boldsymbol{\pi})\mu\|^2_{G(\mu, \pi_{\exp})^{-1}}.$$

*Then, for any $\mu$ and assuming that all policies in $\Pi_\ell$ are deterministic, we have:*

$$\min_{\pi_{\exp}} \max_{\pi, \pi' \in \Pi_\ell} F(\pi_{\exp}, \mu, \pi, \pi') = \max_{\pi, \pi' \in \Pi_\ell} \mathbb{E}_{c \sim \mu}[4\mathbb{I}[\pi(c) \neq \pi'(c)]], \tag{D.2}$$

*Proof.* Below, we refer to $p_{c,a}$ as the probability that $\pi_{\mathrm{exp}}$ plays action $a$, given that we are in context $c$. We have:

$$\min_{\pi_{\mathrm{exp}}} \max_{\pi,\pi'\in\Pi_\ell} \|(\boldsymbol{\pi}'-\boldsymbol{\pi})\mu\|^2_{G(\mu,\pi_{\mathrm{exp}})^{-1}}$$

$$= \max_{\pi,\pi'\in\Pi_\ell} \min_{\pi_{\mathrm{exp}}} \|(\boldsymbol{\pi}'-\boldsymbol{\pi})\mu\|^2_{G(\mu,\pi_{\mathrm{exp}})^{-1}}$$

$$= \max_{\pi,\pi'\in\Pi_\ell} \min_{p_1\dots p_{\mathcal{C}}\in\Delta_{\mathcal{A}}} \sum_{a,c} \mu_c^2 \frac{(\boldsymbol{\pi}-\boldsymbol{\pi}')^\top e_{a,c}e_{a,c}^\top(\boldsymbol{\pi}-\boldsymbol{\pi}')}{\mu_c p_{c,a}}$$

$$= \max_{\pi,\pi'\in\Pi_\ell} \sum_c \mu_c \min_{p_c} \sum_{a\in\mathcal{A}} \frac{(\boldsymbol{\pi}-\boldsymbol{\pi}')^\top e_{a,c}e_{a,c}^\top(\boldsymbol{\pi}-\boldsymbol{\pi}')}{p_{c,a}}$$

$$= \max_{\pi,\pi'\in\Pi_\ell} \sum_c \mu_c \left(\sum_{a\in\mathcal{A}} \sqrt{(\boldsymbol{\pi}-\boldsymbol{\pi}')^\top e_{a,c}e_{a,c}^\top(\boldsymbol{\pi}-\boldsymbol{\pi}')}\right)^2.$$

Here the first equality follows from Lemma 27, and the last from Lemma D.6 of [30].

We have assumed that the policies in $\Pi_\ell$ are deterministic. Hence, the only two actions in the summation over $\mathcal{A}$ above that are relevant are $\pi(c)$ and $\pi'(c)$. For all other $a \in \mathcal{A}$, the term in the square root evaluates to 0. If $\pi(c) = \pi'(c)$, then the entire summation over $\mathcal{A}$ evaluates to 0; else, the terms indexed by $\pi(c)$ and $\pi'(c)$ are both 1, and the summation evalutes to 2. Hence, we can simplify the expression to exactly the form of Equation (D.2) from the lemma statement, and the proof is complete. $\square$

**Lemma 29.** *For the contextual bandits problem, we have that*

$$\max_{\pi\in\Pi} \mathbb{E}_{c\sim\mu^\star}[\mathbb{E}_{\nu^\star}[(r(c,\pi(c))-r(c,\pi^\star(c)))^2|c]] \le \inf_{\pi_{\mathrm{exp}}} \max_{\pi\in\Pi} \|\phi^\star-\phi^\pi\|^2_{\Lambda(\pi_{\mathrm{exp}})^{-1}}$$

*Proof.* Observe that $r(c,\pi(c))-r(c,\pi^\star(c))=0$ if $\pi(c)=\pi^\star(c)$; else, $|r(c,\pi(c))-r(c,\pi^\star(c))|\le 2$. Then, it follows that

$$\max_{\pi\in\Pi} \mathbb{E}_{c\sim\mu^\star}[\mathbb{E}_{\nu^\star}[(r(c,\pi(c))-r(c,\pi^\star(c)))^2|c]]$$

$$\le \max_{\pi\in\Pi} 4\mathbb{E}_{c\sim\mu^\star}\mathbb{I}(\pi(c)\ne\pi^\star(c))$$

$$= \inf_{\pi_{\mathrm{exp}}} \max_{\pi\in\Pi} \|\phi^\star-\phi^\pi\|^2_{\Lambda(\pi_{\mathrm{exp}})^{-1}},$$

where the equality follows from Lemma 28. $\square$

Now, we state our main upper bound for contextual bandits.

**Corollary 1.** *For the setting of tabular contextual bandits, there exists an algorithm such that with probability at least $1-2\delta$, as long as $\Pi$ contains only deterministic policies, it finds an $\epsilon$-optimal policy and terminates after collecting at most the following number of samples:*

$$\inf_{\pi_{\mathrm{exp}}} \max_{\pi\in\Pi} \frac{\|\phi^\star-\phi^\pi\|^2_{\Lambda(\pi_{\mathrm{exp}})^{-1}}}{\max\{\epsilon^2,\Delta(\pi)^2\}} \cdot \beta^2 \log\frac{1}{\Delta_{\min}\vee\epsilon} + \frac{C_{\mathrm{poly}}}{\max\{\epsilon^{5/3},\Delta_{\min}^{5/3}\}},$$

*for $C_{\mathrm{poly}} = \mathrm{poly}(|\mathcal{S}|, A, \log 1/\delta, \log 1/(\Delta_{\min}\vee\epsilon), \log|\Pi|)$ and $\beta = C\sqrt{\log(\frac{S|\Pi|}{\delta}\cdot\frac{1}{\Delta_{\min}\vee\epsilon})}$.*

*Proof.* In the special case of contextual bandits, $U(\pi,\pi^\star)$ defined in Theorem 1 can be written more simply as $\mathbb{E}_{c\sim\mu^\star}[\mathbb{E}_{\nu^\star}[(r(c,\pi(c))-r(c,\pi^\star(c)))^2|c]]$. Then, by Lemma 29, we have that:

$$\frac{U(\pi,\pi^\star)}{\max\{\epsilon^2,\Delta(\pi)^2,\Delta_{\min}^2\}} \le \inf_{\pi_{\mathrm{exp}}} \max_{\pi\in\Pi} \frac{\|\phi^\star-\phi^\pi\|^2_{\Lambda(\pi_{\mathrm{exp}})^{-1}}}{\max\{\epsilon^2,\Delta(\pi)^2,\Delta_{\min}^2\}}$$

Plugging this into Theorem 1 completes the proof. $\square$

# E   MDPs with Action-Independent Transitions

We consider here a special class of MDPs where the transitions only depend on the states and are independent of the actions selected i.e all $P_h$ are such that $P_h(s, a) = P_h(s, a')$ for all $(a, a') \in \mathcal{A}$. In this special case, we prove in this subsection that the (leading order) complexity of PERP reduces to $O(\rho_\Pi)$.

**Lemma 30.** *For the ergodic MDP problem,*

$$\min_{\pi_{\exp}} \max_{\pi \in \Pi} \|\phi_h^\pi - \phi_h^\star\|_{\Lambda_h(\pi_{\exp})^{-1}}^2 = \max_{\pi \in \Pi} \min_{\pi_{\exp}} \|\phi_h^\pi - \phi_h^\star\|_{\Lambda_h(\pi_{\exp})^{-1}}^2$$

*Proof.* We can rewrite the maximization problem to be over the simplex $\Delta_\Pi$ instead:

$$\min_{\pi_{\exp}} \max_{\lambda \in \Delta_\Pi} \sum_{\pi \in \Pi} \lambda_\pi \|\phi_h^\pi - \phi_h^\star\|_{\Lambda_h(\pi_{\exp})^{-1}}^2 \tag{E.1}$$

This does not change the objective value. To see this, note that for any selection $\pi \in \Pi$ in the original problem, the same objective value can be obtained by setting $\lambda = e_\pi$ in Equation (E.1); hence, the modification to the optimization cannot reduce the value. Further if $\|\phi_h^\pi - \phi_h^\star\|_{\Lambda_h(\pi_{\exp})^{-1}}^2$ is maximized by $\pi$ for any fixed $\pi_{\exp}$, setting $\lambda$ as anything other than $e_\pi$ cannot increase the objective value.

Now, note that both the minimization and maximization problems are over simplices, which are compact and convex sets. The objective is linear in the maximization variable, and hence concave. The objective can be rewritten as

$$\sum_a \frac{(\boldsymbol{\pi}_h - \boldsymbol{\pi}_h^\star)^\top e_{s,a} e_{s,a}^\top (\boldsymbol{\pi}_h - \boldsymbol{\pi}_h^\star)}{p_{s,a}}$$

Here, $p_{s,a}$ is the probability that $\pi_{\exp}$ plays action $a$, given that it is in context $s$. From this representation, we can clearly see that the objective is convex in each $p_{s,a}$. Hence, Von Neumann's minimax theorem applies and the proof is complete. $\qquad\square$

**Lemma 31.** *For the setting of ergodic MDPs,*

$$\min_{\pi_{\exp}} \max_{\pi \in \Pi} \|\phi_h^\pi - \phi_h^\star\|_{\Lambda_h(\pi_{\exp})^{-1}}^2 = \max_{\pi \in \Pi} 2\mathbb{E}_{s \sim w_h^\star} \mathbb{I}[\pi_h(s) \neq \pi_h'(s)], \tag{E.2}$$

*Proof.* Below, we refer to $p_{s,a}$ as the probability that $\pi_{\exp}$ plays action $a$, given that it is in context $s$. The second equality follows from Lemma 30.

$$\min_{\pi_{\exp}} \max_{\pi \in \Pi} \|\phi_h^\pi - \phi_h^\star\|_{\Lambda_h(\pi_{\exp})^{-1}}^2$$

$$= \min_{\pi_{\exp}} \max_{\pi \in \Pi} \|(\boldsymbol{\pi}_h - \boldsymbol{\pi}_h^\star) w_h^\star\|_{\Lambda_h(\pi_{\exp})^{-1}}^2$$

$$= \max_{\pi \in \Pi} \min_{\pi_{\exp}} \|(\boldsymbol{\pi}_h - \boldsymbol{\pi}_h^\star) w_h^\star\|_{\Lambda_h(\pi_{\exp})^{-1}}^2$$

$$= \max_{\pi \in \Pi} \min_{p_1 \dots p_S \in \Delta_\mathcal{A}} \sum_{s,a} (w_h^\star(s))^2 \frac{(\boldsymbol{\pi}_h - \boldsymbol{\pi}_h^\star)^\top e_{s,a} e_{s,a}^\top (\boldsymbol{\pi}_h - \boldsymbol{\pi}_h^\star)}{w_h^\star(s) p_{s,a}}$$

$$= \max_{\pi \in \Pi} \sum_s w_h^\star(s) \min_{p_s \in \Delta_\mathcal{A}} \sum_a \frac{(\boldsymbol{\pi}_h - \boldsymbol{\pi}_h^\star)^\top e_{s,a} e_{s,a}^\top (\boldsymbol{\pi}_h - \boldsymbol{\pi}_h^\star)}{p_{s,a}}$$

$$= \max_{\pi \in \Pi} \sum_s w_h^\star(s) \left( \sum_a \sqrt{(\boldsymbol{\pi}_h - \boldsymbol{\pi}_h^\star)^\top e_{s,a} e_{s,a}^\top (\boldsymbol{\pi}_h - \boldsymbol{\pi}_h^\star)} \right)^2$$

The optimization problems in the final line were solved using KKT conditions. We assume that the two policies are deterministic. Hence, the only two actions in the summation over $\mathcal{A}$ above that are relevant are $\pi_h(s)$ and $\pi_h'(s)$. For all other $a \in \mathcal{A}$, the term in the square root evaluates to 0. If $\pi_h(s) = \pi_h'(s)$, then the entire summation over $\mathcal{A}$ evaluates to 0; else, the terms indexed by $\pi(c)$ and $\pi'(c)$ are both 1, and the summation evalutes to 2. Hence, we can simplify the expression to exactly the form of Equation (E.2) from the lemma statement, and the proof is complete. $\qquad\square$

**Lemma 32.** *For the ergodic MDP problem, we have that*

$$\max_{\pi \in \Pi} \frac{HU(\pi, \pi^\star)}{\max\{\epsilon^2, \Delta(\pi)^2, \Delta_{\min}^2\}} \leq 2H^4 \sum_{h=1}^{H} \inf_{\pi_{\exp}} \max_{\pi \in \Pi} \frac{\|\phi_h^\star - \phi_h^\pi\|_{\Lambda_h(\pi_{\exp})^{-1}}^2}{\max\{\epsilon^2, \Delta(\pi)^2, \Delta_{\min}^2\}}$$

*Proof.* Recall the definition of $U(\pi, \pi^\star)$

$$U(\pi, \pi^\star) = \sum_{h=1}^{H} \mathbb{E}_{s_h \sim w_h^{\pi^\star}}[(Q_h^\pi(s_h, \pi_h(s)) - Q_h^\pi(s_h, \pi_h^\star(s)))^2].$$

Then, we have that

$$\max_{\pi \in \Pi} \frac{HU(\pi, \pi^\star)}{\max\{\epsilon^2, \Delta(\pi)^2, \Delta_{\min}^2\}}$$

$$= \max_{\pi \in \Pi} \frac{H \sum_{h=1}^{H} \mathbb{E}_{s_h \sim w_h^{\pi^\star}}[(Q_h^\pi(s_h, \pi_h(s)) - Q_h^\pi(s_h, \pi_h^\star(s)))^2]}{\max\{\epsilon^2, \Delta(\pi)^2, \Delta_{\min}^2\}}$$

$$\leq H \sum_{h=1}^{H} \max_{\pi \in \Pi} \frac{\mathbb{E}_{s_h \sim w_h^{\pi^\star}}[(Q_h^\pi(s_h, \pi_h(s)) - Q_h^\pi(s_h, \pi_h^\star(s)))^2]}{\max\{\epsilon^2, \Delta(\pi)^2, \Delta_{\min}^2\}}$$

$$\leq H \sum_{h=1}^{H} \max_{\pi \in \Pi} \frac{2H^2 \mathbb{E}_{s \sim w_h^\star} \mathbb{I}[\pi_h(s) \neq \pi_h'(s)]}{\max\{\epsilon^2, \Delta(\pi)^2, \Delta_{\min}^2\}}$$

$$= H^4 \sum_{h=1}^{H} \inf_{\pi_{\exp}} \max_{\pi \in \Pi} \frac{\|\phi_h^\star - \phi_h^\pi\|_{\Lambda_h(\pi_{\exp})^{-1}}^2}{\max\{\epsilon^2, \Delta(\pi)^2, \Delta_{\min}^2\}}.$$

The final equality follows from Lemma 31. $\qquad\square$

**Corollary 2.** *Assume that all $P_h$ are such that $P_h(s'|s, a) = P_h(s'|s, a')$ for all $(a, a') \in \mathcal{A}$. Then, with probability at least $1 - 2\delta$, PERP (Algorithm 2) finds an $\epsilon$-optimal policy and terminates after collecting at most the following number of episodes:*

$$\sum_{h=1}^{H} \inf_{\pi_{\exp}} \max_{\pi \in \Pi} \frac{\|\phi_h^\star - \phi_h^\pi\|_{\Lambda_h(\pi_{\exp})^{-1}}^2}{\max\{\epsilon^2, \Delta(\pi)^2\}} \cdot \iota H^4 \beta^2 + \frac{C_{\text{poly}}}{\max\{\epsilon^{5/3}, \Delta_{\min}^{5/3}\}}$$

*for $C_{\text{poly}}, \beta$ as defined in Theorem 1.*

*Proof.* The proof follows directly from Theorem 1 and Lemma 32. $\qquad\square$

## F  Tabular Franke Wolfe

**Theorem 2.** *Fix parameters $K_{\text{unif}} > 0$, $\epsilon_{\exp} > 0$, and consider some $\Phi \subseteq \mathbb{R}^{SA}$ and set $\mathcal{S}_0 \subseteq \mathcal{S}$. Let $\epsilon_{\text{unif}} > 0$ be some value satisfying*

$$W_h^\star(s) > \epsilon_{\text{unif}}, \forall s \in \mathcal{S}_0, \quad \text{and} \quad K_{\text{unif}} \geq \epsilon_{\text{unif}}^{-1}.$$

*Assume that $|[\phi]_{(s,a)}| \leq C_\phi \cdot (W_h^\star(s) + \sqrt{\epsilon_\phi})$ for all $s \in \mathcal{S}_0$, $\phi \in \Phi$, and some $C_\phi > 0$, and that $[\phi]_{(s,a)} = 0$ for $s \notin \mathcal{S}_0$. Additionally, let the parameters be such that $\epsilon_\phi/(K_{\text{unif}}\epsilon_{\text{unif}}) \leq \epsilon_{\exp}$. Then with probability at least $1 - \delta$, algorithm Algorithm 3 run with these parameters will collect at most*

$$\min\left\{ C \cdot \frac{\inf_{\Lambda \in \Omega_h} \max_{\phi \in \Phi} \|\phi\|_{\Lambda^{-1}}^2}{\epsilon_{\exp}} + \frac{C_{\text{fw}}}{\epsilon_{\exp}^{4/5}}, C_{\text{fw}}(\frac{1}{\epsilon_{\exp}} + K_{\text{unif}}) \right\} + \frac{C_{\text{fw}}}{\epsilon_{\text{unif}}} + \log(C_{\text{fw}}) \cdot K_{\text{unif}}$$

*episodes, for $C$ a universal constant and $C_{\text{fw}} = \text{poly}(S, A, H, C_\phi, \log 1/\delta, \log 1/\epsilon_{\exp}, \log |\Phi|)$, and will produce covariates $\widehat{\Sigma}$ such that*

$$\max_{\phi \in \Phi} \|\phi\|_{\widehat{\Sigma}^{-1}}^2 \leq \epsilon_{\exp} \tag{F.1}$$

*and, for all $s \in \mathcal{S}_0$,*

$$[\widehat{\Sigma}]_{(s,a)} \geq \frac{\epsilon_{\text{unif}}}{2SA} \cdot K_{\text{unif}}. \tag{F.2}$$

---

**Algorithm 3** Online Experiment Design (OPTCOV)

---

1: **input:** directions $\Phi$, tolerance $\epsilon_{\text{exp}}$, confidence $\delta$, minimum reachability $\epsilon_{\text{unif}}$, minimum exploration $K_{\text{unif}}$, pruned states $\mathcal{S}_0$, step $h$
2: $i \leftarrow 1$
3: **while** $T_i K_i \leq \text{poly}(S, A, H, C_\phi, \log 1/\delta, \log 1/\epsilon_{\text{exp}}, \log |\Phi|) \cdot \epsilon_{\text{exp}}^{-1}$ **do**
4:     $\mathfrak{D}_{\text{unif}}^i \leftarrow \text{UNIFEXP}(\epsilon_{\text{unif}}, K_i T_i + K_{\text{unif}}, \delta/8i^2)$
5:     $\mathbf{\Lambda}_0^i \leftarrow \frac{1}{T_i K_i}\text{diag}(v^i)$ where $[v^i]_{sa} = \sum_{(s',a') \in \mathfrak{D}_{\text{unif}}^i} \mathbb{I}\{(s', a') = (s, a)\}$ for $s \in \mathcal{S}_0$, and $T_i K_i$ otherwise
6:     Run iteration $i$ of Algorithm 4 of [43] on objective

$$f_i(\mathbf{\Lambda}) \leftarrow \frac{1}{\eta_i} \log\left(\sum_{\phi \in \Phi} e^{\eta_i \|\phi\|_{\mathbf{A}(\mathbf{\Lambda})^{-1}}^2}\right) \quad \text{for} \quad \mathbf{A}(\mathbf{\Lambda}) = \mathbf{\Lambda} + \mathbf{\Lambda}_0^i, \eta_i = 2^{2i/5}$$

    to obtain data $\mathfrak{D}^i$
7:     **if** Algorithm 4 reaches termination condition **then**
8:         **return** $\mathfrak{D}^i \cup \mathfrak{D}_{\text{unif}}^i$
9:     **end if**
10:     $i \leftarrow i + 1$
11: **end while**
12: $\mathfrak{D} \leftarrow \text{UNIFEXP}(\epsilon_{\text{unif}}, \frac{8S^2 A^2 C_\phi^2}{\epsilon_{\text{exp}}} + (8S^2 A^2 C_\phi^2 + 1)K_{\text{unif}}, \delta/4)$
13: **return** $\mathfrak{D}$

---

*Proof.* To prove this result, we apply Lemma 37 combined with Lemma 36.

Let $\mathcal{E}_{\text{exp}}^i$ denote the success event of running Algorithm 4 at epoch $i$, as defined in Lemma 36. On this event, and under the assumption that $W_h^\star(s) > \epsilon_{\text{unif}}$ for each $s \in \mathcal{S}_0$, we have that $[\mathbf{\Sigma}_i]_{(s,a)} \geq \frac{W_h^\star(s)}{2SA} \cdot (T_i K_i + K_{\text{unif}})$ for each $(s, a)$ with $s \in \mathcal{S}_0$ and $\mathbf{\Sigma}_i$ the covariates induced by $\mathfrak{D}_{\text{unif}}^i$, which implies that

$$[\mathbf{\Lambda}_0^i]_{(s,a)} \geq \frac{1}{T_i K_i} \frac{W_h^\star(s)}{2SA} \cdot (T_i K_i + K_{\text{unif}}) \geq \frac{W_h^\star(s)}{2SA}$$

for each $(s, a)$ with $s \in \mathcal{S}_0$, and, furthermore, Algorithm 4 collects at most

$$T_i K_i + K_{\text{unif}} + \text{poly}(S, A, H, \log \frac{T_i K_i i^2}{\delta \epsilon_{\text{unif}}}) \cdot \frac{1}{\epsilon_{\text{unif}}} \tag{F.3}$$

episodes. Furthermore, by Lemma 36, we have $\mathbb{P}[\mathcal{E}_{\text{exp}}^i] \geq \delta/2i^2$, so it follows that

$$\mathbb{P}[\cup_{i \geq 1} (\mathcal{E}_{\text{exp}}^i)^c] \leq \sum_{i=1}^\infty \frac{\delta}{8i^2} \leq \delta/4.$$

Henceforth, we therefore assume that $\mathcal{E}_{\text{exp}}^i$ holds for each $i$. This immediately implies that (F.2) holds.

It remains to show that (F.1) is satisfied, and that our sample complexity guarantee is met. To this end we apply Lemma 37 with $\mathbf{\Lambda}_0$ a diagonal matrix, with $[\mathbf{\Lambda}_0]_{(s,a)} = \frac{W_h^\star(s)}{2SA}$ for $s \in \mathcal{S}_0$, and otherwise $[\mathbf{\Lambda}_0]_{(s,a)} = 1$. Note that with this choice of $\mathbf{\Lambda}_0$, by what we just showed above, we have $\mathbf{\Lambda}_0^i \succeq \mathbf{\Lambda}_0$, as required by Lemma 37.

We next turn to bounding the smoothness constants, $\beta$ and $M$. First, note that by Lemma 34, at epoch $i$ we have that all iterates of FWREGRET live in the set $\widehat{\mathbf{\Omega}}_{h, T_i K_i}(\delta/8i^2)$ with probability $1 - \delta/8i^2$. Union bounding over this event for all $i$, with probability at least $1 - \delta/4$, we have that for each $i$ all iterates of FWREGRET live in the set $\widehat{\mathbf{\Omega}}_{h, T_i K_i}(\delta/8i^2)$. By Lemma 35, since we have assumed that $|[\phi]_{(s,a)}| \leq C_\phi \cdot (W_h^\star(s) + \sqrt{\epsilon_\phi})$ for all $(s, a)$ with $s \in \mathcal{S}_0$ and otherwise $[\phi]_{(s,a)} = 0$ for all

$\phi \in \Phi$, we can then bound

$$M_i \leq \max_{s \in \mathcal{S}_0} \left( \frac{2SAC_\phi^2}{C'} + \frac{2SAC_\phi^2 \epsilon_\phi}{C' \cdot W_h^\star(s)} \right) \cdot \left( \frac{2}{C'} + \frac{2}{C'T_iK_iW_h^\star(s)} \cdot \log \frac{SAH}{\delta} \right)$$

$$\beta_i \leq \max_{s \in \mathcal{S}_0}(2\eta_i + 2) \left( \frac{2SAC_\phi^2}{C'} + \frac{2SAC_\phi^2 \epsilon_\phi}{C' \cdot W_h^\star(s)} \right)^2 \cdot \left( \frac{2}{C'} + \frac{2}{C'T_iK_iW_h^\star(s)} \cdot \log \frac{SAH}{\delta} \right)^2$$

On the event $\mathcal{E}_{\exp}^i$, as noted above we have $[\mathbf{\Lambda}_0^i]_{(s,a)} \geq \frac{W_h^\star(s)}{2SA}(1 + \frac{K_{\mathrm{unif}}}{T_iK_i})$ for $s \in \mathcal{S}_0$, so we can take $C' = \frac{1}{2SA}(1 + \frac{K_{\mathrm{unif}}}{T_iK_i})$. We can then bound

$$\max_{s \in \mathcal{S}_0} \left( \frac{2SAC_\phi^2}{C'} + \frac{2SAC_\phi^2 \epsilon_\phi}{C' \cdot W_h^\star(s)} \right) \cdot \left( \frac{2}{C'} + \frac{2}{C'T_iK_iW_h^\star(s)} \cdot \log \frac{SAH}{\delta} \right)$$

$$\leq \left( 4S^2A^2C_\phi + \frac{4S^2A^2C_\phi^2\epsilon_\phi \cdot T_iK_i}{K_{\mathrm{unif}}\epsilon_{\mathrm{unif}}} \right) \cdot \left( 4SA + \frac{4SA}{K_{\mathrm{unif}}\epsilon_{\mathrm{unif}}} \log \frac{SAH}{\delta} \right)$$

where we have used that $W_h^\star(s) \geq \epsilon_{\mathrm{unif}}$ for all $s \in \mathcal{S}_0$, by assumption. By assumption we have $\frac{\epsilon_\phi}{K_{\mathrm{unif}}\epsilon_{\mathrm{unif}}} \leq \epsilon_{\exp}$. Note that by construction, the while statement on Line 3 will ensure that we always have $T_iK_i \leq \mathrm{poly}(S, A, H, C_\phi, \log 1/\delta, \log 1/\epsilon_{\exp}, \log |\Phi|) \cdot \epsilon_{\exp}^{-1}$, so we can bound

$$\epsilon_{\exp} \cdot T_iK_i \leq \mathrm{poly}(S, A, H, C_\phi, \log 1/\delta, \log 1/\epsilon_{\exp}, \log |\Phi|).$$

It follows that it suffices to take

$$\beta, M \leq \mathrm{poly}(S, A, H, C_\phi, \log 1/\delta, \log 1/\epsilon_{\exp}, \log |\Phi|).$$

We now consider two cases. In the first case, when the termination criteria on Line 7 is met, we can apply Lemma 37, to get that with probability at least $1 - \delta/4$ we have that the procedure terminates after running for at most

$$\max \left\{ \min_N 16N \quad \text{s.t.} \quad \inf_{\mathbf{\Lambda} \in \mathbf{\Omega}} \max_{\phi \in \Phi} \phi^\top (N\mathbf{\Lambda} + \mathbf{\Lambda}_0)^{-1}\phi \leq \frac{\epsilon_{\exp}}{6}, \right.$$
$$\left. \frac{\mathrm{poly}(\beta, R, d, H, M, \log 1/\delta, \log 1/\epsilon_{\exp}, \log |\Phi|)}{\epsilon_{\exp}^{4/5}} \right\}$$

$$\leq \max \left\{ \min_N 16N \quad \text{s.t.} \quad \inf_{\mathbf{\Lambda} \in \mathbf{\Omega}} \max_{\phi \in \Phi} \phi^\top (N\mathbf{\Lambda} + \mathbf{\Lambda}_0)^{-1}\phi \leq \frac{\epsilon_{\exp}}{6}, \right.$$
$$\left. \frac{\mathrm{poly}(S, A, H, C_\phi, \log 1/\delta, \log 1/\epsilon_{\exp}, \log |\Phi|)}{\epsilon_{\exp}^{4/5}} \right\}$$

episodes, and returns data $\widehat{\mathbf{\Sigma}}_N$ such that

$$f_{\widehat{i}}(N^{-1}\widehat{\mathbf{\Sigma}}_N) \leq N\epsilon_{\exp},$$

where $\widehat{i}$ is the index of the epoch on which it terminates. By Lemma D.1 of [42], we have

$$\max_{\phi \in \Phi} \|\phi\|_{\mathbf{A}(N^{-1}\widehat{\mathbf{\Sigma}}_N)^{-1}}^2 \leq f_{\widehat{i}}(N^{-1}\widehat{\mathbf{\Sigma}}_N) \leq N\epsilon_{\exp}$$

which implies

$$\max_{\phi \in \Phi} \|\phi\|_{(\widehat{\mathbf{\Sigma}}_N + \mathbf{\Sigma}_{\widehat{i}})^{-1}}^2 \leq \epsilon_{\exp},$$

which proves (F.1). Furthermore, (F.2) holds since as noted $[\mathbf{\Sigma}_i]_{(s,a)} \geq \frac{W_h^\star(s)}{2SA} \cdot (T_iK_i + K_{\mathrm{unif}})$ for each $(s,a)$ with $s \in \mathcal{S}_0$, and since $W_h^\star(s) \geq \epsilon_{\mathrm{unif}}$ for all $s \in \mathcal{S}_0$.

In the second case, when the while loop on Line 3 terminates since $T_iK_i \leq \mathrm{poly}(S, A, H, C_\phi, \log 1/\delta, \log 1/\epsilon_{\exp}, \log |\Phi|) \cdot \epsilon_{\exp}^{-1}$, we can bound the total number of episodes collected within the calls to Algorithm 4 of [43] within the while loop by

$\text{poly}(S, A, H, C_\phi, \log 1/\delta, \log 1/\epsilon_{\exp}, \log |\Phi|) \cdot \epsilon_{\exp}^{-1}$. Furthermore, by Lemma 36, with probability at least $1 - \delta/4$, we have that the call to UNIFEXP on Line 12 terminates after running for at most

$$\frac{8S^2 A^2 C_\phi^2}{\epsilon_{\exp}} + (8S^2 A^2 C_\phi^2 + 1)K_{\text{unif}} + \text{poly}(S, A, H, \log \frac{T_i K_i i^2}{\delta \epsilon_{\text{unif}}}) \cdot \frac{1}{\epsilon_{\text{unif}}}$$

episodes, and that the returned data satisfies $N_h(s, a) \geq \frac{W_h^\star(s)}{2SA} \cdot (\frac{8S^2 A^2 C_\phi^2}{\epsilon_{\exp}} + 8S^2 A^2 C_\phi^2 K_{\text{unif}} + K_{\text{unif}})$. Since $|[\phi]_{(s,a)}| \leq C_\phi \cdot (W_h^\star(s) + \sqrt{\epsilon_\phi})$ and $\epsilon_\phi/(K_{\text{unif}} \epsilon_{\text{unif}}) \leq \epsilon_{\exp}$ by assumption, some manipulation shows that

$$\frac{[\phi]_{(s,a)}^2}{N_h(s, a)} \leq \frac{C_\phi^2 \cdot (W_h^\star(s) + \sqrt{\epsilon_\phi})^2}{\frac{W_h^\star(s)}{2SA} \cdot (\frac{8S^2 A^2 C_\phi^2}{\epsilon_{\exp}} + 8S^2 A^2 C_\phi^2 K_{\text{unif}} + K_{\text{unif}})} \leq \frac{\epsilon_{\exp}}{SA}.$$

It follows then that, letting $\widehat{\Sigma}$ denote the covariance obtained by the call to UNIFEXP on Line 12,

$$\max_{\phi \in \Phi} \|\phi\|_{\widehat{\Sigma}^{-1}}^2 \leq \epsilon_{\exp}$$

as desired. Furthermore, it is straightforward to see that $[\widehat{\Sigma}]_{(s,a)} \geq \frac{\epsilon_{\text{unif}}}{2SA} \cdot K_{\text{unif}}$ for $s \in \mathcal{S}_0$ as well.

To complete the proof, we union bound over these events holding, and take the minimum of the sample complexity bounds from either case. $\qquad \square$

### F.1 Data Conditioning

**Lemma 33.** *Consider running any algorithm for $K$ episodes. Let $K_h(s, a)$ denote the number of visits to $(s, a, h)$. Then with probability at least $1 - \delta$, for all $(s, a, h)$ simultaneously, we have*

$$K_h(s, a) \leq W_h^\star(s)K + \sqrt{2W_h^\star(s)K \cdot \log \frac{SAH}{\delta}} + \log \frac{SAH}{\delta}.$$

*Proof.* By definition, we have

$$\sup_\pi w_h^\pi(s) = W_h^\star(s).$$

This implies that any policy will reach $(s, h)$ with probability at most $W_h^\star(s)$. We can therefore think of this as the sum of Bernoullis with parameter at most $W_h^\star(s)$, so the bound follows by applying Bernstein's inequality and a union bound. $\qquad \square$

**Lemma 34.** *Consider the set*

$$\widehat{\Omega}_{h,K}(\delta) := \left\{ \text{diag}(\boldsymbol{v}) \ : \ \boldsymbol{v} \in \mathbb{R}_+^{SA}, [\boldsymbol{v}]_{(s,a)} \leq W_h^\star(s) + \sqrt{\frac{2W_h^\star(s)}{K} \cdot \log \frac{SAH}{\delta}} + \frac{1}{K} \log \frac{SAH}{\delta} \right\}.$$

*Consider running some set of policies for $K$ episodes, and let $\widehat{\Lambda}$ be defined as*

$$\widehat{\Lambda}_h = \text{diag}(\widehat{\boldsymbol{v}}), \quad [\boldsymbol{v}]_{(s,a)} = \frac{K_h(s, a)}{K}.$$

*Then with probability at least $1 - \delta$, we have that $\widehat{\Lambda}_h \in \widehat{\Omega}_{h,K}(\delta)$ for all $h \in [H]$ simultaneously.*

*Proof.* This is an immediate consequence of Lemma 33. $\qquad \square$

We will denote $\widehat{\Omega}_{h,K} := \widehat{\Omega}_{h,K}(\delta)$ when the choice of $\delta$ is clear from context.

**Lemma 35.** *Consider the function*

$$f(\boldsymbol{\Lambda}) = \frac{1}{\eta} \log \left( \sum_{\phi \in \Phi} e^{\eta \|\phi\|_{\mathbf{A}(\boldsymbol{\Lambda})^{-1}}^2} \right) \quad \text{for} \quad \mathbf{A}(\boldsymbol{\Lambda}) = \boldsymbol{\Lambda} + \boldsymbol{\Lambda}_0$$

*Assume that for all $\phi \in \Phi$ we have*

$$\max_{\phi \in \Phi} |[\phi]_{(s,a)}| \leq C_\phi \cdot (W_h^\star(s) + \epsilon), \quad \forall s \in \mathcal{S}_0$$

*for some $\mathcal{S}_0$ and some $C_\phi, \epsilon > 0$, and otherwise $[\phi]_{(s,a)} = 0$. Assume that $\mathbf{\Lambda}_0 = \mathrm{diag}(\mathbf{v})$ for some $\mathbf{v}$ satisfying*

$$[\mathbf{v}]_{(s,a)} \geq C' \cdot W_h^\star(s), \quad \forall s \in \mathcal{S}_0$$

*and otherwise $[\mathbf{v}]_{(s,a)} \geq \lambda$, for some $C', \lambda > 0$. Then we can bound*

$$\sup_{\widehat{\mathbf{\Lambda}}, \widehat{\mathbf{\Lambda}}' \in \widehat{\mathbf{\Omega}}_{h,K}} |\nabla_{\mathbf{\Lambda}} f(\mathbf{\Lambda})|_{\mathbf{\Lambda} = \widehat{\mathbf{\Lambda}}}[\widehat{\mathbf{\Lambda}}']|$$

$$\leq \max_{s \in \mathcal{S}_0} \left( \frac{2SAC_\phi^2}{C'} + \frac{2SAC_\phi^2 \epsilon^2}{C' \cdot W_h^\star(s)} \right) \cdot \left( \frac{2}{C'} + \frac{2}{C'KW_h^\star(s)} \cdot \log \frac{SAH}{\delta} \right)$$

*and*

$$\sup_{\widehat{\mathbf{\Lambda}}, \widehat{\mathbf{\Lambda}}', \widehat{\mathbf{\Lambda}}'' \in \widehat{\mathbf{\Omega}}_{h,K}} |\nabla_{\mathbf{\Lambda}}^2 f(\mathbf{\Lambda})|_{\mathbf{\Lambda} = \widehat{\mathbf{\Lambda}}}[\widehat{\mathbf{\Lambda}}', \widehat{\mathbf{\Lambda}}'']|$$

$$\leq \max_{s \in \mathcal{S}_0} (2 + 2\eta) \left( \frac{2SAC_\phi^2}{C'} + \frac{2SAC_\phi^2 \epsilon^2}{C' \cdot W_h^\star(s)} \right)^2 \cdot \left( \frac{2}{C'} + \frac{2}{C'KW_h^\star(s)} \cdot \log \frac{SAH}{\delta} \right)^2.$$

*Proof.* By Lemma D.5 of [42], we have that

$$\nabla_{\mathbf{\Lambda}} f(\mathbf{\Lambda})|_{\mathbf{\Lambda} = \widehat{\mathbf{\Lambda}}}[\widehat{\mathbf{\Lambda}}'] = - \left( \sum_{\phi \in \Phi} e^{\eta \|\phi\|_{\mathbf{A}(\widehat{\mathbf{\Lambda}})^{-1}}^2} \right) \cdot \sum_{\phi \in \Phi} e^{\eta \|\phi\|_{\mathbf{A}(\widehat{\mathbf{\Lambda}})^{-1}}^2} \phi^\top \mathbf{A}(\widehat{\mathbf{\Lambda}})^{-1} \widehat{\mathbf{\Lambda}}' \mathbf{A}(\widehat{\mathbf{\Lambda}})^{-1} \phi.$$

We have

$$\phi^\top \mathbf{A}(\widehat{\mathbf{\Lambda}})^{-1} \widehat{\mathbf{\Lambda}}' \mathbf{A}(\widehat{\mathbf{\Lambda}})^{-1} \phi = \sum_{s,a} \frac{[\phi]_{(s,a)}^2 \cdot [\widehat{\mathbf{\Lambda}}']_{(s,a)}}{[\mathbf{A}(\widehat{\mathbf{\Lambda}})]_{(s,a)}^2} = \sum_{s \in \mathcal{S}_0} \sum_a \frac{[\phi]_{(s,a)}^2 \cdot [\widehat{\mathbf{\Lambda}}']_{(s,a)}}{[\mathbf{A}(\widehat{\mathbf{\Lambda}})]_{(s,a)}^2}$$

where the last equality follows since, for $s \notin \mathcal{S}_0$, we have assumed $[\phi]_{(s,a)} = 0$.

Now consider some $s \in \mathcal{S}_0$. By assumption we have $[\phi]_{(s,a)}^2 \leq 2C_\phi^2 \cdot (W_h^\star(s)^2 + \epsilon^2)$ and by our assumption on $\mathbf{\Lambda}_0$ we can lower bound $[\mathbf{A}(\widehat{\mathbf{\Lambda}})]_{(s,a)} \geq C' \cdot W_h^\star(s)$. Furthermore, since $\widehat{\mathbf{\Lambda}}' \in \widehat{\mathbf{\Omega}}_{h,K}$, we have

$$[\widehat{\mathbf{\Lambda}}']_{(s,a)} \leq W_h^\star(s) + \sqrt{\frac{2W_h^\star(s)}{K} \cdot \log \frac{SAH}{\delta}} + \frac{1}{K} \log \frac{SAH}{\delta}$$
$$\leq 2W_h^\star(s) + \frac{2}{K} \log \frac{SAH}{\delta}.$$

Putting this together, we have

$$\frac{[\phi]_{(s,a)}^2 \cdot [\widehat{\mathbf{\Lambda}}']_{(s,a)}}{[\mathbf{A}(\widehat{\mathbf{\Lambda}})]_{(s,a)}^2} \leq \frac{4C_\phi^2 \cdot (W_h^\star(s)^2 + \epsilon^2) \cdot (W_h^\star(s) + \frac{1}{K} \log \frac{SAH}{\delta})}{(C' \cdot W_h^\star(s))^2}$$
$$\leq \left( \frac{2C_\phi^2}{C'} + \frac{2C_\phi^2 \epsilon^2}{C'W_h^\star(s)} \right) \cdot \left( \frac{2}{C'} + \frac{2}{C'KW_h^\star(s)} \log \frac{SAH}{\delta} \right).$$

It follows that

$$\sum_{s \in \mathcal{S}_0} \sum_a \frac{[\phi]_{(s,a)}^2 \cdot [\widehat{\mathbf{\Lambda}}']_{(s,a)}}{[\mathbf{A}(\widehat{\mathbf{\Lambda}})]_{(s,a)}^2} \leq \max_{s \in \mathcal{S}_0} \left( \frac{2SAC_\phi^2}{C'} + \frac{2SAC_\phi^2 \epsilon^2}{C'W_h^\star(s)} \right) \cdot \left( \frac{2}{C'} + \frac{2}{C'KW_h^\star(s)} \log \frac{SAH}{\delta} \right).$$

The second bound follows in an analogous fashion, using the expression for the second derivative given in Lemma D.5 of [42].

$\square$

---

**Algorithm 4** Uniform Exploration (UNIFEXP)

---

**input:** tolerance $\epsilon_{\text{unif}}$, reruns $K$, confidence $\delta$, step $h$
$\mathfrak{D} \leftarrow \emptyset$
**for** $(s, a) \in \mathcal{S} \times \mathcal{A}$ **do**
    // LEARN2EXPLORE is as defined in [46]
    $\{(\mathcal{X}_j, \Pi_j, N_j)\}_{j=1}^{\lceil \log_2 1/\epsilon_{\text{unif}} \rceil} \leftarrow$ LEARN2EXPLORE$(\{(s, a)\}, h, \frac{\delta}{2SA}, \frac{\delta}{2KSA}, \epsilon_{\text{unif}})$
    **if** $\exists j_{sa}$ such that $(s, a) \in \mathcal{X}_{j_{sa}}$ **then**
        Rerun every policy in $\Pi_{j_{sa}}$ $K_{sa} := \lceil \frac{K}{SA|\Pi_{j_{sa}}|} \rceil$ times, store observed transitions in $\mathfrak{D}$
    **end if**
**end for**
**return** $\mathfrak{D}$

---

**Lemma 36.** *With probability at least $1 - \delta$, Algorithm 4 will terminate after running for at most*

$$K + \text{poly}(S, A, H, \log \frac{K}{\delta \epsilon_{\text{unif}}}) \cdot \frac{1}{\epsilon_{\text{unif}}}$$

*episodes and will collect at least $\frac{W_h^\star(s)K}{2SA}$ samples from each $(s, a)$ such that $W_h^\star(s) > \epsilon_{\text{unif}}$.*

*Proof.* By Theorem 13 of [46], with probability at least $1 - \delta/2SA$, for any $(s, a)$:

- LEARN2EXPLORE will run for at most $\text{poly}(S, A, H, \log \frac{K}{\delta \epsilon_{\text{unif}}}) \cdot \frac{1}{\epsilon_{\text{unif}}}$ episodes.

- Rerunning every policy in $\Pi_{j_{sa}}$ once, with probability at least $1 - \delta/K$ we will collect $N = 2^{-j_{sa}}|\Pi_{j_{sa}}|$ samples from $(s, a)$, for $|\Pi_{j_{sa}}| = \mathcal{O}(2^{j_{sa}} \cdot S^3 A^2 H^4 \log^3 1/\delta)$.

- We have that $W_h^\star(s) \leq 2^{-j_{sa}+1}$.

- IF $(s, a) \notin \mathcal{X}_j$ for all $j = 1, 2, \ldots, \lceil \log 1/\epsilon_{\text{unif}} \rceil$, then $W_h^\star(s) \leq \epsilon_{\text{unif}}$.

By the above conclusions, rerunning policies in $\Pi_{j_{sa}}$ on Line 7, with probability at least $1 - \delta/2SA$ we will collect

$$N \cdot K_{sa} \geq N \cdot \frac{K}{SA|\Pi_{j_{sa}}|} = \frac{2^{-j_{sa}}K}{SA}$$

samples from $(s, a)$. As noted, $W_h^\star(s) \leq 2^{-j_{sa}+1}$, so this implies that we will collect at least $\frac{W_h^\star(s)K}{2SA}$ samples from $(s, a)$. Union bounding over this holding for all $(s, a)$, and noting that we only fail to collect this many samples if $W_h^\star(s) \leq \epsilon_{\text{unif}}$ gives the collection guarantee.

To bound the total number of episodes, we note that the procedure on Line 7 will, in total collect at most

$$\sum_{s,a : j_{sa} \text{ exists}} |\Pi_{j_{sa}}| \lceil K_{sa} \rceil \leq \sum_{s,a : j_{sa} \text{ exists}} |\Pi_{j_{sa}}| + \sum_{s,a} \frac{K}{SA} = \sum_{s,a} |\Pi_{j_{sa}}| + K$$

episodes. IF $j_{sa}$ exists, this implies that $|\Pi_{j_{sa}}| \leq \mathcal{O}(2^{j_{sa}} \cdot S^3 A^2 H^4 \log^3 1/\delta)$, and since $j_{sa} \in \{1, 2, \ldots, \lceil \log 1/\epsilon_{\text{unif}} \rceil\}$, this implies that the above is bounded by

$$K + \mathcal{O}(\epsilon_{\text{unif}}^{-1} \cdot S^3 A^2 H^4 \log^3 1/\delta).$$

Combining this with our bound on the total number of episodes collected by LEARN2EXPLORE, we have that the number of episodes collected by Algorithm 4 is bounded by

$$K + \text{poly}(S, A, H, \log \frac{K}{\delta \epsilon_{\text{unif}}}) \cdot \frac{1}{\epsilon_{\text{unif}}}.$$

$\square$

### F.2 Online Frank-Wolfe

**Lemma 37.** *Let*

$$f_i(\mathbf{\Lambda}) = \frac{1}{\eta_i} \log \left( \sum_{\boldsymbol{\phi} \in \Phi} e^{\eta_i \|\boldsymbol{\phi}\|^2_{\mathbf{A}_i(\mathbf{\Lambda})^{-1}}} \right), \quad \mathbf{A}_i(\mathbf{\Lambda}) = \mathbf{\Lambda} + \frac{1}{T_i K_i} \mathbf{\Lambda}_{0,i}$$

*for some $\mathbf{\Lambda}_{0,i}$ satisfying $\mathbf{\Lambda}_{0,i} \succeq \mathbf{\Lambda}_0$ for all $i$, and $\eta_i = 2^{2i/5}$. Let $(\beta_i, M_i)$ denote the smoothness and magnitude constants for $f_i$. Let $(\beta, M)$ be some values such that $\beta_i \le \eta_i \beta, M_i \le M$ for all $i$, and $R$ the diameter of the domain of possible values of $\mathbf{\Lambda}$.*

*Then, if we run Algorithm 4 of [43] on $(f_i)_i$ with constraint tolerance $\epsilon$ and confidence $\delta$ and $K_i = T_i = 2^i$, we have that with probability at least $1 - \delta$, it will run for at most*

$$\max \left\{ \min_N 16N \ s.t. \ \inf_{\mathbf{\Lambda} \in \Omega} \max_{\boldsymbol{\phi} \in \Phi} \boldsymbol{\phi}^\top (N\mathbf{\Lambda} + \mathbf{\Lambda}_0)^{-1} \boldsymbol{\phi} \le \frac{\epsilon}{6}, \frac{\mathrm{poly}(\beta, R, d, H, M, \log 1/\delta, \log |\Phi|)}{\epsilon^{4/5}} \right\}.$$

*episodes, and will return data $\{\boldsymbol{\phi}_\tau\}_{\tau=1}^N$ with covariance $\widehat{\mathbf{\Sigma}}_N = \sum_{\tau=1}^N \boldsymbol{\phi}_\tau \boldsymbol{\phi}_\tau^\top$ such that*

$$f_{\widehat{i}}(N^{-1}\widehat{\mathbf{\Sigma}}_N) \le N\epsilon,$$

*where $\widehat{i}$ is the iteration on which* OPTCOV *terminates.*

*Proof.* Our goal is to simply find a setting of $i$ that is sufficiently large to guarantee the condition $f_i(\widehat{\mathbf{\Lambda}}_i) \le K_i T_i \epsilon$ is met. By Lemma C.1 of [43], we have with probability at least $1 - \delta/(2i^2)$:

$$f_i(\widehat{\mathbf{\Lambda}}_i) \le \inf_{\mathbf{\Lambda} \in \Omega} f_i(\mathbf{\Lambda}) + \frac{\beta_i R^2 (\log T_i + 3)}{2T_i} + \sqrt{\frac{4M^2 \log(8i^2 T_i/\delta)}{K_i}}$$

$$+ \sqrt{\frac{c_1 M^2 d^4 H^4 \log^3(8i^2 H K_i T_i/\delta)}{K_i}} + \frac{c_2 M d^4 H^3 \log^{7/2}(4i^2 H K_i T_i/\delta)}{K_i}$$

$$\le 3 \max \left\{ \inf_{\mathbf{\Lambda} \in \Omega} f_i(\mathbf{\Lambda}), \frac{\beta_i R^2 (\log T_i + 3)}{2T_i}, \sqrt{\frac{4M^2 \log(8i^2 T_i/\delta)}{K_i}} \right.$$

$$\left. + \sqrt{\frac{c_1 M^2 d^4 H^4 \log^3(8i^2 H K_i T_i/\delta)}{K_i}} + \frac{c_2 M d^4 H^3 \log^{7/2}(4i^2 H K_i T_i/\delta)}{K_i} \right\}.$$

So a sufficient condition for $f_i(\widehat{\mathbf{\Lambda}}_i) \le K_i T_i \epsilon$ is that

$$K_i T_i \ge \frac{3}{\epsilon} \max \left\{ \inf_{\mathbf{\Lambda} \in \Omega} f_i(\mathbf{\Lambda}), \frac{\beta_i R^2 (\log T_i + 3)}{2T_i}, \sqrt{\frac{4M^2 \log(8i^2 T_i/\delta)}{K_i}} \right.$$

$$\left. + \sqrt{\frac{c_1 M^2 d^4 H^4 \log^3(8i^2 H K_i T_i/\delta)}{K_i}} + \frac{c_2 M d^4 H^3 \log^{7/2}(4i^2 H K_i T_i/\delta)}{K_i} \right\}. \tag{F.4}$$

Recall that

$$f_i(\mathbf{\Lambda}) = \frac{1}{\eta_i} \log \left( \sum_{\boldsymbol{\phi} \in \Phi} e^{\eta_i \|\boldsymbol{\phi}\|^2_{\mathbf{A}_i(\mathbf{\Lambda})^{-1}}} \right), \quad \mathbf{A}_i(\mathbf{\Lambda}) = \mathbf{\Lambda} + \frac{1}{T_i K_i} \mathbf{\Lambda}_{0,i}.$$

By Lemma D.1 of [42], we can bound

$$\max_{\boldsymbol{\phi} \in \Phi} \|\boldsymbol{\phi}\|^2_{\mathbf{A}_i(\mathbf{\Lambda})^{-1}} \le f_i(\mathbf{\Lambda}) \le \max_{\boldsymbol{\phi} \in \Phi} \|\boldsymbol{\phi}\|^2_{\mathbf{A}_i(\mathbf{\Lambda})^{-1}} + \frac{\log |\Phi|}{\eta_i}.$$

Thus,

$$\inf_{\mathbf{\Lambda} \in \Omega} f_i(\mathbf{\Lambda}) \le \inf_{\mathbf{\Lambda} \in \Omega} \max_{\boldsymbol{\phi} \in \Phi} \|\boldsymbol{\phi}\|^2_{\mathbf{A}_i(\mathbf{\Lambda})^{-1}} + \frac{\log |\Phi|}{\eta_i}$$

$$= \inf_{\mathbf{\Lambda} \in \Omega} \max_{\boldsymbol{\phi} \in \Phi} T_i K_i \boldsymbol{\phi}^\top (T_i K_i \mathbf{\Lambda} + \mathbf{\Lambda}_{0,i} + \mathbf{\Lambda}_{\mathrm{off}})^{-1} \boldsymbol{\phi} + \frac{\log |\Phi|}{\eta_i}$$

By our choice of $\eta_i = 2^{2i/5}$, and $K_i = 2^i$, $T_i = 2^i$, we can ensure that

$$K_i T_i \geq \frac{6}{\epsilon} \frac{\log |\Phi|}{\eta_i}$$

as long as $i \geq \frac{2}{5} \log_2[\frac{6 \log |\Phi|}{\epsilon}]$. To ensure that

$$T_i K_i \geq \frac{6}{\epsilon} \inf_{\boldsymbol{\Lambda} \in \boldsymbol{\Omega}} \max_{\boldsymbol{\phi} \in \Phi} T_i K_i \boldsymbol{\phi}^\top (T_i K_i \boldsymbol{\Lambda} + \boldsymbol{\Lambda}_{0,i})^{-1} \boldsymbol{\phi}$$

it suffices to take

$$i \geq \arg\min_i \quad \text{s.t.} \quad \inf_{\boldsymbol{\Lambda} \in \boldsymbol{\Omega}} \max_{\boldsymbol{\phi} \in \Phi} \boldsymbol{\phi}^\top (2^{3i} \boldsymbol{\Lambda} + \boldsymbol{\Lambda}_{0,i})^{-1} \boldsymbol{\phi} \leq \frac{\epsilon}{6}.$$

Since we assume that we can lower bound $\boldsymbol{\Lambda}_{0,i} \succeq \boldsymbol{\Lambda}_0$ for each $i$, so this can be further simplified to

$$i \geq \arg\min_i \quad \text{s.t.} \quad \inf_{\boldsymbol{\Lambda} \in \boldsymbol{\Omega}} \max_{\boldsymbol{\phi} \in \Phi} \boldsymbol{\phi}^\top (2^{3i} \boldsymbol{\Lambda} + \boldsymbol{\Lambda}_0)^{-1} \boldsymbol{\phi} \leq \frac{\epsilon}{6}. \tag{F.5}$$

We next want to show that

$$T_i K_i \geq \frac{3}{\epsilon} \cdot \frac{\beta_i R^2 (\log T_i + 3)}{2 T_i}.$$

Bounding $\beta_i \leq \eta_i \beta$, a sufficient condition for this is that

$$i \geq \frac{2}{5} \left( \log_2(12\beta R^2 i) + \log_2 \frac{1}{\epsilon} \right).$$

By Lemma A.1 of [43], it suffices to take

$$i \geq \frac{6}{5} \log_2(9\beta R^2 \log_2 \frac{1}{\epsilon}) + \frac{2}{5} \log_2 \frac{1}{\epsilon} \tag{F.6}$$

to meet this condition (this assumes that $12\beta R^2 \geq 1$ and $\frac{2}{5} \log_2 \frac{1}{\epsilon} \geq 1$—if either of these is not the case we can just replace them with 1 without changing the validity of the final result).

Finally, we want to ensure that

$$T_i K_i \geq \frac{3}{\epsilon} \left( \sqrt{\frac{4M^2 \log(8i^2 T_i/\delta)}{K_i}} \right.$$
$$\left. + \sqrt{\frac{c_1 M^2 d^4 H^4 \log^3(8i^2 H K_i T_i/\delta)}{K_i}} + \frac{c_2 M d^4 H^3 \log^{7/2}(4i^2 H K_i T_i/\delta)}{K_i} \right).$$

To guarantee this, it suffices that

$$2^{5i/2} \geq \frac{c}{\epsilon} \sqrt{M^2 d^4 H^4 i^3 \log^3(iH/\delta)}, \quad 2^{3i} \geq \frac{c}{\epsilon} \cdot M d^4 H^3 i^{7/2} \log^{7/2}(iH/\delta).$$

or

$$i \geq \frac{4}{5} \log_2(cMdHi \log(H/\delta)) + \frac{2}{5} \log_2 \frac{1}{\epsilon}, \quad i \geq \frac{4}{3} \log_2(cMdH \log(H/\delta)) + \frac{1}{3} \log_2 \frac{1}{\epsilon}.$$

By Lemma A.1 of [43], it then suffices to take

$$i \geq \frac{12}{5} \log(cMdH \log(H/\delta) \log_2 1/\epsilon) + \frac{2}{5} \log_2 \frac{1}{\epsilon},$$
$$i \geq 4 \log_2(cMdH \log(H/\delta) \log_2 1/\epsilon) + \frac{1}{3} \log_2 \frac{1}{\epsilon} \tag{F.7}$$

Thus, a sufficient condition to guarantee (F.4) is that $i$ is large enough to satisfy (F.5), (F.6), and (F.7) and $i \geq \frac{2}{5} \log_2[\frac{6 \log |\Phi|}{\epsilon}]$.

If $\widehat{i}$ is the final round, the total complexity scales as

$$\sum_{i=1}^{\widehat{i}} T_i K_i = \sum_{i=1}^{\widehat{i}} 2^{2i} \leq 2 \cdot 2^{2\widehat{i}}.$$

Using the sufficient condition on $i$ given above, we can bound the total complexity as

$$\max \left\{ \min_N 16N \text{ s.t. } \inf_{\boldsymbol{\Lambda} \in \boldsymbol{\Omega}} \max_{\boldsymbol{\phi} \in \Phi} \boldsymbol{\phi}^\top (N\boldsymbol{\Lambda} + \boldsymbol{\Lambda}_0)^{-1} \boldsymbol{\phi} \leq \frac{\epsilon}{6}, \frac{\text{poly}(\beta, R, d, H, M, \log 1/\delta, \log |\Phi|)}{\epsilon^{4/5}} \right\}.$$

$\square$

## F.3 Pruning Hard-to-Reach States

---
**Algorithm 5** PRUNE: Prune Hard-to-Reach States

---
**input:** tolerance $\epsilon_{\text{unif}}$, confidence $\delta$
$\mathcal{S}^{\text{keep}} \leftarrow \emptyset$
**for** $h \in [H]$ **do**
  **for** $s \in \mathcal{S}$ **do**
    // LEARN2EXPLORE is as defined in [46]
    $\{(\mathcal{X}_j, \Pi_j, N_j)\}_{j=1}^{\lceil \log_2 \frac{1}{32\epsilon_{\text{unif}}} \rceil} \leftarrow$ LEARN2EXPLORE$(\{(s,a)\}, h, \frac{\delta}{SH}, \frac{1}{2}, 32\epsilon_{\text{unif}})$ for any $a \in$
    $\mathcal{A}$
    **if** $\exists j_s$ such that $(s,a) \in \mathcal{X}_{j_s}$ **then**
      $\mathcal{S}^{\text{keep}} = \mathcal{S}^{\text{keep}} \cup \{(s,h)\}$
    **end if**
  **end for**
**end for**
**return** $\mathcal{S}^{\text{keep}}$

---

**Lemma 38.** *With probability at least $1 - \delta$, Algorithm 5 will terminate after running for at most*

$$\text{poly}(S, A, H, \log \frac{1}{\delta\epsilon_{\text{unif}}}) \cdot \frac{1}{\epsilon_{\text{unif}}}$$

*episodes and will return a set $\mathcal{S}^{\text{keep}}$ such that, for every $(s,h) \in \mathcal{S}^{\text{keep}}$, we have $W_h^\star(s) \geq \epsilon_{\text{unif}}$, and, if $(s,h) \notin \mathcal{S}^{\text{keep}}$, then $W_h^\star(s) \leq 32\epsilon_{\text{unif}}$.*

*Proof.* As in Lemma 36, by Theorem 13 of [46], with probability at least $1 - \delta/SH$, for any $(s,h)$:

- LEARN2EXPLORE will run for at most $\text{poly}(S, A, H, \log \frac{1}{\delta\epsilon_{\text{unif}}}) \cdot \frac{1}{\epsilon_{\text{unif}}}$ episodes.

- Rerunning every policy in $\Pi_{j_s}$ once, with probability at least $1/2$ we will collect $N = 2^{-j_s}|\Pi_{j_s}|$ samples from $(s,a,h)$.

- If $(s,a) \notin \mathcal{X}_j$ for all $j = 1, 2, \ldots, \lceil \log 1/\epsilon_{\text{unif}} \rceil$, then $W_h^\star(s) \leq 32\epsilon_{\text{unif}}$.

We union bound over this event holding for all $(s,h)$, which occurs with probability at least $1 - \delta$.

It is immediate by the last property that, if $(s,h) \notin \mathcal{S}^{\text{keep}}$ then $W_h^\star(s) \leq 32\epsilon_{\text{unif}}$.

We next show that if $(s,h) \in \mathcal{S}^{\text{keep}}$, then this implies that $W_h^\star(s) \geq \epsilon_{\text{unif}}$. Let $X$ be a random variable denoting the total number of samples we collect from $(s,a,h)$ when rerunning all policies in $\Pi_{j_s}$. Then by Markov's Inequality, by the above properties we have

$$\frac{1}{2} \leq \mathbb{P}[X \geq N_{j_s}/2] \leq \frac{2\mathbb{E}[X]}{N_{j_s}} \leq \frac{2|\Pi_{j_s}|W_h^\star(s)}{N_{j_s}} = 8 \cdot 2^{j_s} W_h^\star(s).$$

It follows that

$$W_h^\star(s) \geq \frac{1}{16 \cdot 2^{j_s}} \geq \frac{1}{16 \cdot 2^{\lceil \log_2 \frac{1}{32\epsilon_{\text{unif}}} \rceil}} \geq \frac{1}{32 \cdot 2^{\log_2 \frac{1}{32\epsilon_{\text{unif}}}}} = \epsilon_{\text{unif}}.$$

This completes the proof.

$\square$

