# OpenReview forum: "Sample Complexity Reduction via Policy Difference Estimation in Tabular Reinforcement Learning"
_NeurIPS.cc/2024/Conference — NeurIPS 2024 spotlight_

### Official Review · Reviewer_FF9h · 2024-07-12

**Soundness:** 4
**Presentation:** 4
**Contribution:** 3
**Rating:** 7
**Confidence:** 2

**Summary:**

The work is interested in answering the following question: "In the RL setting, can we identify the best policy faster if we only estimate the difference between the value of individual policies?". The paper provides a positive answer in the contextual bandit setting and a negative, but more nuanced answer for tabular RL. Exploiting the difference between value estimates of a reference policy and target policies, the authors propose an algorithm that achieves a better sample complexity than the best one in the literature so far, while matching the sample complexity achieved for contextual bandit in special settings.

**Strengths:**

- **Relevant Problem**: The question of reaching the optimal sample complexity of identifying the best policy (with high probability) is an important theoretical question that can have a big practical impact.

- **Novel, important results**: The authors analyse the $(\epsilon, \delta)$-PAC policy identification for tabular RL, give valuable insight about the problem and derive an algorithm that improves on the best sample complexity obtained so far.

- **The paper is beautifully written**: I genuinely enjoyed reading the first two sections. The authors do a magnificent job at introducing the problem and position their contribution in the literature. The next sections get more technical but they are still pleasant to read. The authors invest in sharing their intuition with the reader to alleviate the technical difficulties. The paper is well structured, and helps you understand the thought process of the authors which is highly appreciated.

**Weaknesses:**

- **The proposed algorithm is complex to implement**: As pointed out by the authors, enumerating all policies from $\Pi$ can make the algorithm impractical. This narrows the applicability of the algorithm to simple problems even if I understand that the primary goal of the paper is of a theoretical nature.

**Questions:**

- In an A/B test scenario, we have only two policies that are compared. How does the PERP algorithm improve on naively looking at the difference (or relative difference) of the estimated value of A and B?

- Intuitively, what makes the tabular RL way more difficult than the bandit setting? Can it be alleviated by adding more structure? Do we have the same guarantees for contextual bandit, with say, a linear assumption?

- The PERP algorithm is complex and encapsulates other algorithms in the inner loop. Is there a way to reduce the complexity of the algorithm and come up with a more practical variant even if we lose some of the guarantees?

**Limitations:**

N/A.

---

> ### Author Rebuttal · Authors · 2024-08-05
>
> Thank you for your efforts to review the paper and for the positive feedback. Please find answers inline to your questions below.
>
> **Gains for simpler setting with two policies**
> > In an A/B test scenario, we have only two policies that are compared. How does the PERP algorithm improve on naively looking at the difference (or relative difference) of the estimated value of A and B?
>
> In the case of two policies, if we naively rolled out each policy to estimate the difference in values, we would require $O(\frac{1}{\epsilon^2})$ samples from each policy to attain $\epsilon$-optimality in the value estimate for $\pi_1$ or $\pi_2$. Our PERP complexity would be much smaller than this in cases where the policies agree on many states.
>
> For instance, in the example from Figure~1, we have two policies that disagree on only the low-probability red states. Here, we attain a complexity of $O(\frac{1}{\epsilon})$ rather than $O(\frac{1}{\epsilon^2})$.
>
> **Comparison of tabular RL difficulty with contextual bandit**
> > Intuitively, what makes the tabular RL way more difficult than the bandit setting? Can it be alleviated by adding more structure? Do we have the same guarantees for contextual bandit, with say, a linear assumption?
>
> Tabular RL is more difficult than the contextual bandit setting because of the cost of estimating the transitions – this is a well-known observation in the existing literature. In the contextual bandit setting, we show (Corollary 1) that the cost of learning the context distribution is at most the cost of learning the rewards. However, in the tabular RL setting, the story is exactly reversed - the cost of learning the rewards is at most that of learning the transitions.
>
> If there is additional structure imposed, the sample complexity would accordingly reduce. For instance, when the transitions are action-independent, the complexity from Corollary 2 is the same as that from Corollary 1 for contextual bandits. For other types of structural assumptions, the complexity would depend on the type of assumption and it is difficult to say much.
>
> **Computational complexity of PERP**
> > The PERP algorithm is complex and encapsulates other algorithms in the inner loop. Is there a way to reduce the complexity of the algorithm and come up with a more practical variant even if we lose some of the guarantees?
>
> While the complexity of PERP could likely be reduced somewhat (for example, some of the subroutines it relies on were originally developed for linear MDPs, and could likely be simplified for tabular MDPs), at present each component of PERP seems critical for achieving the stated complexity. Developing simpler and easier-to-implement algorithms with similar sample complexity is an interesting direction for future work.
>
> Please let us know if any of your questions were not clarified, and we would be happy to elaborate further.

---

> > ### Comment · Reviewer_FF9h · 2024-08-12
> >
> > Thank you for clarifying the points raised. I think that this paper is worth sharing with the NeuRIPS community, I will keep my score.

---

### Official Review · Reviewer_76qf · 2024-07-13

**Soundness:** 3
**Presentation:** 3
**Contribution:** 4
**Rating:** 7
**Confidence:** 2

**Summary:**

The author investigate if estimating the difference in policies is sufficient in determining the best policy for contexual bandits and tabular RL.
A (somewhat) practical algorithm is proposed to determine the number of samples needed without any unknown quantities.

**Strengths:**

- The motivating example clearly explained the difference between estimating policy values directly vs their difference
- Identifying the when $\rho_\Pi$  is sufficient in determining the optimal policy in is novel
- Limitations of the algorithm is clear

**Weaknesses:**

- Some intuition could have been provided to describe certain value such as $U(\pi, \bar{\pi})$ or $\hat{\delta}^\pi_h$ to make the resulting bounds easier to interpret.
- In section 4.2,  it is difficult to see why the difference estimate has reduced variance

**Questions:**

- In Lemma 1, what is the difference when $E^M[\tau]$ when compared to $E[\tau]$ on line 182?
- Could an example be provided of an MDP with Action-Independent Transitions. It's unclear to me how a sub-optimal policy can exist in this setting.

**Limitations:**

No concerns.

---

> ### Author Rebuttal · Authors · 2024-08-05
>
> Thank you for taking the time to review our paper and for the positive feedback. Please find answers inline to your questions below.
>
> **Intuition for terms**
>
> > Some intuition could have been provided to describe certain value such as $U(\pi, \bar{\pi})$ or $\delta_h^\pi$ to make the resulting bounds easier to interpret.
>
> The notation $\delta_h^\pi(s) = w_h^\pi(s) - w_h^{\bar{\pi}}(s)$ is used to refer to the difference in state visitations between policy $\pi$ and the reference policy $\bar{\pi}$. We will include additional description on this in the paper.
>
> The $U(\pi, \bar{\pi})$ term is, to the best of our knowledge, novel in the literature. As we argue in Section 4.2, however, this term naturally arises when computing the variance of our estimator used to estimate the difference between the value of two policies. More precisely, this term corresponds to the cost of estimating where $\bar{\pi}$ visits, if our goal is to estimate the difference in value between policy $\pi$ and $\bar{\pi}$. If for a given state $s$ the actions taken by $\pi$ and $\bar{\pi}$ achieve the same long-term reward, then it is not critical that the frequency with which $\bar{\pi}$ visits this state is estimated, as it does not affect the difference in values between $\pi$ and $\bar{\pi}$; if the actions take by $\pi$ and $\bar{\pi}$ do achieve different long-term reward at $s$, then we must estimate the behavior of each policy at this state. This is reflected by the term in the expectation of $U(\pi,\bar{\pi})$: $(Q_h^\pi(s,\pi_h(s)) - Q_h^{\pi}(s, \bar{\pi}_h(s)))^2$; this will be 0 in the former case, and scale with the difference between long-term action reward in the latter case. We will add this additional intuition to the final version.
>
> **Clarifications on results**
>
> > In section 4.2, it is difficult to see why the difference estimate has reduced variance.
>
> The PERP algorithm, which uses the differences estimator, achieves the sample complexity from Theorem 1, which we show can be arbitrarily smaller than that of prior work such as [40]. We attribute this reduction in complexity to the reduced variance of the estimator. The intuition for this is that only states where policies differ (“red states” in Figure 1) contribute to the upper bound for PERP, whereas all states contribute to the upper bound in [40].
>
> > In Lemma 1, what is the difference when $\mathbf{E}^{\mathcal{M}}[\tau]$ when compared to $\mathbf{E}[\tau]$ on line 182?
>
> Lemma 1 is making a claim about the specific MDP $\mathcal{M}$ from Figure 1, which is why we make the dependence on $\mathcal{M}$ explicit here. Line 182 refers to the lower bound from [2], which applies to any MDP, which is why we drop the $\mathcal{M}$ for notational convenience. Note that Lemma 1 shows that there exist instances where the bound from [2] is not tight. We will make this more explicit in the text.
>
> > Could an example be provided of an MDP with Action-Independent Transitions. It's unclear to me how a sub-optimal policy can exist in this setting.
>
> In this class of MDPs that we consider for Corollary 2, the transition dynamics are independent of actions chosen by the agent but the rewards are not; the case of contextual bandits is a special case of this class of MDPs. Hence, policies which choose actions with low rewards would be suboptimal.
>
> Please let us know if any of your questions were not clarified, and we would be happy to elaborate further.

---

> > ### Comment · Reviewer_76qf · 2024-08-12
> >
> > I appreciate the authors for their response and for the clarification. I will keep my decision as it.

---

### Official Review · Reviewer_f5AC · 2024-07-22

**Soundness:** 3
**Presentation:** 4
**Contribution:** 3
**Rating:** 8
**Confidence:** 4

**Summary:**

This paper studies the problem in tabular RL: finding an \epislon-optimal policy given a set of policy with high probability. The author proposed with a new lower-bound of this problem and explained why the previous one is not correct with an example. The author proposed one algorithm PERP that computes reference policy, measures the difference between policies, and refines policy set through episodes. The author provided sample complexity analysis.

**Strengths:**

This paper is well presented. The idea is conveyed clearly.
The paper proposes a new lower bound and provide one example that is easy to understand. The explanation of algorithm is clear.

**Weaknesses:**

The paper didn't provide the computational complexity of the algorithm. It would be better to mention the computational complexity by the end of the section describing algorithm.

**Questions:**

How the sample complexity change when abitrary choosing a reference policy? The U(\pi, \bar{\pi}) should be upper bounded by a constant value, if I could arbitrary choose a reference policy, does it mean that I sacrifice sample complexity for computational complexity? If so, how much? Since it seems costly computationally when I iterate through the whole policy set to find a reference policy but the sample complexity won't decrease much. Please correct me, thanks.

**Limitations:**

The authors have confronted the limitations in section 7.

---

> ### Author Rebuttal · Authors · 2024-08-05
>
> Thank you for taking the time to review our paper and for the positive feedback. Please find answers to your questions below.
>
> **Computational Complexity**
>
> > The paper didn't provide the computational complexity of the algorithm
>
> Thank you for pointing this out, and we will include additional description on the computational complexity in the final version of the paper, if accepted. The computational complexity of PERP is $\text{poly}\left(S, A, H, \frac{1}{\epsilon}, |\Pi|, \log \left(\frac{1}{\delta}\right)\right)$. The primary contributor to the computational complexity is the use of the Frank-Wolfe algorithm for experiment design in the *OnlineExpD* subroutine. Lemma 37 from [1] shows that the number of iterations of the Frank-Wolfe algorithm is bounded polynomially in problem parameters, and from the definition of this procedure given in [1], we see that each iteration of Frank-Wolfe has computational complexity polynomial in problem parameters.
>
> [1] Wagenmaker, Andrew, and Aldo Pacchiano. "Leveraging offline data in online reinforcement learning." International Conference on Machine Learning. PMLR, 2023.
>
> **Impact of fixing reference policy**
>
> > How the sample complexity change when abitrary choosing a reference policy? The U(\pi, \bar{\pi}) should be upper bounded by a constant value, if I could arbitrary choose a reference policy, does it mean that I sacrifice sample complexity for computational complexity? Since it seems costly computationally when I iterate through the whole policy set to find a reference policy but the sample complexity won't decrease much.
>
> If we fix the reference policy to a fixed $\bar{\pi}$, the numerator of the second term of the sample complexity would be replaced by $U(\pi, \bar{\pi})$ as you observe and, additionally, the numerator of the first term would scale with $\phi_h^{\bar{\pi}} - \phi_h^\pi$ rather than $\phi_h^{\star} - \phi_h^\pi$. However, the computational complexity would not be significantly affected since we need to iterate over $\Pi$ in other parts of the algorithm (lines 7, 12, 18) as well, so the computational complexity will still scale as $|\Pi|$. It is an exciting direction of work to design more practical algorithms that incorporate experiment design for exploration.
>
> Please let us know if any of your questions were not clarified, and we would be happy to elaborate.

---

> > ### Comment · Reviewer_f5AC · 2024-08-13
> >
> > Thanks for your reply. I will keep my scores and think it would be a good paper to appear at Neurips.

---

### Decision · Program_Chairs · 2024-09-25

**Decision:**

Accept (spotlight)

**Comment:**

This paper presents a very interesting finding regarding the sample complexity in PAC RL vs the gap of values among policies. A brilliant piece of theoretical work that is interesting to the RL community at large.